# Informative Outlier Matters: Robustifying Out-of-distribution Detection Using Outlier Mining

## Abstract

Detecting out-of-distribution (OOD) inputs is critical for safely deploying deep learning models in an open-world setting. However, existing OOD detection solutions can be brittle in the open world, facing various types of adversarial OOD inputs. While methods leveraging auxiliary OOD data have emerged, our analysis reveals a key insight that the majority of auxiliary OOD examples may not meaningfully improve the decision boundary of the OOD detector. In this paper, we provide a theoretically motivated method, *Adversarial Training with informative Outlier Mining* (ATOM), which improves the robustness of OOD detection. We show that, by mining informative auxiliary OOD data, one can significantly improve OOD detection performance, and somewhat surprisingly, generalize to unseen adversarial attacks. ATOM achieves state-of-the-art performance under a broad family of classic and adversarial OOD evaluation tasks. For example, on the CIFAR-10 in-distribution dataset, ATOM reduces the FPR95 by up to 57.99% under adversarial OOD inputs, surpassing the previous best baseline by a large margin.

## 1 Introduction

Out-of-distribution (OOD) detection has become an indispensable part of building reliable open-world machine learning models (Amodei et al., 2016). An OOD detector determines whether an input is from the same distribution as the training data, or a different distribution (*i.e.*, out-of-distribution). The performance of the OOD detector is central for safety-critical applications such as autonomous driving (Eykholt et al., 2018) or rare disease identification (Blauwkamp et al., 2019).

Despite exciting progress made in OOD detection, previous methods mostly focused on clean OOD data (Hendrycks & Gimpel, 2016; Liang et al., 2018; Lee et al., 2018; Lakshminarayanan et al., 2017; Hendrycks et al., 2018; Mohseni et al., 2020). Scant attention has been paid to the robustness aspect of OOD detection. Recent works (Hein et al., 2019; Sehwag et al., 2019; Bitterwolf et al., 2020) considered worst-case OOD detection under adversarial perturbations (Papernot et al., 2016; Goodfellow et al., 2014; Biggio et al., 2013; Szegedy et al., 2013). For example, an OOD image (*e.g.*, mailbox) can be perturbed to be misclassified by the OOD detector as in-distribution (traffic sign data). Such an adversarial OOD example is then passed to the image classifier and trigger undesirable prediction and action (*e.g.*, speed limit 70). Therefore, it remains an important question to make out-of-distribution detection algorithms robust in the presence of small perturbations to OOD inputs.

In this paper, we begin with formally formulating the task of robust OOD detection and providing theoretical analysis in a simple Gaussian data model. While recent OOD detection methods (Hendrycks et al., 2018; Hein et al., 2019; Meinke & Hein, 2019; Mohseni et al., 2020) have leveraged auxiliary OOD data, they often sample randomly uniformly from the auxiliary dataset. Contrary to the common practice, our analysis reveals a key insight that *the majority of auxiliary OOD examples may not provide useful information to improve the decision boundary of OOD detector*. Under a Gaussian model of the data, we theoretically show that using outlier mining significantly improves the error bound of OOD detector in the presence of non-informative auxiliary OOD data.

Motivated by this insight, we propose *Adversarial Training with informative Outlier Mining* (ATOM), which justifies the theoretical intuitions above and achieves state-of-the-art performance on a broad

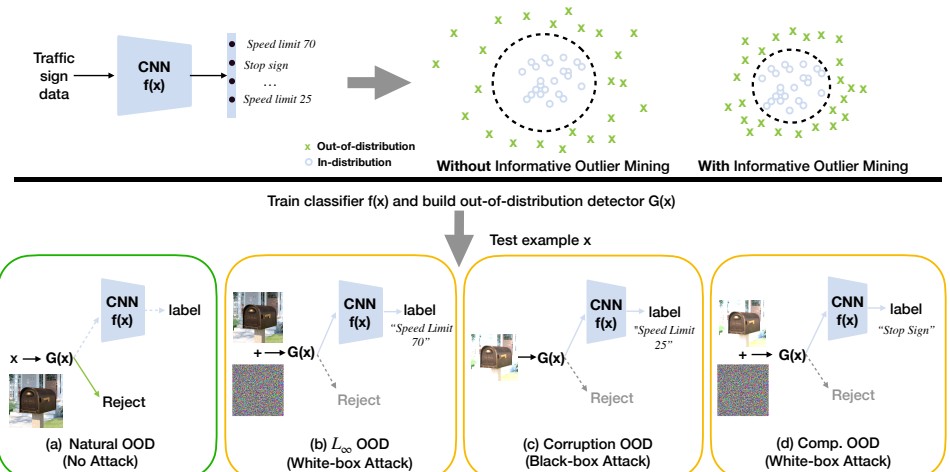

Figure 1: When deploying an image classification system (OOD detector $G(x)$ + image classifier $f(x)$) in an open world, there can be multiple types of out-of-distribution examples. We consider a broad family of OOD inputs, including (a) Natural OOD, (b) $L_\infty$ OOD, (c) corruption OOD, and (d) Compositional OOD. A detailed description of these OOD inputs can be found in Section 5.1. In (b-d), a perturbed OOD input (e.g., a perturbed mailbox image) can mislead the OOD detector to classify it as an in-distribution sample. This can trigger the downstream image classifier $f(x)$ to predict it as one of the in-distribution classes (e.g., speed limit 70). Through *adversarial training with informative outlier mining* (ATOM), our method can robustify the decision boundary of OOD detector $G(x)$, which leads to improved performance across all types of OOD inputs. Solid lines are actual computation flow.

family of classic and adversarial OOD evaluation tasks for modern neural networks. We show that, by carefully choosing which OOD data to train on, one can significantly improve the robustness of an OOD detector, and somewhat surprisingly, generalize to unseen adversarial attacks. We note that while hard negative mining has been extensively used in various learning tasks such as object recognition (Felzenszwalb et al., 2009; Gidaris & Komodakis, 2015; Shrivastava et al., 2016), to the best of our knowledge, we are the first to exploit the novel connection between hard example mining and OOD detection. We show both empirically and theoretically that hard example mining significantly improves the generalization and robustness of OOD detection.

To evaluate our method, we provide a unified framework that allows examining the robustness of OOD detection algorithms under a broad family of OOD inputs, as illustrated in Figure 1. Our evaluation includes existing classic OOD evaluation task – Natural OOD, and adversarial OOD evaluation task – $L_\infty$ OOD. Besides, we also introduce new adversarial OOD evaluation tasks – Corruption OOD and Compositional OOD. Under these evaluation tasks, ATOM achieves state-of-the-art performance compared to eight competitive OOD detection methods (refer to Appendix B.3 for a detailed description of these methods). On the Natural OOD evaluation task, ATOM achieves comparable and often better performance than current state-of-the-art methods. On $L_\infty$ OOD evaluation task, ATOM outperforms current state-of-the-art method ACET by a large margin (e.g. on CIFAR-10, outperforms it by **53.9%**). Under the new Corruption OOD evaluation task, where the attack is unknown during training time, ATOM also achieves much better results than previous methods (e.g. on CIFAR-10, outperform previous best method by **30.99%**). While almost every method fails under the hardest Compositional OOD evaluation task, ATOM still achieves impressive results (e.g. on CIFAR-10, reduce the FPR by **57.99%**). The performance is noteworthy since ATOM is not trained explicitly on corrupted OOD inputs.

In summary, our contributions are:

- Firstly, we contribute theoretical analysis formalizing the intuition of mining hard outliers for improving the robustness of OOD detection.
- Secondly, we contribute a theoretically motivated method, ATOM, which leads to state-of-the-art performance on both classic and adversarial OOD evaluation tasks. We conduct extensive evaluations and ablation analysis to demonstrate the effectiveness of informative outlier mining.
- Lastly, we provide a unified evaluation framework that allows future research examining the robustness of OOD detection algorithms under a broad family of OOD inputs.

## 2 PRELIMINARIES

In this section, we formulate the problem of robust out-of-distribution detection, and provide background knowledge on the use of auxiliary data for OOD detection.

**Problem Statement.** We consider a training dataset $\mathcal{D}_{\text{in}}^{\text{train}}$ drawn i.i.d. from a data distribution $P_{\mathbf{X},Y}$, where $\mathbf{X}$ is the sample space and $Y = \{1, 2, \cdots, K\}$ is the set of labels. A classifier $f(\mathbf{x})$ is trained on the in-distribution, $P_{\mathbf{X}}$, the marginal distribution of $P_{\mathbf{X},Y}$. The OOD examples are revealed during test time, which are from a different distribution $Q_{\mathbf{X}}$, potentially with perturbations added. The task of *robust out-of-distribution detection* is to learn a detector $G : \mathbf{x} \to \{-1, 1\}$, which outputs 1 for $\mathbf{x}$ from $P_{\mathbf{X}}$ and output $-1$ for a clean or perturbed OOD example $\mathbf{x}$ from $Q_{\mathbf{X}}$.

Formally, let $\Omega(\mathbf{x})$ be a set of small perturbations on an OOD example $\mathbf{x}$. The detector is evaluated on $\mathbf{x}$ from $P_{\mathbf{X}}$ and on the worst-case input inside $\Omega(\mathbf{x})$ for an OOD example from $Q_{\mathbf{X}}$. The false negative rate (FNR) and false positive rate (FPR) are defined as:

$$\text{FNR}(G) = \mathbb{E}_{\mathbf{x} \sim P_{\mathbf{X}}} \mathbb{I}[G(\mathbf{x}) = -1], \quad \text{FPR}(G; Q_{\mathbf{X}}, \Omega) = \mathbb{E}_{\mathbf{x} \sim Q_{\mathbf{X}}} \max_{\delta \in \Omega(\mathbf{x})} \mathbb{I}[G(\mathbf{x} + \delta) = 1]. \quad (1)$$

Note that no data from the test OOD distribution $Q_{\mathbf{X}}$ are available for training.

**Use of Auxiliary Data for OOD Detection** While it is impossible to anticipate the test OOD data distributions $Q_{\mathbf{X}}$ for training, recent works (Hendrycks et al., 2018; Hein et al., 2019; Meinke & Hein, 2019; Mohseni et al., 2020; Liu et al., 2020) have shown the promise using auxiliary data as a proxy for estimating the decision boundary between in- vs. OOD data. The idea is illustrated in Figure 1, where outlier data is randomly sampled to regularize the model outputs (*e.g.,* low confidence for OOD data and high confidence for in-distribution data). Formally, we assume the auxiliary OOD dataset $\mathcal{D}_{\text{out}}^{\text{auxiliary}}$ is sampled from a different distribution $U_{\mathbf{X}}$. The difference between the auxiliary data $U_{\mathbf{X}}$ and test OOD data $P_{\mathbf{X}}$ raises the fundamental question of how to effectively leverage $\mathcal{D}_{\text{out}}^{\text{auxiliary}}$ for improving learning the decision boundary between in- vs. OOD data.

## 3 THEORETICAL ANALYSIS: INFORMATIVE OUTLIERS MATTER

In this section, we present a theoretical analysis[1] that motivates the use of informative outlier mining for OOD detection. To establish formal guarantees, we use a Gaussian data model to model data $P_{\mathbf{X}}, Q_{\mathbf{X}}$, and $U_{\mathbf{X}}$. Different from previous work by Schmidt et al. (2018) and Carmon et al. (2019), our analysis gives rise to a *separation result* with or without informative outlier mining for OOD detection. To this end, we note that while hard negative mining has been explored in different domains of learning (e.g. object detection, deep metric learning, please refer to Section 6 for details), the vast literature of out-of-distribution detection has not explored this idea. Moreover, most uses of hard negative mining are on a heuristic basis, but in our case, the simplicity of the definition of OOD (see Section 2) allows us to derive precise formal guarantees, which further differs us from previous studies of hard negative mining. As a remark, our analysis also establishes formal evidence of the importance of using auxiliary outlier data for OOD detection, which is lacking in the current OOD detection studies. We refer readers to Section A for these results.

At a high level, our analysis provides two important insights: **(1)** First, we show that a detection algorithm can work very well if *all* data is informative; yet it can fail completely in a natural setting where we mix informative auxiliary data with non-informative auxiliary data. **(2)** Second, we show that, tweaking the algorithm with simple thresholding to choose mildly hard auxiliary data (in our setting they are exactly the informative ones), can lead to good detection performance. Combining both thus provides direct evidence about the importance of hard negative mining for OOD.

**Gaussian Data Model.** We now describe the Gaussian data model, inspired by the model in (Schmidt et al., 2018; Carmon et al., 2019), but with important adjustment to the OOD detection setting. In particular, our setting has a family $\mathcal{Q}$ of possible test OOD distributions and have only in-distribution data for training, modeling that the test OOD distribution is unknown for training. Given $\mu \in \mathbb{R}^d, \sigma > 0, \nu > 0$, we consider the following model:

- $P_{\mathbf{X}}$ (**in-distribution data**): $\mathcal{N}(\mu, \sigma^2 I)$; The in-distribution data $\{\mathbf{x}_i\}_{i=1}^n$ is drawn from $P_{\mathbf{X}}$.
- $Q_{\mathbf{X}}$ (**out-of-distribution data**) can be any distribution from the family $\mathcal{Q} = \{\mathcal{N}(-\mu + v, \sigma^2 I) : v \in \mathbb{R}^d, \|v\|_2 \le \nu\}$.

---

[1]Due to lack of space, proofs are deferred to Appendix A.

- **Hypothesis class of OOD detector**: $\mathcal{G} = \{G_\theta(\mathbf{x}) = \text{sign}(\theta^\top \mathbf{x}) : \theta \in \mathbb{R}^d\}$.

A concrete instance of the model is defined by a set of parameter values for $d$, $\mu$, $\sigma$, and $\nu$; see Appendix A.2 for the family of instances we analyze. While the Gaussian model may be much simpler than the practical data, its simplicity is desirable for our analytical purpose for demonstrating the insights. Furthermore, the analysis in this simple model has implications for more complicated and practical methods, which we present in Section 4. Finally, the analysis can be generalized to mixtures of Gaussians which models practical data much better. Below, we consider the FNR and the FPR under $\ell_\infty$ perturbations of magnitude $\epsilon$. Since $Q_{\mathbf{X}}$ is not accessible at training time, our goal is to bound $\sup_{Q_{\mathbf{X}} \in \mathcal{Q}} \text{FPR}(G; Q_{\mathbf{X}}, \Omega_{\infty,\epsilon}(\mathbf{x}))$.

**Failing a good detector by mixing non-informative auxiliary data**. We start by considering the case where *all* auxiliary data are informative: That is, all auxiliary $\{\mathbf{x}'_i\}_{i=1}^{n'}$ come from uniform mixture of the possible test OOD distributions in $\mathcal{Q}$. In this case, it is straightforward to show that a simple averaging-based detector,

$$\hat{\theta}_{n,n'} = \frac{1}{n + n'} \left( \sum_{i=1}^{n} \mathbf{x}_i - \sum_{i=1}^{n'} \tilde{\mathbf{x}}_i \right), \tag{2}$$

performs very well (See Proposition 4 in the appendix).

Unfortunately, this detector can be easily failed by considering the following simple auxiliary data distribution $U_{\text{mix}}$, which mixes the ideal auxiliary data with *non-informative data*

- $U_{\text{mix}}$ **(Non-ideal mixture)**: $U_{\text{mix}}$ is a uniform mixture of $\mathcal{N}(-\mu, \sigma^2 I)$ and $\mathcal{N}(\mu_o, \sigma^2 I)$ with $\mu_o = 10\mu$.

Importantly, the distribution $U_{\text{mix}}$ models the case where the auxiliary OOD data has some non-informative outliers, and also with a small probability mass of samples (e.g., tail of $\mathcal{N}(-\mu, \sigma^2 I)$) in the support of in-distribution. In this case, the simple average method leads to $\mathbb{E}[\hat{\theta}_{n,n'}] = -7\mu/4$ with a large error, since $\hat{\theta}_{n,n'}$ is misled by auxiliary data from $\mathcal{N}(\mu_o, \sigma^2 I)$ or tail of $\mathcal{N}(-\mu, \sigma^2 I)$ in the support of in-distribution.

**Fixing the detector with informative outlier mining**. We now show an important modification to the detection algorithm by using informative outlier mining, which leads to good detection performance. Specifically, we first use in-distribution data to get an intermediate solution: $\hat{\theta}_{\text{int}} = \frac{1}{n} \sum_{i=1}^{n} \mathbf{x}_i$. Then, we use a simple thresholding mechanism to only pick points with *mild* confidence scores, which removes *non-informative outliers*. Specifically, we only select outliers $\tilde{\mathbf{x}}$ whose confidence scores $f(\tilde{\mathbf{x}}) = 1/(1 + e^{-\tilde{\mathbf{x}}^\top \hat{\theta}_{\text{int}}/d})$ fall in an interval $[a, b]$. The final solution $\hat{\theta}_{\text{om}}$ is $-1$ times the average of the selected outliers. We can prove the following:

**Proposition 1. (Error bound with outlier mining.)** *For any $\epsilon \in (0, 1/2)$ and any integer $n_0 > 0$, there exist a family of instances of the Gaussian data model such that the following is true. $n'$ auxiliary OOD data from $U_{\text{mix}}$ specified above. There exist thresholds $a$ and $b$ for $\hat{\theta}_{\text{om}}$ and a universal constant $c > 0$ such that if the number of in-distribution data $n \geq c(n_0 \log d + \sqrt{d n_0})$ and the number of auxiliary data $n' \geq (d + n_0 \cdot 4\epsilon^2)\sqrt{d/n_0}$, then $\hat{\theta}_{\text{om}}$ has small errors:*[2]

$$\mathbb{E}_{\hat{\theta}_{\text{om}}} \text{FNR}(G_{\hat{\theta}_{\text{om}}}) \leq 10^{-3}, \quad \mathbb{E}_{\hat{\theta}_{\text{om}}} \sup_{Q_{\mathbf{X}} \in \mathcal{Q}} \text{FPR}(G_{\hat{\theta}_{\text{om}}}; Q_{\mathbf{X}}, \Omega_{\infty,\epsilon}(\mathbf{x})) \leq 10^{-3}. \tag{3}$$

Intuitively, the mining method removes the misleading points (most points in $\mathcal{N}(\mu_o, \sigma^2 I)$ and tail of $\mathcal{N}(-\mu, \sigma^2 I)$ in the support of in-distribution). Outliers selected in this way are mostly informative and thus give an accurate final detector, which justifies outlier mining in the presence of non-informative data. If we compare this bound with that for the case without auxiliary data (Proposition 3), we can see that for sufficiently high dimension $d$, with the same amount of in-distribution data, any algorithm without outliers must fail but our outlier mining method can learn a good detector. We also note that our analysis and the result also hold for many other auxiliary data distributions $U_{\text{mix}}$, and the particular $U_{\text{mix}}$ used here is for the simplicity of demonstration; see the appendix for more discussions. In the following section, we design a practical algorithm based on this insight and present empirical evidence of its effectiveness.

---

[2]The error bound in the proposition can be made arbitrarily small and with high probability. The current bound is presented for simplicity.

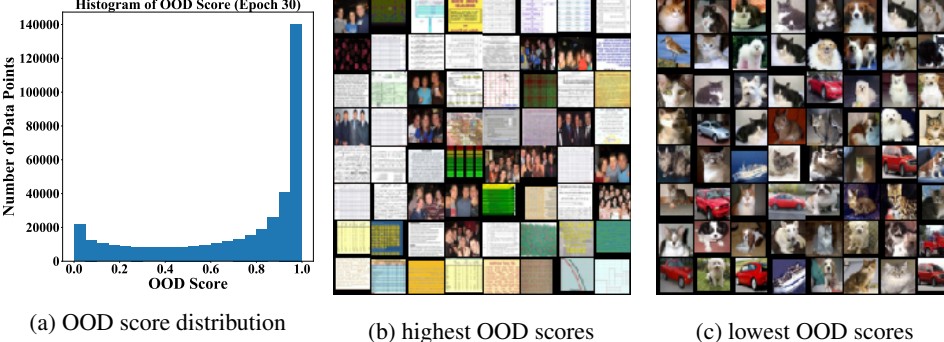

(a) OOD score distribution     (b) highest OOD scores     (c) lowest OOD scores

Figure 2: On CIFAR-10, we train a DenseNet with objective (4) for 100 epochs **without** informative outlier mining. At epoch 30, we randomly sample 400,000 data points from $\mathcal{D}_{\text{out}}^{\text{auxiliary}}$, and plot the OOD score frequency distribution (a). We observe that the model quickly converges to solution where OOD score distribution becomes dominated by *easy* examples with score closer to 1, as shown in (b). Therefore, training on these easy OOD data points can no longer help improve the decision boundary of OOD detector. (c) shows the hardest examples mined from TinyImages w.r.t CIFAR-10.

## 4   ATOM: ADVERSARIAL TRAINING WITH INFORMATIVE OUTLIER MINING

In this section, we introduce *Adversarial Training with informative Outlier Mining* (ATOM), which justifies our theoretical analysis and effectiveness in the context of modern neural networks. We first present the adversarial training objective and then describe how we use informative outlier mining to robustify OOD detection.

**Training Objective.** We consider a $(K + 1)$-way classifier network $\hat{f}$, where the $(K + 1)$-th class label indicates out-of-distribution class. Denote by $\hat{F}_\theta(\mathbf{x})$ the softmax output of $\hat{f}$ on $\mathbf{x}$. The robust training objective is given by

$$\underset{\theta}{\text{minimize}} \quad \mathbb{E}_{(\mathbf{x},y)\sim\mathcal{D}_{\text{in}}^{\text{train}}}[\ell(\mathbf{x}, y; \hat{F}_\theta)] + \lambda \cdot \mathbb{E}_{\mathbf{x}\sim\mathcal{D}_{\text{out}}^{\text{train}}} \max_{\mathbf{x}'\in\Omega_{\infty,\epsilon}(\mathbf{x})}[\ell(\mathbf{x}', K + 1; \hat{F}_\theta)], \quad (4)$$

where $\ell$ is the cross entropy loss, and $\mathcal{D}_{\text{out}}^{\text{train}}$ is the OOD training dataset. We use Projected Gradient Descent (PGD) (Madry et al., 2017) to solve the inner max of the objective, and apply it to half of a minibatch while keeping the other half clean to ensure performance on both clean and perturbed data. Once trained, the OOD detector $G(\mathbf{x})$ can be constructed by:

$$G(\mathbf{x}) = \begin{cases} -1 & \text{if } \hat{F}(\mathbf{x})_{K+1} \geq \gamma, \\ 1 & \text{if } \hat{F}(\mathbf{x})_{K+1} < \gamma, \end{cases} \quad (5)$$

where $\gamma$ is the threshold, and in practice can be chosen on the in-distribution data so that a high fraction of the test examples are correctly classified by $G$. We call $\hat{F}(\mathbf{x})_{K+1}$ the *OOD score* of $\mathbf{x}$. For an input that is labeled as in-distribution by $G$, one can obtain its semantic label using $F(\mathbf{x})$:

$$F(\mathbf{x}) = \underset{y\in\{1,2,\cdots,K\}}{\arg\max} \hat{F}(\mathbf{x})_y \quad (6)$$

**Informative Outlier Mining.** Motivated by our theoretical analysis in Section 3 and our empirical observation shown in Figure 2, we propose to adaptively choose OOD training examples where the detector is uncertain about. We provide the complete training algorithm using informative outlier mining in Algorithm 1. Our method is different from random sampling as used in previous works (Hendrycks et al., 2018; Hein et al., 2019; Meinke & Hein, 2019; Mohseni et al., 2020). Specifically, during each training epoch, we randomly sample $N$ data points from the auxiliary OOD dataset $\mathcal{D}_{\text{out}}^{\text{auxiliary}}$, and use the current model to infer the OOD scores[3]. Next, we sort the data points according to the OOD scores and select a subset of $n < N$ data points, starting with the $qN^{\text{th}}$ data in the sorted list. We then use the selected samples as OOD training data $\mathcal{D}_{\text{out}}^{\text{train}}$ for the next epoch of training. Intuitively, $q$ determines the *informativeness* of the sampled points w.r.t the OOD detector. The larger $q$ is, the less informative those sampled examples become. Note that informative outlier

---

[3]Since the inference stage can be fully parallel, outlier mining can be applied with relatively low overhead.

---

**Algorithm 1** ATOM: Adversarial Training with informative Outlier Mining

---

**input** $\mathcal{D}_{\text{in}}^{\text{train}}, \mathcal{D}_{\text{out}}^{\text{auxiliary}}, \hat{F}_{\theta}, m, N, n, q$
**output** $F, G$
    **for** $t = 1, 2, \cdots, m$ **do**
        Randomly sample $N$ data points from $\mathcal{D}_{\text{out}}^{\text{auxiliary}}$ to get a candidate set $\mathcal{S}$.
        Compute OOD scores on $\mathcal{S}$ using current model $\hat{F}_{\theta}$ to get set $V = \{\hat{F}(\mathbf{x})_{K+1} \mid \mathbf{x} \in \mathcal{S}\}$.
        Sort scores in $V$ from the lowest to the highest.
        $\mathcal{D}_{\text{out}}^{\text{train}} \leftarrow V[qN : qN + n]$     $\triangleright \{q \in [0, 1 - n/N]\}$
        Train $\hat{F}_{\theta}$ for one epoch using the training objective of (4).
    **end for**
    Build $G$ and $F$ using (5) and (6) respectively.

---

mining is performed on (non-adversarial) auxiliary OOD data. Selected examples are then used in the robust training objective (4).

To see how informative outlier mining alone improves the OOD detection, we also consider the following objective without adversarial training:

$$\underset{\theta}{\text{minimize}} \quad \mathbb{E}_{(\mathbf{x}, y) \sim \mathcal{D}_{\text{in}}^{\text{train}}}[\ell(\mathbf{x}, y; \hat{F}_{\theta})] + \lambda \cdot \mathbb{E}_{\mathbf{x} \sim \mathcal{D}_{\text{out}}^{\text{train}}}[\ell(\mathbf{x}, K + 1; \hat{F}_{\theta})], \tag{7}$$

which we name *Natural Training with informative Outlier Mining* (NTOM).

## 5 EXPERIMENTS

In this section, we describe our experimental setup (Section 5.1) and show that ATOM can substantially improve OOD detection performance on both clean OOD data and adversarially perturbed OOD inputs. We also conducted extensive ablation analysis to explore different aspects of our algorithm. Our experiments are mainly on image data, which is common in previous work, but we believe that our insights and method can be applied to other types of data, which is left for future work.

### 5.1 SETUP

**In-distribution Datasets.** We use SVHN (Netzer et al., 2011), CIFAR-10, and CIFAR-100 (Krizhevsky et al., 2009) datasets as in-distribution datasets.

**Auxiliary OOD Datasets.** By default, we use 80 Million Tiny Images (TinyImages) (Torralba et al., 2008) as $\mathcal{D}_{\text{out}}^{\text{auxiliary}}$, which is a common setting in prior works. We also use ImageNet-RC, a variant of ImageNet (Chrabaszcz et al., 2017) as an alternative auxiliary OOD dataset.

**Out-of-distribution Datasets.** For OOD test dataset, we follow the procedure in (Liang et al., 2018; Hendrycks et al., 2018) and use six different natural image datasets. For CIFAR-10 and CIFAR-100, we use SVHN, Textures (Cimpoi et al., 2014), Places365 (Zhou et al., 2017), LSUN (crop), LSUN (resize) (Yu et al., 2015), and iSUN (Xu et al., 2015). For SVHN, we use CIFAR-10, Textures, Places365, LSUN (crop), LSUN (resize), and iSUN. Besides natural image datasets, we also consider Gaussian Noise and Uniform Noise as OOD test data.

**Hyperparameters.** The hyperparameter $q$ is chosen on a separate validation set from TinyImages, which does not depend on test-time OOD data (see Appendix B.7). Based on the validation results in Table 5, we set $q = 0$ for SVHN, $q = 0.125$ for CIFAR-10 and $q = 0.5$ for CIFAR-100. To ensure fair comparison, in each epoch, ATOM uses the same amount of outlier data as OE, where $n$ is twice larger than the in-distribution data size (i.e., 50,000). For all experiments, we set $\lambda = 1$. For CIFAR-10 and CIFAR-100, we set $N = 400,000$, and $n = 100,000$; For SVHN, we set $N = 586,056$, and $n = 146,514$. More details about experimental set up are in Appendix B.1.

**Robust OOD Evaluation Tasks.** We consider the following family of OOD inputs, for which we provide visualizations in Appendix B.5:

- **Natural OOD**: This is equivalent to the classic OOD evaluation with clean OOD input $\mathbf{x}$, and $\Omega = \emptyset$.
- $L_{\infty}$ **attacked OOD (white-box)**: We consider small $L_{\infty}$-norm bounded perturbations on OOD input $\mathbf{x}$ (Madry et al., 2017; Athalye et al., 2018), which induce the model to produce high confidence scores (or low OOD scores) for OOD inputs. We denote the adversarial

| $\mathcal{D}_{in}^{test}$ | Method | FPR (5% FNR) ↓ | AUROC ↑ | FPR (5% FNR) ↓ | AUROC ↑ | FPR (5% FNR) ↓ | AUROC ↑ | FPR (5% FNR) ↓ | AUROC ↑ |
|---|---|---|---|---|---|---|---|---|---|
| | | Natural OOD | | Corruption OOD | | $L_\infty$ OOD | | Comp. OOD | |
| **SVHN** | MSP | 38.84 | 93.57 | 99.68 | 68.48 | 99.89 | 1.39 | 100.00 | 0.19 |
| | ODIN | 31.45 | 93.52 | 97.11 | 63.21 | 99.86 | 0.61 | 100.00 | 0.05 |
| | Mahalanobis | 22.80 | 95.57 | 93.14 | 60.78 | 97.33 | 8.89 | 99.89 | 0.23 |
| | SOFL | 0.06 | 99.98 | 3.78 | 99.07 | 75.31 | 46.78 | 99.81 | 2.75 |
| | OE | 0.60 | 99.88 | 23.44 | 96.23 | 69.36 | 52.19 | 99.65 | 1.27 |
| | ACET | 0.49 | 99.91 | 17.03 | 97.23 | 29.33 | 86.75 | 99.85 | 5.13 |
| | CCU | 0.50 | 99.90 | 24.17 | 96.11 | 52.17 | 62.24 | 99.42 | 1.60 |
| | ROWL | 2.04 | 98.87 | 55.03 | 72.37 | 77.24 | 61.27 | 99.79 | **50.00** |
| | NTOM (ours) | **0.04** | **99.98** | **2.87** | **99.16** | 60.28 | 64.06 | 99.78 | 1.50 |
| | ATOM (ours) | 0.07 | 99.97 | 5.47 | 98.52 | **7.02** | **98.00** | **96.33** | 49.52 |
| **CIFAR-10** | MSP | 50.54 | 91.79 | 100.00 | 58.35 | 100.00 | 13.82 | 100.00 | 13.67 |
| | ODIN | 21.65 | 94.66 | 99.37 | 51.44 | 99.99 | 0.18 | 100.00 | 0.01 |
| | Mahalanobis | 26.95 | 90.30 | 91.92 | 43.94 | 95.07 | 12.47 | 99.88 | 1.58 |
| | SOFL | 2.78 | 99.04 | 62.07 | 88.65 | 99.98 | 1.01 | 100.00 | 0.76 |
| | OE | 3.66 | 98.82 | 56.25 | 90.66 | 99.94 | 0.34 | 99.99 | 0.16 |
| | ACET | 12.28 | 97.67 | 66.93 | 88.43 | 74.45 | 78.05 | 96.88 | 53.71 |
| | CCU | 3.39 | 98.92 | 56.76 | 89.38 | 99.91 | 0.35 | 99.97 | 0.21 |
| | ROWL | 25.03 | 86.96 | 94.34 | 52.31 | 99.98 | 49.49 | 100.00 | 49.48 |
| | NTOM (ours) | 1.87 | **99.28** | 30.58 | 94.67 | 99.90 | 1.22 | 99.99 | 0.45 |
| | ATOM (ours) | **1.69** | 99.20 | **25.26** | **95.29** | 20.55 | 88.94 | 38.89 | 86.71 |
| **CIFAR-100** | MSP | 78.05 | 76.11 | 100.00 | 30.04 | 100.00 | 2.25 | 100.00 | 2.06 |
| | ODIN | 56.77 | 83.62 | 100.00 | 36.95 | 100.00 | 0.14 | 100.00 | 0.00 |
| | Mahalanobis | 42.63 | 87.86 | 95.92 | 42.96 | 95.44 | 15.87 | 99.86 | 2.08 |
| | SOFL | 43.36 | 91.21 | 99.93 | 45.23 | 100.00 | 0.35 | 100.00 | 0.27 |
| | OE | 49.21 | 88.05 | 99.96 | 45.01 | 100.00 | 0.94 | 100.00 | 0.59 |
| | ACET | 50.93 | 89.29 | 99.53 | 54.19 | 76.27 | 59.45 | 99.71 | 38.63 |
| | CCU | 43.04 | 90.95 | 99.90 | 48.34 | 100.00 | 0.75 | 100.00 | 0.48 |
| | ROWL | 93.35 | 53.02 | 100.00 | 49.69 | 100.00 | 49.69 | 100.00 | 49.69 |
| | NTOM (ours) | 36.94 | 92.61 | 98.17 | 65.70 | 99.97 | 0.76 | 100.00 | 0.16 |
| | ATOM (ours) | **32.30** | **93.06** | **93.15** | **71.96** | **38.72** | **88.03** | **93.44** | **69.15** |

Table 1: Comparison with competitive OOD detection methods. We use DenseNet as network architecture for all methods. We evaluate on four types of OOD inputs: (1) natural OOD, (2) corruption attacked OOD, (3) $L_\infty$ attacked OOD, and (4) compositionally attacked OOD inputs. The description of these OOD inputs can be found in Section 5.1. ↑ indicates larger value is better, and ↓ indicates lower value is better. All values are percentages and are averaged over six different OOD test datasets described in Section 5.1. **Bold** numbers are superior results. Additional results on a different architecture, WideResNet, are provided in Appendix B.9.

perturbations by $\Omega_{\infty,\epsilon}(\mathbf{x})$, where $\epsilon$ is the adversarial budget. We provide attack algorithms for all eight OOD detection methods in Appendix B.4.

- **Corruption attacked OOD (black-box)**: We consider a more realistic type of attack based on common corruptions (Hendrycks & Dietterich, 2019), which could appear naturally in the physical world. For each OOD image, we generate 75 corrupted images (15 corruption types × 5 severity levels), and then select the one with the lowest OOD score.
- **Compositionally attacked OOD (white-box)**: Lastly, we consider applying $L_\infty$-norm bounded attack and corruption attack jointly to an OOD input $\mathbf{x}$, as considered in (Laidlaw & Feizi, 2019).

**Evaluation Metrics.** We measure the following metrics: the false positive rate (FPR) at 5% false negative rate (FNR) and the area under the receiver operating characteristic curve (AUROC).

## 5.2 RESULTS

**How does ATOM compare to existing solutions?** We show in Table 1 that ATOM outperforms competitive OOD detection methods on both classic and adversarial OOD evaluation tasks. First, on classic OOD evaluation task (clean OOD data), ATOM achieves comparable or often even better performance than the current state-of-the-art methods. Second, on the existing adversarial OOD evaluation task – $L_\infty$ OOD, ATOM outperforms current state-of-the-art method ACET (Hein et al., 2019) by a large margin (e.g. on CIFAR-10, our method outperforms ACET by **53.9%** measured by FPR). Third, while ACET is somewhat brittle under the new Corruption OOD evaluation task, our method can generalize surprisingly well to the unknown corruption attacked OOD inputs, outperforming the best baseline by a large margin (e.g. on CIFAR-10, by up to **30.99%** measured by FPR). Finally, while almost every method fails under the hardest compositional OOD evaluation task, our method still achieves impressive results (e.g. on CIFAR-10, reduces the FPR by **57.99%**). The performance is noteworthy since our method is not trained explicitly on corrupted OOD inputs.

| $\mathcal{D}_{in}^{test}$ | Model | FPR (5% FNR) ↓ | AUROC ↑ | FPR (5% FNR) ↓ | AUROC ↑ | FPR (5% FNR) ↓ | AUROC ↑ | FPR (5% FNR) ↓ | AUROC ↑ |
|---|---|---|---|---|---|---|---|---|---|
| | | Natural OOD | | Corruption OOD | | $L_\infty$ OOD | | Comp. OOD | |
| SVHN | ATOM (rand. sample) | 0.35 | 99.91 | 13.09 | 97.50 | 11.72 | 96.63 | 98.66 | 40.48 |
| | ATOM (q=0.0) | 0.07 | 99.97 | 5.47 | 98.52 | 7.02 | 98.00 | 96.33 | 49.52 |
| | ATOM (q=0.125) | 1.30 | 99.63 | 34.97 | 94.97 | 39.61 | 82.92 | 99.92 | 6.30 |
| | ATOM (q=0.25) | 1.36 | 99.60 | 41.98 | 94.30 | 52.39 | 71.34 | 99.97 | 1.35 |
| | ATOM (q=0.5) | 2.11 | 99.46 | 44.85 | 93.84 | 59.72 | 65.59 | 99.97 | 3.15 |
| | ATOM (q=0.75) | 2.91 | 99.26 | 51.33 | 93.07 | 66.20 | 57.16 | 99.96 | 2.04 |
| CIFAR-10 | ATOM (rand. sample) | 2.65 | 99.11 | 42.28 | 91.94 | 44.31 | 68.64 | 65.17 | 72.62 |
| | ATOM (q=0.0) | 2.24 | 99.20 | 40.46 | 92.86 | 36.80 | 73.11 | 66.15 | 73.93 |
| | ATOM (q=0.125) | 1.69 | 99.20 | 25.26 | 95.29 | 20.55 | 88.94 | 38.89 | 86.71 |
| | ATOM (q=0.25) | 2.34 | 99.12 | 22.71 | 95.29 | 24.93 | 94.83 | 41.58 | 91.56 |
| | ATOM (q=0.5) | 4.03 | 98.97 | 33.93 | 93.51 | 22.39 | 95.16 | 45.11 | 90.56 |
| | ATOM (q=0.75) | 5.35 | 98.77 | 41.02 | 92.78 | 21.87 | 93.37 | 43.64 | 91.98 |
| CIFAR-100 | ATOM (rand. sample) | 51.50 | 89.62 | 99.70 | 58.61 | 70.33 | 58.84 | 99.80 | 34.98 |
| | ATOM (q=0.0) | 44.38 | 91.92 | 99.76 | 60.12 | 68.32 | 65.75 | 99.80 | 49.85 |
| | ATOM (q=0.125) | 26.91 | 94.97 | 98.35 | 71.53 | 34.66 | 87.54 | 98.42 | 68.52 |
| | ATOM (q=0.25) | 32.43 | 93.93 | 97.71 | 72.61 | 40.37 | 82.68 | 97.87 | 65.19 |
| | ATOM (q=0.5) | 32.30 | 93.06 | 93.15 | 71.96 | 38.72 | 88.03 | 93.44 | 69.15 |
| | ATOM (q=0.75) | 38.56 | 91.20 | 97.59 | 58.53 | 62.66 | 78.70 | 97.97 | 54.89 |

Table 2: Ablation study on informative outlier mining. We use DenseNet as network architecture. ↑ indicates larger value is better, and ↓ indicates lower value is better. All values are percentages and are averaged over six natural OOD test datasets mentioned in section 5.1. We **do not** use OOD test set for tuning $q$. Please refer to Table 5 for validation results.

Our training method leads to improved OOD detection while preserving classification performance on in-distribution data. Consistent performance improvement is observed on other in-distribution datasets (SVHN and CIFAR-100), alternative network architecture (WideResNet), and with alternative auxiliary dataset (ImageNet-RC).

**How does ATOM compare to NTOM?** We perform an ablation study that isolates the effect of adversarial training. In Table 1, we show that NTOM achieves comparable performance as ATOM on natural OOD and corruption OOD. However, NTOM is less robust under $L_\infty$ OOD (with 79.35% reduction in FPR on CIFAR-10) and compositional OOD inputs. This underlies the importance of having both adversarial training and outlier mining (ATOM) for overall good performance.

**How does the sampling parameter affect performance?** Table 2 shows the performance with different sampling parameter $q$. For all three datasets, training on OOD inputs primarily with large OOD scores (*i.e.*, too easy examples with $q = 0.75$) worsens the performance, which suggests the necessity to include examples on which the OOD detector is uncertain about. We also show that using informative outlier mining overall works better than random sampling under properly chosen $q$. Interestingly, in setting where the in-distribution data and auxiliary OOD data are disjoint (*e.g.*, SVHN/TinyImages), $q = 0$ is optimal, which suggests that the hardest outliers are mostly useful for training. However, in a more realistic setting, the auxiliary OOD data can almost always contain data similar to in-distribution data (*e.g.*, CIFAR/TinyImages). Even without removing near-duplicates exhaustively, ATOM can adaptively avoid training on those near-duplicates of in-distribution data (e.g. using $q = 0.125$ for CIFAR-10 and $q = 0.5$ for CIFAR-100).

**How does the choice of auxiliary OOD dataset affect the performance?** To see this, we additionally experiment with using ImageNet-RC as auxiliary OOD data. We observe consistent improvement of ATOM, and in many cases with performance better than using TinyImages. For example, on CIFAR-100, the FPR under natural OOD inputs is reduced from 32.20% (w/ TinyImages) to 15.49% (w/ ImageNet-RC). Interestingly, in all three datasets, using $q = 0$ (hardest outliers) yields the optimal performance since there is substantially less near-duplicates between ImageNet-RC and in-distribution data. This ablation suggests that ATOM's success does not depend on a particular auxiliary dataset. Full results are provided in Table 6 (Appendix B.8).

## 6 RELATED WORK

**Robustness of OOD Detection.** Worst-case aspects of OOD detection have previously been studied in (Hein et al., 2019; Sehwag et al., 2019). However, these papers are primarily concerned with $L_\infty$ norm bounded adversarial attacks, while our evaluation also includes common image corruption attacks. Besides, Meinke & Hein; Hein et al. only evaluate adversarial robustness of OOD detection on random noise images, while we also evaluate it on natural OOD images. Hein et al. (2019) has

theoretically analyzed why ReLU networks can yield high-confidence but wrong predictions for OOD data. Meinke & Hein has shown the first provable guarantees for worst-case OOD detection on some balls around uniform noise, and Bitterwolf et al. recently studied the provable guarantees for worst-case OOD detection not only for noise but also for images from related but different image classification tasks. In this paper, we propose ATOM which achieves state-of-the-art performance on a broader family of clean and perturbed OOD inputs. The key difference of our method compared to prior work is introducing the informative outlier mining technique, which could significantly improve the generalization and robustness of OOD detection.

**Discriminative Based Out-of-Distribution Detection.** Hendrycks & Gimpel introduced a baseline approach for OOD detection using the maximum softmax probability from a pre-trained network. Several works attempt to improve the OOD uncertainty estimation by using deep ensembles (Lakshminarayanan et al., 2017), the calibrated softmax score (Liang et al., 2018), and the Mahalanobis distance-based confidence score (Lee et al., 2018). Some methods also modify the neural networks by re-training or fine-tuning on some auxiliary anomalous data that are either realistic (Hendrycks et al., 2018; Mohseni et al., 2020; Papadopoulos et al., 2019) or artificially generated by GANs (Lee et al., 2017). Many other works (Subramanya et al., 2017; Malinin & Gales, 2018; Bevandić et al., 2018) also regularize the model to have lower confidence for anomalous examples.

**Generative Modeling Based Out-of-distribution Detection.** Generative models (Dinh et al., 2016; Kingma & Welling, 2013; Rezende et al., 2014; Van den Oord et al., 2016; Tabak & Turner, 2013) can be alternative approaches for detecting OOD examples, as they directly estimate the in-distribution density and can declare a test sample to be out-of-distribution if it lies in the low-density regions. However, as shown by Nalisnick et al., deep generative models can assign a high likelihood to out-of-distribution data. Deep generative models can be more effective for out-of-distribution detection using alternative metrics (Choi & Jang, 2018), likelihood ratio (Ren et al., 2019; Serrà et al., 2019), and modified training technique (Hendrycks et al., 2018). Recently, Pope et al. shows that flow-based generative models are sensitive under adversarial attacks. Note that we mainly considered discriminative-based approaches, which can be more competitive due to the availability of label information (and, in some cases, auxiliary OOD data (Hein et al., 2019; Hendrycks et al., 2018; Meinke & Hein, 2019; Mohseni et al., 2020)).

**Adversarial Robustness.** Adversarial examples (Goodfellow et al., 2014; Papernot et al., 2016; Biggio et al., 2013; Szegedy et al., 2013) have received considerable attention in recent years. Many defense methods have been proposed to mitigate this problem. One of the most effective methods is adversarial training (Madry et al., 2017), which uses robust optimization techniques to render deep learning models resistant to adversarial attacks. Carmon et al.; Najafi et al.; Zhai et al.; Uesato et al. show that unlabeled data can improve adversarial robustness for classification. In particular, the analysis in Carmon et al. (2019) has inspired our analysis in the Gaussian data model. But these studies are for robust classification, while our work focuses on roubst OOD detection where the key challenge is that the auxiliary OOD and test OOD distributions can differ.

**Hard Example Mining.** Hard example mining was introduced in the work (Sung, 1996) for training face detection models, where they gradually grow the set of background examples by selecting those examples for which the detector triggers a false alarm. The idea has been used extensively for object detection literature (Felzenszwalb et al., 2009; Gidaris & Komodakis, 2015; Shrivastava et al., 2016). It also has been used extensively in deep metric learning (Cui et al., 2016; Simo-Serra et al., 2015; Wang & Gupta, 2015; Suh et al., 2019) and deep embedding learning (Yuan et al., 2017; Smirnov et al., 2018; Wu et al., 2017; Duan et al., 2019). To the best of our knowledge, we are the first to explore hard example mining for out-of-distribution detection.

## 7 CONCLUSION

In this paper, we propose Adversarial Training with informative Outlier Mining (ATOM), a method that enhances the robustness of the OOD detector. We show the merit of adaptively selecting the OOD training examples which the OOD detector is uncertain about. Extensive experiments show ATOM can significantly improve the decision boundary of the OOD detector, achieving state-of-the-art performance under a broad family of *clean and perturbed* OOD evaluation tasks. We also provide theoretical analysis that justifies the benefits of outlier mining. Further, our unified evaluation framework allows future research to examine the robustness of the OOD detector. We hope our research can raise more attention to a broader view of robustness in out-of-distribution detection.

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

# Supplementary Material

## Informative Outlier Matters: Robustifying Out-of-distribution Detection Using Outlier Mining

## A THEORETICAL ANALYSIS

In this section, we provide theoretical analysis to the following two questions: 1) Why are auxiliary OOD data useful for training, even when they come from a different distribution than the test OOD distribution? 2) For practical auxiliary OOD data which contain non-informative samples, how can we make use of the auxiliary data to significantly improve the detection performance?

For the first question, we provide an error bound to justify the benefit of auxiliary OOD data. We notice that a key difference of OOD detection from typical classification is that the test OOD distribution is not accessible for training, so one need to use the auxiliary OOD data from a different distribution. This makes OOD detection more challenging. Our intuition is that even if the auxiliary data are different from test OOD data, they can still calibrate detectors in quite general situations, then the detector can generalize to the test OOD data. To formalize this, we adopt the domain adaption framework for our analysis.

For the second question, we provide analysis in a generative model of the data and motivate the importance of careful selection of informative auxiliary OOD data (i.e., informative outlier mining). Intuitively, since the auxiliary OOD data are different from the test OOD data, they may not be all useful. However, it is unclear why outlier mining can lead to significant improvements (see our experimental results in Section 5) and how to formalize this. Our intuition is that some of the auxiliary OOD data can be non-informative or even harmful, and they can overwhelm the benefit of informative outliers, leading to drastic drop in detection performance. To formalize it, one needs distributional assumptions on the data. We thus use a Gaussian data model and derive concrete bounds to illustrate our intuition.

### A.1 GENERALIZATION FROM AUXILIARY OOD DATA TO TEST OOD DATA

To see why detectors trained on the auxiliary OOD data $U_{\mathbf{X}}$ can generalize to the test OOD distribution $Q_{\mathbf{X}}$, we adopt the domain adaption framework (Ben-David et al., 2010). Recall that in domain adaptation there are two domains $s, t$, each being a distribution over the input space $\mathcal{X}$ and label space $\{-1, 1\}$. A classifier is trained on $s$ then applied on $t$. At a high level, we view our OOD detection problem as classification, where the source domain $s$ is $P_{\mathbf{X}}$ with labels 1 and $U_{\mathbf{X}}$ with labels $-1$, and the target domain $t$ is $P_{\mathbf{X}}$ with labels 1 and $Q_{\mathbf{X}}$ with label $-1$.

We focus on the FPR metric below; the argument for FNR is similar. Suppose we learn the OOD detector from a hypothesis class $\mathcal{G}$. Following Ben-David et al. (2010), we define (a variant) of the divergence of $Q_{\mathbf{X}}$ and $U_{\mathbf{X}}$ w.r.t. the hypothesis class $\mathcal{G}$ as

$$d_{\mathcal{G}}(Q_{\mathbf{X}}, U_{\mathbf{X}}) = \sup_{G, G' \in \mathcal{G}} v(G, G'; Q_{\mathbf{X}}) - v(G, G'; U_{\mathbf{X}})$$

where

$$v(G, G'; D) = \mathrm{FPR}(G; D, \Omega) - \mathrm{FPR}(G'; D, \Omega)$$

is the error difference of $G$ and $G'$ on the distribution $D$.

The divergence upper bounds the change of the hypothesis error difference between $Q_{\mathbf{X}}$ and $U_{\mathbf{X}}$. If it is small, then for any $G, G' \in \mathcal{G}$ where $G$ has a smaller error than $G'$ in $U_{\mathbf{X}}$, we know that $G$ will also have a smaller (or not too larger) error than $G'$ in $Q_{\mathbf{X}}$. That is, if the divergence is small, then the ranking of the hypotheses w.r.t. the error is roughly the same in both distributions. This *rank-preserving* property thus makes sure that a good hypothesis learned in $U_{\mathbf{X}}$ will also be good for $Q_{\mathbf{X}}$.

Now we show that, if $d_{\mathcal{G}}(Q_{\mathbf{X}}, U_{\mathbf{X}})$ is small (i.e., $Q_{\mathbf{X}}$ and $U_{\mathbf{X}}$ are aligned w.r.t. the class $\mathcal{G}$), then a detector $G$ with small FPR on $U_{\mathbf{X}}$ will also have small FPR on $Q_{\mathbf{X}}$.

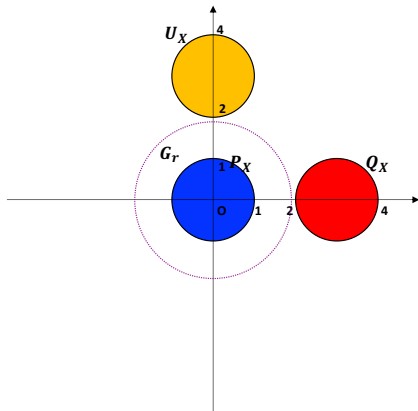

Figure 3: An illustration example to explain why $U_\mathbf{X}$ helps to get a good detector $G_r$. With $U_\mathbf{X}$, we can prune away hypotheses $G_r$ for any $r \geq 1.9$. Thus, the resulting detector $G_r$ can detect OOD samples from $Q_\mathbf{X}$ successfully and robustly.

**Proposition 2.** *For any $G \in \mathcal{G}$,*

$$\mathrm{FPR}(G; Q_\mathbf{X}, \Omega) \leq \inf_{G^* \in \mathcal{G}} \mathrm{FPR}(G^*; Q_\mathbf{X}, \Omega) + \mathrm{FPR}(G; U_\mathbf{X}, \Omega) + d_\mathcal{G}(Q_\mathbf{X}, U_\mathbf{X}).$$

*Proof.* For simplicity, we omit $\Omega$ from $\mathrm{FPR}(G; Q_\mathbf{X}, \Omega)$. For any $G^* \in \mathcal{G}$, we have

$$\mathrm{FPR}(G; Q_\mathbf{X}) = \mathrm{FPR}(G^*; Q_\mathbf{X}) + \mathrm{FPR}(G; Q_\mathbf{X}) - \mathrm{FPR}(G^*; Q_\mathbf{X}) \tag{8}$$
$$= \mathrm{FPR}(G^*; Q_\mathbf{X}) + \mathrm{FPR}(G; U_\mathbf{X}) - \mathrm{FPR}(G^*; U_\mathbf{X}) \tag{9}$$
$$+ [(\mathrm{FPR}(G; Q_\mathbf{X}) - \mathrm{FPR}(G^*; Q_\mathbf{X})) - (\mathrm{FPR}(G; U_\mathbf{X}) - \mathrm{FPR}(G^*; U_\mathbf{X}))]. \tag{10}$$

The last term is

$$(\mathrm{FPR}(G; Q_\mathbf{X}) - \mathrm{FPR}(G^*; Q_\mathbf{X})) - (\mathrm{FPR}(G; U_\mathbf{X}) - \mathrm{FPR}(G^*; U_\mathbf{X})) \tag{11}$$
$$= v(G, G^*; Q_\mathbf{X}) - v(G, G^*; U_\mathbf{X}) \tag{12}$$
$$\leq d_\mathcal{G}(Q_\mathbf{X}, U_\mathbf{X}). \tag{13}$$

Therefore,

$$\mathrm{FPR}(G; Q_\mathbf{X}) \leq \mathrm{FPR}(G^*; Q_\mathbf{X}) + \mathrm{FPR}(G; U_\mathbf{X}) + d_\mathcal{G}(Q_\mathbf{X}, U_\mathbf{X}). \tag{14}$$

Taking $\inf$ over $G^* \in \mathcal{G}$ completes the proof. □

The error of the detector is bounded by three terms: the best error, the error on the training distributions, and the divergence between $Q_\mathbf{X}$ and $U_\mathbf{X}$. Assuming that there exists a ground-truth detector with a small test error, and that the optimization can lead to a small training error, the test error is then characterized by the divergence. So in this case, as long as the rankings of the hypotheses (according to the error) on $Q_\mathbf{X}$ and $U_\mathbf{X}$ are similar, detectors learned on $U_\mathbf{X}$ can generalize to $Q_\mathbf{X}$.

**An illustration example.** In this example, the in-distribution $P_\mathbf{X}$ is uniform over the disk around the origin in $\mathbb{R}^2$ with radius 1, $U_\mathbf{X}$ is uniform over the disk around $(0, 3)$ with radius 1, and $Q_\mathbf{X}$ is uniform over the disk around $(3, 0)$ with radius 1. Assume the adversary budget is $\epsilon = 0.1$, i.e., $\Omega_{\infty, \epsilon} = \{\|\delta\|_\infty \leq 0.1\}$. The hypothesis class for the detector contains all functions of the form $G_r(x) = 2\mathbb{I}[\|x\|_2 \leq r] - 1$ with parameter $r$. See Figure 3.

The example first shows the effect of the auxiliary OOD data: $U_\mathbf{X}$ helps prune away hypotheses $G_r$ for any $r \geq 1.9$. Furthermore, it also shows how learning over $U_\mathbf{X}$ can generalize to $Q_\mathbf{X}$. Although $Q_\mathbf{X}$ and $U_\mathbf{X}$ have non-overlapping supports, $U_\mathbf{X}$ helps to calibrate the error of the hypotheses, so any good detector trained on $P_\mathbf{X}$ and $U_\mathbf{X}$ can be used for distinguishing $P_\mathbf{X}$ and $Q_\mathbf{X}$. Formally, the $d_\mathcal{G}$ is small in Proposition 2.

The analysis also shows the importance of training on *perturbed* instances from the auxiliary OOD data $U_{\mathbf{X}}$. Not using perturbation is equivalent to using $\Omega = \{\mathbf{0}\}$. In this case, the analysis shows that it only guarantees the error on *unperturbed* instances from $Q_{\mathbf{X}}$, even if $Q_{\mathbf{X}}$ and $U_{\mathbf{X}}$ has small divergence and the learned detector can have small training error on $U_{\mathbf{X}}$.

## A.2 IMPORTANCE OF OUTLIER MINING: ANALYSIS IN A GAUSSIAN DATA MODEL

To understand how the outlier training data affect the generalization, we study a concrete distributional model, which is inspired by the models in (Schmidt et al., 2018; Carmon et al., 2019). In this model, we establish in Section A.2.2 a separation of the in-distribution sample sizes needed in the two cases: with and without auxiliary OOD data for training. We also demonstrate in Section A.2.3 the benefit of outlier mining when the auxiliary OOD data consists of uninformative outliers.

While the theoretical model is simple (in fact, much simpler than the practical data distributions), its simplicity is actually desired for our analytical purpose. More precisely, the separation of the sample sizes under this simple model suggests the same phenomenon can happen in more complicated models. This then means the auxiliary OOD data not only help training but are *necessary* for obtaining detectors with reasonable performance when in-distribution data is limited.

**Gaussian Model.** To specify a distributional model for our robust OOD formulation, we need in-distribution $P_{\mathbf{X}}$, family of OOD distributions $\mathcal{Q}$, and the hypothesis class $\mathcal{H}$ for the OOD detector $G$. When auxiliary OOD data is available, we also need to specify their distribution $U_{\mathbf{X}}$. Let $\mu \in \mathbb{R}^d$ be the mean vector, $\sigma > 0$ be the variance parameter, and $\nu > 0$ be a parameter. In our $(\mu, \sigma, \nu)$-Gaussian model:

- $P_{\mathbf{X}}$ is $\mathcal{N}(\mu, \sigma^2 I)$.
- $\mathcal{Q} = \{\mathcal{N}(-\mu + v, \sigma^2 I) : v \in \mathbb{R}^d, \|v\|_2 \leq \nu\}$.
- $\mathcal{H} = \{G_\theta(x) = \text{sign}(\theta^\top \mathbf{x}) : \theta \in \mathbb{R}^d\}$. Here $G_\theta(\mathbf{x}) = 1$ means it predicts $x$ to be an in-distribution example, and $G_\theta(\mathbf{x}) = -1$ means it predicts an OOD example.

We are interested in the False Negative Rate $\text{FNR}(G)$ and worst False Positive Rate $\sup_{Q_{\mathbf{X}} \in \mathcal{Q}} \text{FPR}(G; Q_{\mathbf{X}}, \Omega_{\infty, \epsilon}(\mathbf{x}))$ over $Q_{\mathbf{X}} \in \mathcal{Q}$ under $\ell_\infty$ perturbations of magnitude $\epsilon$. For simplicity, we denote them as $\text{FNR}(G)$ and $\text{FPR}(G; Q_{\mathbf{X}})$ in our proofs.

**Parameter Setting.** The model parameters are set such that:

1. There exists a classifier that achieves very low errors FPR and FNR.

2. We need $n$ in-distribution data from $P_{\mathbf{X}}$ to learn a classifier with non-trivial robust errors.

3. Using $n_0$ in-distribution examples from $P_{\mathbf{X}}$ and $n'$ auxiliary OOD data from $U_{\mathbf{X}}$ where $n_0$ is much smaller than $n$, we can learn a classifier with non-trivial robust errors.

Here $n_0, n, n'$ are sample sizes whose values are specified later in our analysis.

The family of instances of the Gaussian data model used for our analysis is as follows. First, fix an integer $n_0 > 0$, and an $\epsilon \in (0, 1/2)$, then set the following parameter values:

$$d \gg n_0/\epsilon^4 + n_0 \log^2 d, \quad \|\mu\|_2^2 \in \left(\frac{9d}{10}, \frac{11d}{10}\right), \quad \sigma^2 = \sqrt{dn_0}, \quad \nu \leq \|\mu\|_2/4. \quad (15)$$

To interpret the parameter setting, one can view $n_0, \epsilon$ as fixed and $d/n_0$ as a large number.

### A.2.1 EXISTENCE OF ROBUST CLASSIFIERS

We give closed forms of the errors, and show that using $\theta = \mu$ gives small errors under the chosen setting in equation 15. The calculation largely follows that in (Carmon et al., 2019) with some slight modification.

**Closed Forms of the Errors.** By definition, the FNR of a detector $G_\theta$ (on $P_\mathbf{X}$) is:

$$\text{FNR}(G_\theta) = \mathbb{P}_{\mathbf{x} \sim P_\mathbf{X}}[\theta^\top \mathbf{x} \leq 0] = \mathbb{P}_{\mathbf{x} \sim P_\mathbf{X}}\left[\mathcal{N}\left(\frac{\mu^\top \theta}{\sigma\|\theta\|_2}, 1\right) \leq 0\right] =: \Phi\left(\frac{\mu^\top \theta}{\sigma\|\theta\|_2}\right) \quad (16)$$

where

$$\Phi(\mathbf{x}) := \frac{1}{\sqrt{2\pi}} \int_\mathbf{x}^\infty e^{-t^2/2} dt \quad (17)$$

is the Gaussian error function.

Given a test OOD distribution $Q_v = \mathcal{N}(-\mu + v, \sigma^2 I)$, the robust FPR of $G_\theta$ on $Q_v$ is:

$$\text{FPR}(G_\theta; Q_v) = \mathbb{P}_{\mathbf{x} \sim Q_v}\left[\inf_{\|\delta\|_\infty \leq \epsilon} \theta^\top(\mathbf{x} + \delta) \geq 0\right] \quad (18)$$

$$= \mathbb{P}_{\mathbf{x} \sim Q_v}\left[\theta^\top \mathbf{x} + \epsilon\|\theta\|_1 \geq 0\right] \quad (19)$$

$$= \mathbb{P}_{\mathbf{x} \sim Q_v}\left[\mathcal{N}((\mu + v)^\top \theta, (\sigma\|\theta\|_2)^2) \geq -\epsilon\|\theta\|_1\right] \quad (20)$$

$$= \Phi\left(\frac{(\mu + v)^\top \theta}{\sigma\|\theta\|_2} - \frac{\epsilon\|\theta\|_1}{\sigma\|\theta\|_2}\right). \quad (21)$$

Then the worst robust FPR of $G_\theta$ on $\mathcal{Q}$ is:

$$\sup_{Q_v \in \mathcal{Q}} \text{FPR}(G_\theta; Q_v) = \sup_{\|v\|_2 \leq \nu} \Phi\left(\frac{(\mu + v)^\top \theta}{\sigma\|\theta\|_2} - \frac{\epsilon\|\theta\|_1}{\sigma\|\theta\|_2}\right) \quad (22)$$

$$= \Phi\left(\frac{\mu^\top \theta}{\sigma\|\theta\|_2} - \frac{\nu}{\sigma} - \frac{\epsilon\|\theta\|_1}{\sigma\|\theta\|_2}\right) \quad (23)$$

$$\leq \Phi\left(\frac{\mu^\top \theta}{\sigma\|\theta\|_2} - \frac{\nu}{\sigma} - \frac{\epsilon\sqrt{d}}{\sigma}\right). \quad (24)$$

**Small Errors of $G_\mu$.** Given the closed forms, we can now show that $G_\mu$ achieves small FNR and FPR in our parameter setting.

$$\text{FNR}(G_\mu) = \Phi\left(\frac{\|\mu\|_2}{\sigma}\right) \leq \Phi\left(\sqrt{\frac{11}{10}}\left(\frac{d}{n_0}\right)^{1/4}\right) \leq e^{-\frac{11}{20}\sqrt{d/n_0}}. \quad (25)$$

$$\sup_{Q_v \in \mathcal{Q}} \text{FPR}(G_\mu; Q_v) \leq \Phi\left(\frac{\|\mu\|_2}{\sigma} - \frac{\nu}{\sigma} - \frac{\epsilon\sqrt{d}}{\sigma}\right) \quad (26)$$

$$\leq \Phi\left(\left(\sqrt{\frac{11}{10}} - \frac{1}{4} - \epsilon\right)\left(\frac{d}{n_0}\right)^{1/4}\right) \leq e^{-\frac{1}{32}\sqrt{d/n_0}}. \quad (27)$$

Therefore, in the regime $d/n_0 \gg 1$, the detector $G_\mu$ achieves both small FNR on $P_\mathbf{X}$ and robust FPR on any test OOD distribution in $\mathcal{Q}$.

### A.2.2 BENEFIT OF AUXILIARY OOD DATA

We will first consider the case when auxiliary OOD data are not available, and give a lower bound (Proposition 3). We will then consider the case when auxiliary OOD data are used, and give an upper bound (Proposition 4). Comparing Proposition 3 and Proposition 4 will then justify the benefit of the auxiliary OOD data.

**Learning Without Auxiliary OOD Data.** Given in-distribution data $\mathbf{x}_1, \mathbf{x}_2, \ldots, \mathbf{x}_n$, we consider the detector $G_{\hat{\theta}_n}$ given by

$$\hat{\theta}_n = \frac{1}{n} \sum_{i=1}^n \mathbf{x}_i. \quad (28)$$

Next we show a lower bound of the in-distribution data needed for the case without auxiliary outliers. That is, a sample size of order $n_0 \cdot \frac{\epsilon^2 \sqrt{d/n_0}}{\log d}$ is necessary for *all algorithms* to obtain both non-trivial robust FPR and FNR. We emphasize that this lower bound is information theoretic, i.e., it holds without restriction on the computational power of the learning algorithm and the hypothesis class used for the OOD detector.

**Proposition 3.** *(Bound without auxiliary data) Consider the same family of instances as in Preposition 1, without any auxiliary OOD data. If $n \leq n_0 \cdot \frac{\epsilon^2 \sqrt{d/n_0}}{16 \log d}$, then for any algorithm $\mathbb{A}_n$, there exists an instance in the family with*

$$\mathbb{E}\left\{ \mathrm{FNR}(\mathbb{A}_n(S)) + \sup_{Q_\mathbf{X} \in \mathcal{Q}} \mathrm{FPR}(\mathbb{A}_n(S); Q_\mathbf{X}, \Omega_{\infty,\epsilon}(\mathbf{x})) \right\} \geq \frac{1}{4}(1 - d^{-1}). \tag{29}$$

*Proof.* The key for the proof is the observation that robust classification is a special case of our robust OOD problem. More precisely, consider the following robust classification problem. The data $(\mathbf{x}, y)$ with $\mathbf{x} \in \mathbb{R}^d$ and $y \in \{-1, +1\}$ is generated as follows: first draw $y$ uniformly at random, and then draw $\mathbf{x}$ from $\mathcal{N}(y \cdot \mu, \sigma^2 I)$. Given training data $\{(\mathbf{x}_i, y_i)\}_{i=1}^n$, the goal is to find classifier $f_\theta(\mathbf{x}) = \mathrm{sign}(\theta^\top \mathbf{x})$ with small robust classification error

$$\mathrm{err}_{\infty,\epsilon}(f_\theta) = \mathbb{E}_{(\mathbf{x},y)} \max_{\|\delta\|_\infty \leq \epsilon} \mathbb{I}[f_\theta(\mathbf{x} + \delta) \neq y]$$

under $\ell_\infty$ perturbation of magnitude $\epsilon$. It has been shown that (Theorem 6 in (Schmidt et al., 2018) or Theorem 1 in (Carmon et al., 2019)) that when $\mu \sim \mathcal{N}(0, I)$ and $n \leq n_0 \cdot \frac{\epsilon^2 \sqrt{d/n_0}}{8 \log d}$ and with the parameter setting equation 15, for any learning algorithm $\mathbb{A}_n$

$$\mathbb{E}\mathrm{err}_{\infty,\epsilon}(\mathbb{A}_n(S)) \geq \frac{1}{2}(1 - d^{-1}). \tag{30}$$

Now consider the following variant of the robust OOD problem in the proposition. Suppose $\mu \sim \mathcal{N}(0, I)$. Suppose besides the data from $P_\mathbf{X}$, we also have $n$ i.i.d. samples from a test OOD distribution $Q_0 = \mathcal{N}(-\mu, \sigma^2 I)$. Then the above robust classification problem can be reduced to this variant of robust OOD, by viewing the in-distribution data as with label $+1$ and viewing outliers as with label $-1$. Furthermore, it is clear that the sum of the FNR and FPR is larger than the robust classification error. Then

$$\mathbb{E}\left\{ \mathrm{FNR}(\mathbb{A}_n(S)) + \mathrm{FPR}(\mathbb{A}_n(S); Q_0) \right\} \geq \frac{1}{2}(1 - d^{-1}). \tag{31}$$

When $d$ is sufficiently large, we have $\mu$ satisfies the condition in equation 15 with probability at least $9/10$. Then

$$\mathbb{E}\left\{ \mathrm{FNR}(\mathbb{A}_n(S)) + \mathrm{FPR}(\mathbb{A}_n(S); Q_0) \mid \|\mu\|_2^2 \in (9d/10, 11d/10) \right\} \geq \frac{1}{4}(1 - d^{-1}). \tag{32}$$

After conditioning, this variant can be reduced to the original robust OOD problem in the proposition and furthermore $Q_0 \in \mathcal{Q}$. Furthermore, the fact that the expectation over the conditional distribution of $\mu$ is large implies that there exist an instance of $\mu$ with a large error. The statement then follows. $\square$

**Learning With Auxiliary OOD Data.** Assuming we have access to auxiliary OOD data from a distribution $U_\mathbf{X}$ where:

- $U_\mathbf{X}$ is defined by the following distribution: first draw $v$ uniformly at random from the ball $\{v : v \in \mathbb{R}^d, \|v\|_2 \leq \nu\}$, then draw $\tilde{\mathbf{x}}$ from $\mathcal{N}(-\mu + v, \sigma^2 I)$.

Roughly speaking, $U_\mathbf{X}$ is a uniform mixture of distributions in $\mathcal{Q}$.

Given in-distribution data $\mathbf{x}_1, \mathbf{x}_2, \ldots, \mathbf{x}_n$ from $P_\mathbf{X}$ and auxiliary OOD data $\tilde{\mathbf{x}}_1, \tilde{\mathbf{x}}_2, \ldots, \tilde{\mathbf{x}}_{n'}$ from $U_\mathbf{X}$, we consider the detector $G_{\hat{\theta}_{n,n'}}$ given by

$$\hat{\theta}_{n,n'} = \frac{1}{n}\sum_{i=1}^n \mathbf{x}_i - \frac{1}{n'}\sum_{i=1}^{n'} \tilde{\mathbf{x}}_i. \tag{33}$$

We will show that with $n = n_0$ and sufficiently large $n'$, the detector has small errors.

Again, as shown in the closed form solutions, the key factor determining the errors is $\frac{\mu^\top \hat{\theta}_{n,n'}}{\sigma \|\hat{\theta}_{n,n'}\|_2}$. The following lemma bounds this term.

**Lemma 1.** *There exist numerical constants $c_0, c_1, c_2$ such that under parameter setting equation 15 and $d/n_0 > c_0$,*

$$\frac{\mu^\top \hat{\theta}_{n,n'}}{\sigma \|\hat{\theta}_{n,n'}\|_2} \geq \frac{9}{10} \left( \sqrt{\frac{n_0}{d}} + \frac{n_0}{n+n'} \left( 1 + c_1 \left( \frac{n_0}{d} \right)^{1/8} \right) \right)^{-1/2} \tag{34}$$

*with probability $\geq 1 - e^{-c_2 (d/n_0)^{1/4} \min\{n+n', (d/n_0)^{1/4}\}} - e^{-c_2 n'}$.*

*Proof.* The proof follows the argument of Lemma 1 in (Carmon et al., 2019) but needs some modifications accommodating the difference in learning $\theta$. Recall the generation of $\mathbf{x}'_i$: first draw $v_i$ uniformly at random from the ball $\mathbb{B}(\nu) := \{v : v \in \mathbb{R}^d, \|v\|_2 \leq \nu\}$, then draw $\bar{\mathbf{x}}'_i$ from $\mathcal{N}(\mu, \sigma^2 I)$, and finally let $\mathbf{x}'_i = v_i - \bar{\mathbf{x}}'_i$. So we have

$$\hat{\theta}_{n,n'} = \frac{1}{n+n'} \left( \sum_{i=1}^{n} \mathbf{x}_i + \sum_{i=1}^{n'} \bar{\mathbf{x}}'_i \right) - \frac{1}{n+n'} \left( \sum_{i=1}^{n'} v_i \right) \tag{35}$$

$$= \mu + \delta + \delta_v \tag{36}$$

where

$$\delta = \frac{1}{n+n'} \left( \sum_{i=1}^{n} \mathbf{x}_i + \sum_{i=1}^{n'} \bar{\mathbf{x}}'_i \right) - \mu \sim \mathcal{N}(0, \frac{\sigma^2}{n+n'} I), \tag{37}$$

$$\delta_v = -\frac{1}{n+n'} \left( \sum_{i=1}^{n'} v_i \right). \tag{38}$$

To lower bound the term $\frac{\mu^\top \hat{\theta}_{n,n'}}{\|\hat{\theta}_{n,n'}\|_2}$, we upper bound its squared inverse:

$$\frac{\|\hat{\theta}_{n,n'}\|_2^2}{(\mu^\top \hat{\theta}_{n,n'})^2} = \frac{\|\mu + \delta + \delta_v\|_2^2}{(\|\mu\|_2^2 + \mu^\top \delta + \mu^\top \delta_v)^2} \tag{39}$$

$$= \frac{1}{\|\mu\|_2^2} + \frac{\|\delta + \delta_v\|_2^2 - \frac{1}{\|\mu\|_2^2}(\mu^\top \delta + \mu^\top \delta_v)^2}{(\|\mu\|_2^2 + \mu^\top \delta + \mu^\top \delta_v)^2} \tag{40}$$

$$\leq \frac{1}{\|\mu\|_2^2} + \frac{2\|\delta\|_2^2 + 2\|\delta_v\|_2^2}{(\|\mu\|_2^2 + \mu^\top \delta + \mu^\top \delta_v)^2}. \tag{41}$$

For $\delta$, we have

$$\|\delta\|_2^2 \sim \frac{\sigma^2}{n+n'} \chi_d^2 \quad \text{and} \quad \frac{\mu^\top \delta}{\|\mu\|_2} \sim \mathcal{N}\left( 0, \frac{\sigma^2}{n+n'} \right). \tag{42}$$

So standard concentration bounds give

$$\mathbb{P}\left( \|\delta\|_2^2 \geq \frac{\sigma^2}{n+n'} \left( d + \frac{1}{\sigma} \right) \right) \leq e^{-d/8\sigma^2} \quad \text{and} \quad \mathbb{P}\left( \frac{\mu^\top \delta}{\|\mu\|_2} \geq (\sigma\|\mu\|)^{1/2} \right) \leq 2e^{-(n+n')\|\mu\|_2/2\sigma}. \tag{43}$$

For $\delta_v$, by subguassian concentration bounds, we have

$$\mathbb{P}\left( \|\delta_v\|_2 \geq \frac{C\nu}{\sqrt{n'}} \right) \leq e^{-cn'} \tag{44}$$

for some numeric constants $c$ and $C$. Suppose the event $\|\delta_v\|_2 < \frac{C\nu}{\sqrt{n'}}$ is true. Then

$$|\mu^\top \delta_v| \leq \|\mu\|_2 \|\delta_v\|_2 \leq \frac{C\nu\|\mu\|_2}{\sqrt{n'}}. \tag{45}$$

Plugging the concentration bounds in equation 39 and doing the same manipulation leads to the bound. To finish the proof, we also need to show $\mu^\top \hat{\theta}_{n,n'} > 0$, which can be shown by a similar argument as above. $\square$

We then get the following guarantee. Again, the error bound $10^{-3}$ is chosen for simplicity of the statement, but it can be made to arbitrarily small values.

**Proposition 4.** *(Error bound with ideal auxiliary data) Consider the same family of instances as in Preposition 1, with $n'$ auxiliary OOD data from $U_{\mathbf{X}}$ specified above. If $n \geq n_0$ and $n' \geq n_0 \cdot 4\epsilon^2 \sqrt{d/n_0}$, then*

$$\mathbb{E}_{\hat{\theta}_{n,n'}} \mathrm{FNR}(G_{\hat{\theta}_{n,n'}}) \leq 10^{-3}, \quad \mathbb{E}_{\hat{\theta}_{n,n'}} \sup_{Q_{\mathbf{X}} \in \mathcal{Q}} \mathrm{FPR}(G_{\hat{\theta}_{n,n'}}; Q_{\mathbf{X}}, \Omega_{\infty,\epsilon}(\mathbf{x})) \leq 10^{-3}. \quad (46)$$

*Proof.* The proposition comes from Lemma 1, the parameter setting equation 15, and the closed form expressions equation 16 and equation 22 of the errors. $\square$

### A.2.3 BENEFIT OF OUTLIER MINING

The above Gaussian example shows the benefit of having auxiliary OOD data for training. All the auxiliary OOD data given in the example are implicitly related to the ideal parameter for the detector $\theta^* = \mu$ and thus are informative for learning the detector. However, this may not be the case in practice: typically only part of the auxiliary OOD data are informative, while the remaining are not very useful or even can be harmful for the learning. In this section, we study such an example, and shows that how outlier mining can help to identify informative data and improve the learning performance.

Suppose the algorithm gets $n$ in-distribution data $\{\mathbf{x}_1, \mathbf{x}_2, \ldots, \mathbf{x}_n\}$ i.i.d. from $P_{\mathbf{X}}$ and $n'$ auxiliary OOD data $\{\tilde{\mathbf{x}}_1, \tilde{\mathbf{x}}_2, \ldots, \tilde{\mathbf{x}}_{n'}\}$ for training. Instead of from $U_{\mathbf{X}}$ specified above, the auxiliary OOD data are i.i.d. from the distribution $U_{\mathrm{mix}}$.

- $U_{\mathrm{mix}}$ is a uniform mixture of $\mathcal{N}(-\mu, \sigma^2 I)$ and $\mathcal{N}(\mu_o, \sigma^2 I)$ for $\mu_o = 10\mu$.

That is, the distribution is defined by the following process: with probability $1/2$ sample the outlier from the informative part $\mathcal{N}(-\mu, \sigma^2 I)$, and with probability $1/2$ sample the outlier from the uninformative part $\mathcal{N}(\mu_o, \sigma^2 I)$. We also note that $\mu_0 = 10\mu$ is chosen for simplicity of analysis. $\mu_0$ can also be $c\mu$ for some sufficiently large $c > 1$, or even $\mu_o = c\mu + c'\mu_\perp$ for a sufficiently large $c > 1$, a small $c'$ and a unit vector $\mu_\perp$ perpendicular to $\mu$. Our analysis still go through with such assumptions.

**Naïve Method Without Outlier Mining.** It is clear that naïvely applying the method in the previous section can lead to high errors: with $n$ in-distribution examples from $P_{\mathbf{X}}$ and $n' = n$ auxiliary OOD data from $U_{\mathrm{mix}}$, when $n \to \infty$, we have $\hat{\theta}_{n,n'} \to -7\mu/4$ which has the worst errors among all detectors.

**With Outlier Mining.** Here we analyze the following algorithm using the outlier mining approach. The algorithm is simpler than what we used in Section 4 but shares the same intuition.

First, we use the $n$ in-distribution data points to get an intermediate solution:

$$\hat{\theta}_{\mathrm{int}} = \frac{1}{n} \sum_{i=1}^{n} \mathbf{x}_i. \quad (47)$$

We define the confidence score of a point $\tilde{\mathbf{x}}$ being in-distribution as:

$$f(\tilde{\mathbf{x}}) = \sigma(t) = \frac{1}{1 + e^{-t}}, \quad \text{where} \ \ t(\tilde{\mathbf{x}}) = \frac{\tilde{\mathbf{x}}^\top \hat{\theta}_{\mathrm{int}}}{d}. \quad (48)$$

Here $\sigma(t) = \frac{1}{1 + e^{-t}}$ is the sigmoid function. We then select outlier training data whose confidence fall into an interval $[a, b]$ and use them to learn the final solution:

$$\hat{\theta}_{\mathrm{om}} = \frac{\sum_{i=1}^{n'} (-\tilde{\mathbf{x}}_i) \mathbb{I}\{f(\tilde{\mathbf{x}}_i) \in [a, b]\}}{\sum_{i=1}^{n'} \mathbb{I}\{f(\tilde{\mathbf{x}}_i) \in [a, b]\}} \quad (49)$$

where $\mathbb{I}\{\cdot\}$ is the indicator function.

**Proposition 1. (Error bound with outlier mining.)** *For any $\epsilon \in (0, 1/2)$ and any integer $n_0 > 0$, there exist a family of instances of the Gaussian data model such that the following is true. $n'$ auxiliary OOD data from $U_{\mathrm{mix}}$ specified above. There exist thresholds $a$ and $b$ for $\hat{\theta}_{\mathrm{om}}$ and a universal constant $c > 0$ such that if the number of in-distribution data $n \geq c(n_0 \log d + \sqrt{dn_0})$ and the number of auxiliary data $n' \geq (d + n_0 \cdot 4\epsilon^2)\sqrt{d/n_0}$, then $\hat{\theta}_{\mathrm{om}}$ has small errors:*[4]

$$\mathbb{E}_{\hat{\theta}_{\mathrm{om}}}\mathrm{FNR}(G_{\hat{\theta}_{\mathrm{om}}}) \leq 10^{-3}, \quad \mathbb{E}_{\hat{\theta}_{\mathrm{om}}} \sup_{Q_{\mathbf{X}} \in \mathcal{Q}} \mathrm{FPR}(G_{\hat{\theta}_{\mathrm{om}}}; Q_{\mathbf{X}}, \Omega_{\infty,\epsilon}(\mathbf{x})) \leq 10^{-3}. \tag{3}$$

*Proof.* Let $a = \sigma(-3/2), b = \sigma(-1/2)$. By definition we have

$$\delta_{\mathrm{om}} := \hat{\theta}_{\mathrm{om}} - \mu = \frac{\sum_{i=1}^{n'}(-\mu - \tilde{\mathbf{x}}_i)\mathbb{I}\{f(\tilde{\mathbf{x}}_i) \in [a, b]\}}{\sum_{i=1}^{n'} \mathbb{I}\{f(\tilde{\mathbf{x}}_i) \in [a, b]\}}. \tag{50}$$

By the closed form expressions equation 16 and equation 22 of the errors, it is sufficient to lower bound the key term $\frac{\mu^\top \hat{\theta}_{\mathrm{om}}}{\|\hat{\theta}_{\mathrm{om}}\|_2}$, which comes down to show that $\delta_{\mathrm{om}}$ is small.

First, let's consider $\hat{\theta}_{\mathrm{int}}$. Let $\delta_{\mathrm{int}} := \hat{\theta}_{\mathrm{int}} - \mu$. Then

$$\|\delta_{\mathrm{int}}\|_2^2 \sim \frac{\sigma^2}{n}\chi_d^2 \quad \text{and} \quad \frac{\mu^\top \delta_{\mathrm{int}}}{\|\mu\|_2} \sim \mathcal{N}\left(0, \frac{\sigma^2}{n}\right). \tag{51}$$

So standard concentration bounds give

$$\mathbb{P}\left(\|\delta_{\mathrm{int}}\|_2^2 \geq \frac{\sigma^2}{n}\left(d + \frac{1}{\sigma}\right)\right) \leq e^{-d/8\sigma^2} \quad \text{and} \quad \mathbb{P}\left(\frac{|\mu^\top \delta_{\mathrm{int}}|}{\|\mu\|_2} \geq \sqrt{\frac{d}{n}}\right) \leq 2e^{-d/2\sigma^2}. \tag{52}$$

So with probability $\geq 1 - 3e^{-d/8\sigma^2}$ over the randomness of the $n$ in-distribution points, we have the good event $\mathcal{G}_{\mathrm{int}}$: $\|\delta_{\mathrm{int}}\|_2^2 \leq \frac{\sigma^2}{n}\left(d + \frac{1}{\sigma}\right)$ and $\frac{|\mu^\top \delta_{\mathrm{int}}|}{\|\mu\|_2} \leq \sqrt{\frac{d}{n}}$.

Now, condition on a fix $\hat{\theta}_{\mathrm{int}}$ satisfying $\mathcal{G}_{\mathrm{int}}$, and consider $\hat{\theta}_{\mathrm{om}}$. Define

$$z_i := -\mu - \tilde{\mathbf{x}}_i, \tag{53}$$
$$\mathbb{I}_{0i} := \mathbb{I}\{f(\tilde{\mathbf{x}}_i) \in [a, b]\}, \tag{54}$$
$$\mathbb{I}_{1i} := \mathbb{I}\{\tilde{\mathbf{x}}_i \text{ is from } \mathcal{N}(-\mu, \sigma^2 I)\}, \tag{55}$$
$$\mathbb{I}_{2i} := \mathbb{I}\{\tilde{\mathbf{x}}_i \text{ is from } \mathcal{N}(\mu_o, \sigma^2 I)\}. \tag{56}$$

For simplicity, let's omit the subscript $i$ and consider a sample $\tilde{\mathbf{x}}$ from $U_{\mathrm{mix}}$, and the corresponding variables $z, \mathbb{I}_0, \mathbb{I}_1,$ and $\mathbb{I}_2$. Since $\mathbb{I}_1 + \mathbb{I}_2 = 1$,

$$(-\mu - \tilde{\mathbf{x}})\mathbb{I}\{f(\tilde{\mathbf{x}}) \in [a, b]\} = z\mathbb{I}_0\mathbb{I}_1 + z\mathbb{I}_0\mathbb{I}_2. \tag{57}$$

**Case 1.** Let's first consider the case when $\tilde{\mathbf{x}}$ is from $\mathcal{N}(-\mu, \sigma^2 I)$. More precisely, we condition on a fixed $\hat{\theta}_{\mathrm{int}}$ and condition on $\mathbb{I}_1 = 1$. Then $z \sim \mathcal{N}(0, \sigma^2 I)$ and it can be decomposed along the direction $\bar{\theta}_{\mathrm{int}} := \hat{\theta}_{\mathrm{int}}/\|\hat{\theta}_{\mathrm{int}}\|_2$ as follows:

$$z = s \cdot \bar{\theta}_{\mathrm{int}} + z_2 \tag{58}$$

where $s \sim \mathcal{N}(0, \sigma^2)$ and $z_2$ is a Gaussian distribution in the subspace orthogonal to $\bar{\theta}_{\mathrm{int}}$. Then

$$t(\tilde{\mathbf{x}}) = \frac{\tilde{\mathbf{x}}^\top \hat{\theta}_{\mathrm{int}}}{d} = -\frac{\mu^\top \hat{\theta}_{\mathrm{int}}}{d} - \frac{s\|\hat{\theta}_{\mathrm{int}}\|_2}{d}. \tag{59}$$

Therefore, we have

$$\mathbb{E}[z\mathbb{I}_0\mathbb{I}_1 | \mathbb{I}_1 = 1, \hat{\theta}_{\mathrm{int}}] = \mathbb{E}[s \cdot \bar{\theta}_{\mathrm{int}}\mathbb{I}_0 | \mathbb{I}_1 = 1, \hat{\theta}_{\mathrm{int}}] + \mathbb{E}[z_2\mathbb{I}_0 | \mathbb{I}_1 = 1, \hat{\theta}_{\mathrm{int}}] \tag{60}$$

---

[4]The error bound in the proposition can be made arbitrarily small and with high probability. The current bound is presented for simplicity.

Clearly the second term is $0$ since $z_2 \mathbb{I}_0$ is symmetric. So

$$\mathbb{E}[z\mathbb{I}_0\mathbb{I}_1 | \mathbb{I}_1 = 1, \hat{\theta}_{\text{int}}] = \mathbb{E}[s \cdot \bar{\theta}_{\text{int}} \mathbb{I}\{f(\tilde{\mathbf{x}}) \in [a, b]\} | \mathbb{I}_1 = 1, \hat{\theta}_{\text{int}}] \tag{61}$$

$$= \mathbb{E}\left[s \cdot \mathbb{I}\{s \in [a', b']\} | \mathbb{I}_1 = 1, \hat{\theta}_{\text{int}}\right] \cdot \bar{\theta}_{\text{int}} \tag{62}$$

$$= \mathbb{E}\left[s \cdot \mathbb{I}\{s \in [a', b']\}\right] \cdot \bar{\theta}_{\text{int}} \tag{63}$$

where

$$a' = -\frac{\mu^\top \hat{\theta}_{\text{int}}}{\|\hat{\theta}_{\text{int}}\|_2} - \frac{\sigma^{-1}(b)d}{\|\hat{\theta}_{\text{int}}\|_2} \tag{64}$$

$$= \frac{-2\mu^\top \hat{\theta}_{\text{int}} + d}{2\|\hat{\theta}_{\text{int}}\|_2} \tag{65}$$

$$= \frac{-2\mu^\top \delta_{\text{int}} - 2\|\mu\|_2^2 + d}{2\|\hat{\theta}_{\text{int}}\|_2}, \tag{66}$$

$$b' = -\frac{\mu^\top \hat{\theta}_{\text{int}}}{\|\hat{\theta}_{\text{int}}\|_2} - \frac{\sigma^{-1}(a)d}{\|\hat{\theta}_{\text{int}}\|_2} \tag{67}$$

$$= \frac{-2\mu^\top \hat{\theta}_{\text{int}} + 3d}{2\|\hat{\theta}_{\text{int}}\|_2} \tag{68}$$

$$= \frac{-2\mu^\top \delta_{\text{int}} - 2\|\mu\|_2^2 + 3d}{2\|\hat{\theta}_{\text{int}}\|_2}. \tag{69}$$

By the bound on $|\mu^\top \delta_{\text{int}}|$, we have

$$|\mathbb{E}\left[s \cdot \mathbb{I}\{s \in [a', b']\}\right]| \leq \int_{\frac{d}{2\sigma\|\hat{\theta}_{\text{int}}\|_2}(8/10 - 4/\sqrt{n})}^{\frac{d}{2\sigma\|\hat{\theta}_{\text{int}}\|_2}(12/10 + 4/\sqrt{n})} \sigma t \frac{1}{\sqrt{2\pi}} e^{-t^2/2} dt \tag{70}$$

$$\leq \frac{d}{2\sigma\|\hat{\theta}_{\text{int}}\|_2} \cdot \sigma \cdot \frac{2d}{2\sigma\|\hat{\theta}_{\text{int}}\|_2} \cdot \frac{1}{\sqrt{2\pi}} e^{-\frac{1}{2}\left(\frac{d}{4\sigma\|\hat{\theta}_{\text{int}}\|_2}\right)^2}. \tag{71}$$

Given the bound on $\|\delta_{\text{int}}\|_2^2$, we have

$$\|\hat{\theta}_{\text{int}}\|_2 \leq \|\mu\|_2 + \|\delta_{\text{int}}\|_2 \leq \|\mu\|_2 + \sqrt{\frac{\sigma^2}{n}\left(d + \frac{1}{\sigma}\right)} \leq \|\mu\|_2 + \sqrt{\frac{2\sigma^2 d}{n}}, \tag{72}$$

$$\|\hat{\theta}_{\text{int}}\|_2 \geq \|\mu\|_2 - \|\delta_{\text{int}}\|_2 \geq \|\mu\|_2 - \sqrt{\frac{\sigma^2}{n}\left(d + \frac{1}{\sigma}\right)} \geq \|\mu\|_2 - \sqrt{\frac{2\sigma^2 d}{n}}. \tag{73}$$

Since $n \geq Cn_0 \log d$ and $d \geq C^2 n_0 \log^2 d$ for a sufficiently large $C$, we have

$$\frac{\sigma^2 \|\hat{\theta}_{\text{int}}\|_2^2}{d^2} \leq \frac{\sigma^2 d(11/10 + \sqrt{2\sigma^2/n})^2}{d^2} \leq 4\sqrt{\frac{n_0}{d}} + \frac{8n_0}{n} \leq \frac{12}{C \log d} \tag{74}$$

and thus

$$|\mathbb{E}\left[s \cdot \mathbb{I}\{s \in [a', b']\}\right]| \leq \frac{d^2}{\sigma\|\hat{\theta}_{\text{int}}\|_2^2} \frac{1}{\sqrt{2\pi}} e^{-\frac{1}{2}\left(\frac{d}{4\sigma\|\hat{\theta}_{\text{int}}\|_2}\right)^2} \tag{75}$$

$$\leq \frac{d^2}{\sigma\|\hat{\theta}_{\text{int}}\|_2^2} e^{-\frac{d^2}{32\sigma^2\|\hat{\theta}_{\text{int}}\|_2^2}} \tag{76}$$

$$\leq \frac{1}{d^2}. \tag{77}$$

Combining with $\mathbb{E}[z\mathbb{I}_0\mathbb{I}_1 | \mathbb{I}_1 = 0, \hat{\theta}_{\text{int}}] = 0$ we get

$$\mathbb{E}[z\mathbb{I}_0\mathbb{I}_1 | \hat{\theta}_{\text{int}}] = c_1 \cdot \bar{\theta}_{\text{int}} \tag{78}$$

for some $c_1$ satisfying $|c_1| \leq 1/d^2$. Furthermore, $z\mathbb{I}_0\mathbb{I}_1 \mid \hat{\theta}_{\text{int}}$ is truncated Gaussian and thus is sub-Gaussian with sub-Gaussian norm bounded by $\sigma$. Then by sub-Gaussian concentration bounds, we have

$$\mathbb{P}\left(\left|\sum_{i=1}^{n'} \mu^\top z_i \mathbb{I}_{0i}\mathbb{I}_{1i} - \sum_{i=1}^{n'} \mu^\top \mathbb{E}[z_i\mathbb{I}_{0i}\mathbb{I}_{1i}|\hat{\theta}_{\text{int}}]\right| \geq \sqrt{n'd} \mid \hat{\theta}_{\text{int}}\right) \leq e^{-cd/\sigma^2}, \tag{79}$$

$$\mathbb{P}\left(\left\|\sum_{i=1}^{n'} z_i \mathbb{I}_{0i}\mathbb{I}_{1i}\right\|_2 \geq 4\sigma\sqrt{n'd} + 2\sqrt{n'd} \mid \hat{\theta}_{\text{int}}\right) \leq e^{-d/\sigma^2}. \tag{80}$$

for some constant $c > 0$. In other words, with probability $\geq 1 - 2e^{-cd/\sigma^2}$, we have

$$\left|\sum_{i=1}^{n'} \mu^\top z_i \mathbb{I}_{0i}\mathbb{I}_{1i}\right| \leq \sqrt{n'd} + \frac{n'}{d^{3/2}}, \tag{81}$$

$$\left\|\sum_{i=1}^{n'} z_i \mathbb{I}_{0i}\mathbb{I}_{1i}\right\|_2 \leq 6\sigma\sqrt{n'd}. \tag{82}$$

Conditioned on $\mathbb{I}_1 = 1$, we also have

$$\mathbb{E}[\mathbb{I}_0\mathbb{I}_1|\mathbb{I}_1 = 1, \hat{\theta}_{\text{int}}] = \mathbb{P}(s \in [a', b']) \tag{83}$$

$$\geq \int_{\frac{d}{2\sigma\|\hat{\theta}_{\text{int}}\|_2}(-8/10+4/\sqrt{n})}^{\frac{d}{2\sigma\|\hat{\theta}_{\text{int}}\|_2}(8/10-4/\sqrt{n})} \frac{1}{\sqrt{2\pi}} e^{-t^2/2} dt \tag{84}$$

$$\geq 1 - 2\int_{\frac{d}{4\sigma\|\hat{\theta}_{\text{int}}\|_2}}^{+\infty} \frac{1}{\sqrt{2\pi}} e^{-t^2/2} dt \tag{85}$$

$$\geq 1 - 2\int_{\frac{\sqrt{C\log d}}{12}}^{+\infty} \frac{1}{\sqrt{2\pi}} e^{-t^2/2} dt \tag{86}$$

$$\geq 1 - \frac{1}{d}. \tag{87}$$

Let $m = \frac{n'}{2}\left(1 - \frac{1}{d}\right)$. Then by Chernoff's bound, we have

$$\mathbb{P}\left(\left|\sum_{i=1}^{n'} \mathbb{I}_{0i}\mathbb{I}_{1i} - m\right| \geq \frac{1}{2}m\right) \leq e^{-c'm} \tag{88}$$

for an absolute constant $c' > 0$. That is, with probality $\geq 1 - e^{-cn'}$, we have $\sum_{i=1}^{n'} \mathbb{I}_{0i}\mathbb{I}_{1i} \geq n'/5$.

**Case 2.** Next, let's consider the case when $\tilde{\mathbf{x}}$ is from $\mathcal{N}(\mu_o, \sigma^2 I)$. More precisely, we condition on a fixed $\hat{\theta}_{\text{int}}$ and condition on $\mathbb{I}_2 = 1$. Similar to case 1, we have

$$z = -11\mu + s \cdot \bar{\theta}_{\text{int}} + z_2 \tag{89}$$

where $s \sim \mathcal{N}(0, \sigma^2)$ and $z_2$ is a Gaussian distribution in the subspace orthogonal to $\bar{\theta}_{\text{int}}$. So

$$\mathbb{E}[(z + 11\mu)\mathbb{I}_0\mathbb{I}_2|\mathbb{I}_2 = 1, \hat{\theta}_{\text{int}}] = \mathbb{E}[s\mathbb{I}_0|\mathbb{I}_2 = 1, \hat{\theta}_{\text{int}}] \cdot \bar{\theta}_{\text{int}}. \tag{90}$$

For this,

$$\mathbb{E}[s\mathbb{I}\{f(\tilde{\mathbf{x}}) \in [a, b]\}|\mathbb{I}_2 = 1, \hat{\theta}_{\text{int}}] \cdot \bar{\theta}_{\text{int}} = \mathbb{E}\left[s \cdot \mathbb{I}\{s \in [a'', b'']\}\right] \cdot \bar{\theta}_{\text{int}} \tag{91}$$

where

$$a'' = \frac{-20\mu^\top \delta_{\text{int}} - 20\|\mu\|_2^2 + d}{2\|\hat{\theta}_{\text{int}}\|_2}, \tag{92}$$

$$b'' = \frac{-20\mu^\top \delta_{\text{int}} - 20\|\mu\|_2^2 + 3d}{2\|\hat{\theta}_{\text{int}}\|_2}. \tag{93}$$

By the bound on $|\mu^\top \delta_{\text{int}}|$ and $\|\delta_{\text{int}}\|_2$, we have

$$|\mathbb{E}\left[s \cdot \mathbb{I}\{s \in [a'', b'']\}\right]| \leq \left| \int_{\frac{-d}{2\sigma\|\hat{\theta}_{\text{int}}\|_2}(21+20/\sqrt{n})}^{\frac{-d}{2\sigma\|\hat{\theta}_{\text{int}}\|_2}(15-20/\sqrt{n})} \sigma t \frac{1}{\sqrt{2\pi}} e^{-t^2/2} dt \right| \tag{94}$$

$$\leq \frac{8d}{2\sigma\|\hat{\theta}_{\text{int}}\|_2}\sigma \frac{22d}{2\sigma\|\hat{\theta}_{\text{int}}\|_2} \frac{1}{\sqrt{2\pi}} e^{-\frac{1}{2}\left(\frac{14d}{2\sigma\|\hat{\theta}_{\text{int}}\|_2}\right)^2} \tag{95}$$

$$\leq \frac{44d^2}{\sigma\|\hat{\theta}_{\text{int}}\|_2^2} e^{-\frac{20d^2}{\sigma^2\|\hat{\theta}_{\text{int}}\|_2^2}} \tag{96}$$

$$\leq \frac{1}{d^2}. \tag{97}$$

We also have

$$\left|\mathbb{E}\left[\mathbb{I}_0 \mathbb{I}_2 | \mathbb{I}_2 = 1, \hat{\theta}_{\text{int}}\right]\right| = |\mathbb{E}\left[\mathbb{I}\{s \in [a'', b'']\}\right]| \tag{98}$$

$$\leq \int_{\frac{-d}{2\sigma\|\hat{\theta}_{\text{int}}\|_2}(21+20/\sqrt{n})}^{\frac{-d}{2\sigma\|\hat{\theta}_{\text{int}}\|_2}(15-20/\sqrt{n})} \frac{1}{\sqrt{2\pi}} e^{-t^2/2} dt \tag{99}$$

$$\leq \frac{d}{2\sigma\|\hat{\theta}_{\text{int}}\|_2}(6 + 40/\sqrt{n}) \frac{1}{\sqrt{2\pi}} e^{-\frac{1}{2}\left(\frac{14d}{2\sigma\|\hat{\theta}_{\text{int}}\|_2}\right)^2} \tag{100}$$

$$\leq \frac{1}{d^3}. \tag{101}$$

Combining the above, we have

$$\mathbb{E}[(z + 11\mu)\mathbb{I}_0 \mathbb{I}_2 | \hat{\theta}_{\text{int}}] = c_1 \cdot \bar{\theta}_{\text{int}} \tag{102}$$

for a constant $c_1$ satisfying $|c_1| \leq 1/d^2$. Furthermore, $(z + 11\mu)\mathbb{I}_0 \mathbb{I}_2 \mid \hat{\theta}_{\text{int}}$ is truncated Gaussian and thus is sub-Gaussian with sub-Gaussian norm bounded by $\sigma$. Then by sub-Gaussian concentration bounds, we have

$$\mathbb{P}\left(\left|\sum_{i=1}^{n'} \mu^\top (z_i + 11\mu)\mathbb{I}_{0i}\mathbb{I}_{2i} - \sum_{i=1}^{n'} \mu^\top \mathbb{E}[(z_i + 11\mu)\mathbb{I}_{0i}\mathbb{I}_{2i}|\hat{\theta}_{\text{int}}]\right| \geq \sqrt{n'd} \mid \hat{\theta}_{\text{int}}\right) \leq e^{-cd/\sigma^2}, \tag{103}$$

$$\mathbb{P}\left(\left\|\sum_{i=1}^{n'}(z_i + 11\mu)\mathbb{I}_{0i}\mathbb{I}_{2i}\right\|_2 \geq 4\sigma\sqrt{n'd} + 2\sqrt{n'd} \mid \hat{\theta}_{\text{int}}\right) \leq e^{-d/\sigma^2}, \tag{104}$$

for some constant $c > 0$. Also by Hoeffding's bound, we have

$$\mathbb{P}\left(\left|\sum_{i=1}^{n'}\mathbb{I}_{0i}\mathbb{I}_{2i} - \sum_{i=1}^{n'}\mathbb{E}[\mathbb{I}_{0i}\mathbb{I}_{2i}|\hat{\theta}_{\text{int}}]\right| \geq \sqrt{n'd/\sigma^2} \mid \hat{\theta}_{\text{int}}\right) \leq 2e^{-2d/\sigma^2}. \tag{105}$$

In other words, with probability $\geq 1 - 4e^{-cd/\sigma^2}$, we have

$$\left|\sum_{i=1}^{n'}\mu^\top z_i \mathbb{I}_{0i}\mathbb{I}_{2i}\right| \leq \sqrt{n'd} + 22d\sqrt{n'd}\sigma^2 + \frac{n'}{d^2}\sqrt{d} + 22d \cdot \frac{n'}{d^3} \leq \sqrt{n'd}\left(1 + \frac{22d}{\sigma}\right) + \frac{n'}{d^{3/2}}, \tag{106}$$

$$\left\|\sum_{i=1}^{n'} z_i \mathbb{I}_{0i}\mathbb{I}_{2i}\right\|_2 \leq 6\sigma\sqrt{n'd} + 22\sqrt{d}\left(\sqrt{\frac{n'd}{\sigma^2}} + \frac{n'}{d^3}\right) \leq \sqrt{n'd}\left(6\sigma + 22\sqrt{\frac{d}{\sigma^2}}\right) + \frac{22n'}{d^{5/2}}. \tag{107}$$

Combining equation 79, equation 80, equation 88 and equation 106, equation 107 together, we get with probability $\geq 1 - Ce^{-cd/\sigma^2}$,

$$|\mu^\top \delta_{\text{om}}| \leq C\sqrt{\frac{d}{n'}} \left(1 + \frac{22d}{\sigma}\right) + \frac{C}{d^{3/2}}, \tag{108}$$

$$\|\delta_{\text{om}}\|_2 \leq C\sqrt{\frac{d}{n'}} \left(6\sigma + 22\sqrt{\frac{d}{\sigma^2}}\right) + \frac{C}{d^{5/2}}. \tag{109}$$

Then $\frac{\mu^\top \hat{\theta}_{\text{om}}}{\|\hat{\theta}_{\text{om}}\|_2}$ can be lower bounded by

$$\frac{\mu^\top \hat{\theta}_{\text{om}}}{\|\hat{\theta}_{\text{om}}\|_2} = \frac{\mu^\top \mu + \mu^\top \delta_{\text{om}}}{\|\mu + \delta_{\text{om}}\|_2} \tag{110}$$

$$\geq \frac{\mu^\top \mu + \mu^\top \delta_{\text{om}}}{\|\mu\|_2 + \|\delta_{\text{om}}\|_2} \tag{111}$$

$$\geq \frac{d(1 - 1/\sqrt{d})}{\sqrt{d}(1 + 1/\sqrt{d})} \tag{112}$$

$$\geq \sqrt{d}\left(1 - \frac{2}{\sqrt{d}}\right). \tag{113}$$

The proof is completed by plugging the above into the closed form expressions equation 16 and equation 22 of the errors. $\square$

## B    DETAILS OF EXPERIMENTS

### B.1    EXPERIMENTAL SETTINGS

**Software and Hardware.** We run all experiments with PyTorch and NVIDIA GeForce RTX 2080Ti GPUs.

**Number of Evaluation Runs.** We run all experiments once with fixed random seeds.

**In-distribution Datasets.**        We use SVHN (Netzer et al., 2011), CIFAR-10 and CIFAR-100 (Krizhevsky et al., 2009) as in-distribution datasets. SVHN has 10 classes and contains 73,257 training images. CIFAR-10 and CIFAR-100 have 10 and 100 classes, respectively. Both datasets consist of 50,000 training images and 10,000 test images.

**Auxiliary OOD Datasets.** We provide the details of auxiliary OOD datasets below. For each auxiliary OOD dataset, we use random cropping with padding of 4 pixels to generate $32 \times 32$ images, and further augment the data by random horizontal flipping. We don't use any image corruptions to augment the data.

1. **TinyImages.** 80 Million Tiny Images (TinyImages) (Torralba et al., 2008) is a dataset that contains 79,302,017 images collected from the Web. The images in the dataset are stored as $32 \times 32$ color images. Since CIFAR-10 and CIFAR-100 are labeled subsets of the TinyImages dataset, we need to remove those images in the dataset that belong to CIFAR-10 or CIFAR-100. We follow the same deduplication procedure as in (Hendrycks et al., 2018) and remove all examples in this dataset that appear in CIFAR-10 or CIFAR-100. Even after deduplication, the auxiliary OOD dataset may still contain some in-distribution data if we use CIFAR-10 or CIFAR-100 as in-distribution datasets, but the fraction of them is low.

2. **ImageNet-RC.** We use the downsampled ImageNet dataset (ImageNet$64 \times 64$) (Chrabaszcz et al., 2017), which is a downsampled variant of the original ImageNet dataset. It contains 1,281,167 images with image size of $64 \times 64$ and 1,000 classes. Some of the classes overlap with CIFAR-10 or CIFAR-100 classes. Since we don't use any label information from the dataset, we can say that the auxiliary OOD dataset is unlabeled. Since we randomly crop the $64 \times 64$ images into $32 \times 32$ images with padding of 4 pixels, with high probability, the resulting images won't contain objects belonging to the in-distribution classes even if the original images contain objects belonging to those classes. Therefore, we still can have a

lot of OOD data for training and the fraction of in-distribution data in the auxiliary OOD dataset is low. We call this auxiliary OOD dataset ImageNet-RC.

**OOD Test Datasets.** We provide the details of OOD test datasets below. All images are of size $32 \times 32$.

1. **SVHN.** The SVHN dataset (Netzer et al., 2011) contains color images of house numbers. There are ten classes of digits `0-9`. The original test set has 26,032 images. We randomly select 1,000 test images for each class and form a new test dataset of 10,000 images for evaluation.

2. **Textures.** The Describable Textures Dataset (DTD) (Cimpoi et al., 2014) contains textural images in the wild. We include the entire collection of 5640 images for evaluation.

3. **Places365.** The Places365 dataset (Zhou et al., 2017) contains large-scale photographs of scenes with 365 scene categories. There are 900 images per category in the test set. We randomly sample 10,000 images from the test set for evaluation.

4. **LSUN (crop) and LSUN (resize).** The Large-scale Scene UNderstanding dataset (LSUN) has a testing set of 10,000 images of 10 different scenes (Yu et al., 2015). We construct two datasets, `LSUN-C` and `LSUN-R`, by randomly cropping image patches of size $32 \times 32$ and downsampling each image to size $32 \times 32$, respectively.

5. **iSUN.** The iSUN (Xu et al., 2015) consists of a subset of SUN images. We include the entire collection of 8925 images in iSUN.

6. **CIFAR-10.** We use the test set of CIFAR-10, which contains 10,000 images.

7. **Gaussian Noise.** The synthetic Gaussian noise dataset consists of 10,000 random 2D Gaussian noise images, where each RGB value of every pixel is sampled from an i.i.d Gaussian distribution with mean 0.5 and unit variance. We further clip each pixel value into the range [0,1].

8. **Uniform Noise.** The synthetic uniform noise dataset consists 10,000 images where each RGB value of every pixel is independently and identically sampled from a uniform distribution on [0,1].

**Architectures and Training Configurations.** We use the state-of-the-art neural network architecture DenseNet (Huang et al., 2017) and WideResNet (Zagoruyko & Komodakis, 2016). For DenseNet, we follow the same setup as in (Huang et al., 2017), with depth $L = 100$, growth rate $k = 12$ (Dense-BC) and dropout rate 0. For WideResNet, we also follow the same setup as in (Zagoruyko & Komodakis, 2016), with depth of 40 and widening parameter $k = 4$ (WRN-40-4). All neural networks are trained with stochastic gradient descent with Nesterov momentum (Duchi et al., 2011; Kingma & Ba, 2014). We set momentum 0.9 and $\ell_2$ weight decay with a coefficient of $10^{-4}$ for all model training. Specifically, for SVHN, we train the networks for 20 epochs and the initial learning rate of 0.1 decays by 0.1 at 10, 15, 18 epoch; for CIFAR-10 and CIFAR-100, we train the networks for 100 epochs and the initial learning rate of 0.1 decays by 0.1 at 50, 75, 90 epoch. In ATOM and NTOM, we use batch size 64 for in-distribution data and 128 for out-of-distribution data. To solve the inner max of the robust training objective in ATOM, we use PGD with $\epsilon = 8/255$, the number of iterations of 5, the step size of $2/255$, and random start.

## B.2  AVERAGE RUNTIME

We run our experiments using a single GPU on a machine with 4 GPUs and 32 cores. The estimated average runtime for each method is summarized in Table 3.

## B.3  OOD DETECTION METHODS

We consider eight common OOD detection methods listed in Table 4 and describe each method in detail below.

**Maximum Softmax Probability (MSP).** Hendrycks & Gimpel propose to use $\max_i F_i(x)$ as confidence scores to detect OOD examples, where $F(x)$ is the softmax output of the neural network.

| Method | Training | Evaluation |
|--------|----------|------------|
| MSP | 2.5 h | 4 h |
| ODIN | 2.5 h | 4 h |
| Mahalanobis | 2.5 h | 20 h |
| SOFL | 14 h | 4 h |
| OE | 5 h | 4 h |
| ACET | 17 h | 4 h |
| CCU | 6.7 h | 4 h |
| ROWL | 24 h | 4 h |
| ATOM (ours) | 21 h | 4 h |

Table 3: The estimated average runtime for each result. We use DenseNet as network architecture. $h$ means hour. For MSP, ODIN, and Mahalanobis, we use standard training. The evaluation includes four OOD detection tasks listed in Section 2.

**ODIN.** Liang et al. computes calibrated confidence scores using temperature scaling and input perturbation techniques. In all of our experiments, we set temperature scaling parameter $T = 1000$. We choose perturbation magnitude $\eta$ by validating on 1000 images randomly sampled from in-distribution test set $\mathcal{D}_{\text{in}}^{\text{test}}$ and 1000 images randomly sampled from auxiliary OOD dataset $\mathcal{D}_{\text{out}}^{\text{auxiliary}}$, which does not depend on prior knowledge of test OOD datasets. For DenseNet, we set $\eta = 0.0006$ for SVHN, $\eta = 0.0016$ for CIFAR-10, and $\eta = 0.0012$ for CIFAR-100. For WideResNet, we set $\eta = 0.0002$ for SVHN, $\eta = 0.0006$ for CIFAR-10, and $\eta = 0.0012$ for CIFAR-100.

**Mahalanobis.** Lee et al. propose to use Mahalanobis distance-based confidence scores to detect OOD samples. Following Lee et al., we use 1000 examples randomly selected from in-distribution test set $\mathcal{D}_{\text{in}}^{\text{test}}$ and adversarial examples generated by FGSM (Goodfellow et al., 2014) on them with perturbation size of 0.05 to train the Logistic Regression model and tune the noise perturbation magnitude $\eta$. $\eta$ is chosen from $\{0.0, 0.01, 0.005, 0.002, 0.0014, 0.001, 0.0005\}$, and the optimal parameters are chosen to minimize the FPR at FNR 5%.

**Outlier Exposure (OE).** Outlier Exposure (Hendrycks et al., 2018) makes use of a large, auxiliary OOD dataset $\mathcal{D}_{\text{out}}^{\text{auxiliary}}$ to enhance the performance of existing OOD detection. We train from scratch with $\lambda = 0.5$, and use in-distribution batch size of 64 and out-distribution batch size of 128 in our experiments. Other training parameters are specified in Section B.1.

**Self-Supervised OOD Feature Learning (SOFL).** Mohseni et al. add an auxiliary head to the network and train in for the OOD detection task. They first use a full-supervised training to learn in-distribution training data for the main classification head and then a self-supervised training with OOD training set for the auxiliary head. Following the original setting, we set $\lambda = 5$ and use an in-distribution batch size of 64 and an out-distribution batch size of 320 in all of our experiments. In SVHN and CIFAR-10, we use 5 reject classes, while in CIFAR-100, we use 10 reject classes. We first train the model with the full-supervised learning using the training parameters specified in Section B.1 and then continue to train with the self-supervised OOD feature learning using the same training parameters. We use the large, auxiliary OOD dataset $\mathcal{D}_{\text{out}}^{\text{auxiliary}}$ as out-of-distribution training dataset.

**Adversarial Confidence Enhancing Training (ACET).** Hein et al. propose Adversarial Confidence Enhancing Training to enforce low model confidence for the OOD data point, as well as worst-case adversarial example in the neighborhood of an OOD example. We use the large, auxiliary OOD dataset $\mathcal{D}_{\text{out}}^{\text{auxiliary}}$ as an OOD training dataset instead of using random noise data for a fair comparison. In all of our experiments, we set $\lambda = 1.0$. For both in-distribution and out-distribution, we use a batch size of 128. To solve the inner max of the training objective, we also apply PGD with $\epsilon = 8/255$, the number of iterations of 5, the step size of $2/255$, and random start to a half of a minibatch while keeping the other half clean to ensure proper performance on both perturbed and clean OOD examples for a fair comparison. Other training parameters are specified in Section B.1.

**Certified Certain Uncertainty (CCU).** Certified Certain Uncertainty (Meinke & Hein, 2019) gives guarantees on the confidence of the classifier decision far away from the training data. We use the same training set up as in the paper and code, except we use our training configurations specified in Section B.1.

| | MSP | ODIN | Mahalanobis | SOFL | OE | ACET | CCU | ROWL | ATOM (ours) |
|---|---|---|---|---|---|---|---|---|---|
| Natural OOD | ✓ | ✓ | ✓ | ✓ | ✓ | ✓ | ✓ | ✓ | ✓ |
| $L_\infty$ OOD | ✗ | ✗ | ✗ | ✗ | ✗ | ✓ | ✓ | ✓ | ✓ |
| Corruption OOD | ✗ | ✗ | ✗ | ✗ | ✗ | ✗ | ✗ | ✗ | ✓ |
| Comp. OOD | ✗ | ✗ | ✗ | ✗ | ✗ | ✗ | ✗ | ✗ | ✓ |

Table 4: Common OOD detection methods and a family of natural and perturbed OOD examples we considered.

**Robust Open-World Deep Learning (ROWL).** Sehwag et al. propose to introduce additional background classes for OOD datasets and perform adversarial training on both the in- and out-of-distribution datasets to achieve robust open-world classification. When an input is classified as the background classes, it is considered as an OOD example. Thus, ROWL gives binary OOD scores (either 0 or 1) to the inputs. In our experiments, we only have one background class and randomly sample data points from the large, auxiliary OOD dataset $\mathcal{D}_{\text{out}}^{\text{auxiliary}}$ to form the OOD dataset. To ensure data balance across classes, we include 7,325 OOD data points for SVHN, 5,000 OOD data points for CIFAR-10 and 500 OOD data points for CIFAR-100. During training, we mix the in-distribution data and OOD data, and use a batch size of 128. To solve the inner max of the training objective, we use PGD with $\epsilon = 8/255$, the number of iterations of 5, the step size of $2/255$, and random start. Other training parameters are specified in Section B.1.

### B.4 ADVERSARIAL ATTACKS FOR OOD DETECTION METHODS

We propose adversarial attack objectives for different OOD detection methods. We consider a family of adversarial perturbations for the OOD inputs: (1) $L_\infty$-norm bounded attack (white-box attack); (2) common image corruptions attack (black-box attack); (3) compositional attack which combines common image corruptions attack and $L_\infty$ norm bounded attack (white-box attack).

$L_\infty$ **norm bounded attack.** For data point $\mathbf{x} \in \mathbb{R}^d$, the $L_\infty$ norm bounded perturbation is defined as

$$\Omega_{\infty,\epsilon}(\mathbf{x}) = \{\delta \in \mathbb{R}^d \mid \|\delta\|_\infty \leq \epsilon \wedge \mathbf{x} + \delta \text{ is valid}\}, \tag{114}$$

where $\epsilon$ is the adversarial budget. $\mathbf{x} + \delta$ is considered valid if the values of $\mathbf{x} + \delta$ are in the image pixel value range.

For MSP, ODIN, OE, ACET, and CCU methods, we propose the following attack objective to generate adversarial OOD example on a clean OOD input $\mathbf{x}$:

$$\mathbf{x}' = \underset{\mathbf{x}' \in \Omega_{\infty,\epsilon}(\mathbf{x})}{\arg\max} -\frac{1}{K}\sum_{i=1}^{K} \log F(\mathbf{x}')_i \tag{115}$$

where $F(\mathbf{x})$ is the softmax output of the classifier network.

For Mahalanobis method, we propose the following attack objective to generate adverasrial OOD example on OOD input $\mathbf{x}$:

$$\mathbf{x}' = \underset{\mathbf{x}' \in \Omega_{\infty,\epsilon}(\mathbf{x})}{\arg\max} -\log \frac{1}{1 + e^{-(\sum_\ell \alpha_\ell M_\ell(\mathbf{x}')+b)}}, \tag{116}$$

where $M_\ell(\mathbf{x}')$ is the Mahalanobis distance-based confidence score of $\mathbf{x}'$ from the $\ell$-th feature layer, $\{\alpha_\ell\}$ and $b$ are the parameters of the logistic regression model.

For SOFL method, we propose the following attack objective to generate adversarial OOD example for an input $\mathbf{x}$:

$$\mathbf{x}' = \underset{\mathbf{x}' \in \Omega_{\infty,\epsilon}(\mathbf{x})}{\arg\max} -\log \sum_{i=K+1}^{K+R} \bar{F}(\mathbf{x}')_i \tag{117}$$

where $\bar{F}(\mathbf{x})$ is the softmax output of the whole neural network (including auxiliary head) and $R$ is the number of reject classes.

For ROWL and ATOM method, we propose the following attack objective to generate adverasrial OOD example on OOD input $\mathbf{x}$:

$$\mathbf{x}' = \underset{\mathbf{x}' \in \Omega_{\infty,\epsilon}(\mathbf{x})}{\arg\max} -\log \hat{F}(\mathbf{x}')_{K+1} \tag{118}$$

where $\hat{F}(\mathbf{x})$ is the softmax output of the (K+1)-way neural network.

Due to computational concerns, by default, we will use PGD with $\epsilon = 8/255$, the number of iterations of 40, the step size of 1/255 and random start to solve these attack objectives. We also perform ablation study experiments on the attack strength for ACET and ATOM, see Appendix B.10.

**Common Image Corruptions attack.** We use common image corruptions introduced in (Hendrycks & Dietterich, 2019). We apply 15 types of algorithmically generated corruptions from noise, blur, weather, and digital categories to each OOD image. Each type of corruption has five levels of severity, resulting in 75 distinct corruptions. Thus, for each OOD image, we generate 75 corrupted images and then select the one with the lowest OOD score (or highest confidence score to be in-distribution). Note that we only need the outputs of the OOD detectors to construct such adversarial OOD examples; thus it is a black-box attack.

**Compositional Attack.** For each OOD image, we first apply common image corruptions attack, and then apply the $L_\infty$-norm bounded attack to generate adversarial OOD examples.

### B.5 VISUALIZATIONS OF FOUR TYPES OF OOD SAMPLES

We show visualizations of four types of OOD samples in Figure 4.

### B.6 HISTOGRAM OF OOD SCORES

In Figure 5, we show histogram of OOD scores for model snapshots trained on CIFAR-10 (in-distribution) using objective (4) **without** informative outlier mining. We plot every ten epochs for a model trained for a total of 100 epochs. We observe that the model quickly converges to a solution where OOD score distribution becomes dominated by *easy* examples with scores closer to 1. This is exacerbated as the model is trained for longer.

### B.7 CHOOSE BEST q USING VALIDATION DATASET

We create a validation OOD dataset by sampling 10,000 images from the 80 Million Tiny Images (Torralba et al., 2008), which is disjoint from our training data. We choose $q$ from $\{0, 0.125, 0.25, 0.5, 0.75\}$. The results on the validation dataset are shown in Table 5. We select the best model based on the average FPR at 5% FNR across four types of OOD inputs. Based on the results, the optimal $q$ is 0 for SVHN, 0.125 for CIFAR-10 and 0.5 for CIFAR-100.

### B.8 EFFECT OF AUXILIARY OOD DATASETS

We present results in Table 6, where we use an alternative auxiliary OOD dataset ImageNet-RC. The details of ImageNet-RC is provided in Section B.1. We use the same hyperparameters as used in training with TinyImages auxiliary data. For all three in-distribution datasets, we find that using $q = 0$ results in the optimal performance.

### B.9 EFFECT OF NETWORK ARCHITECTURE

We perform experiments to evaluate different OOD detection methods using WideResNet, see Table 7. For both ATOM and NTOM, we use the same hyperparameters as those selected for DenseNet and find that it also leads to good results.

### B.10 EFFECT OF PGD ATTACK STRENGTH

To see the effect of using stronger PGD attacks, we evaluate ACET (best baseline) and ATOM on $L_\infty$ attacked OOD and compositionally attacked OOD inputs with 100 iterations and 5 random restarts. Results are provided in Table 8. Under stronger PGD attack, ATOM outperforms ACET.

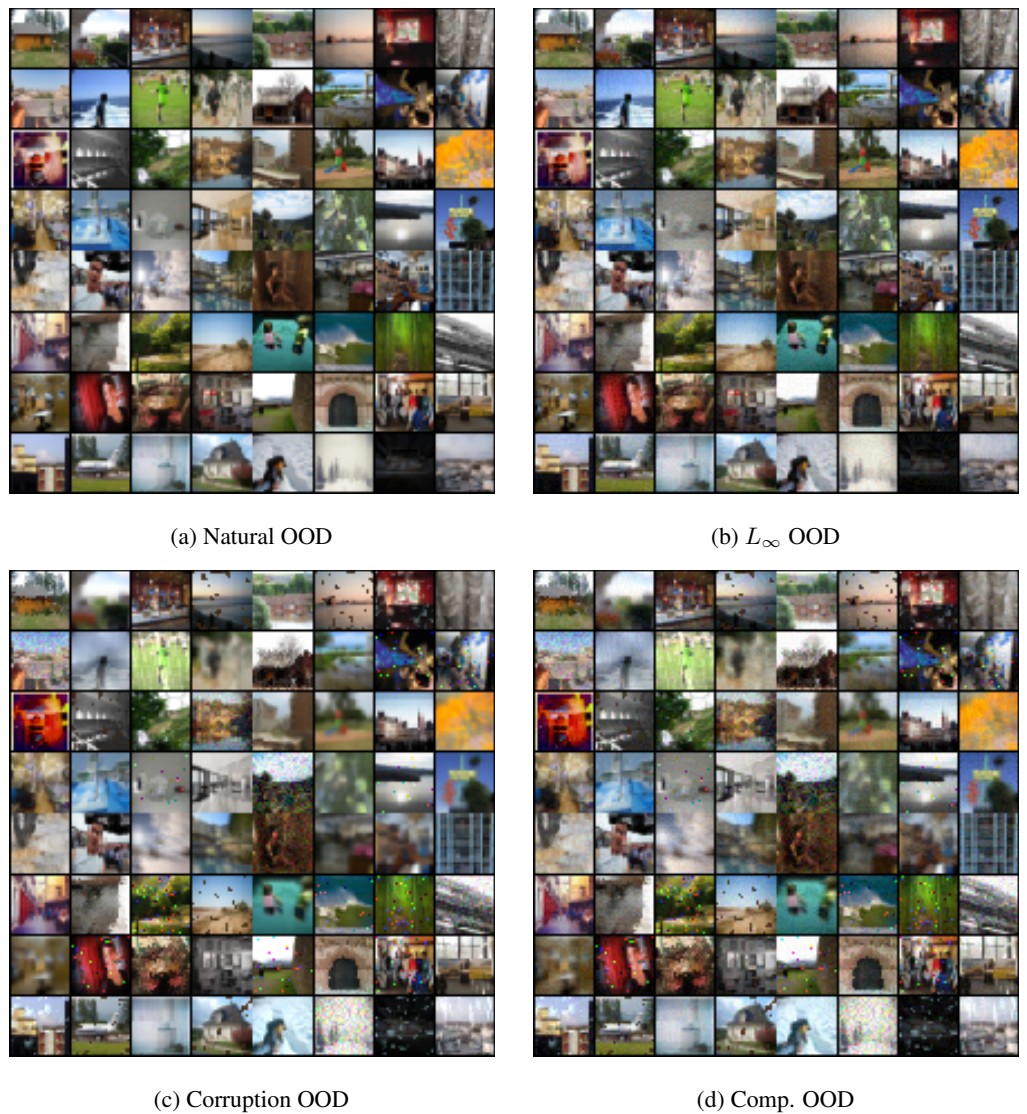

(a) Natural OOD

(b) $L_\infty$ OOD

(c) Corruption OOD

(d) Comp. OOD

Figure 4: Examples of four types of OOD samples.

## B.11 Evaluation on Random Noise OOD Data

We report the performance of OOD detectors using DenseNet on random noise OOD test datasets in Table 9 (SVHN), Table 10 (CIFAR-10) and Table 11 (CIFAR-100).

## B.12 Performance of OOD Detector and Classifier on In-distribution Data

We summarize the performance of OOD detector $G(x)$ and image classifier $f(x)$ on in-distribution test data. See Table 12 for DenseNet and Table 13 for WideResNet. From the results, we can see that ATOM improves the OOD detection performance while achieving in-distribution classification accuracy that is on par with a pre-trained network.

## B.13 Complete Experimental Results

We report the performance of OOD detectors using DenseNet on each of the six natural OOD test datasets in Table 14 (SVHN), Table 15 (CIFAR-10) and Table 16 (CIFAR-100).

| $\mathcal{D}_{\text{in}}^{\text{test}}$ | Method | FPR (5% FNR) ↓ | AUROC ↑ | FPR (5% FNR) ↓ | AUROC ↑ | FPR (5% FNR) ↓ | AUROC ↑ | FPR (5% FNR) ↓ | AUROC ↑ |
|---|---|---|---|---|---|---|---|---|---|
| | | Natural OOD | | Corruption OOD | | $L_\infty$ OOD | | Comp. OOD | |
| | ATOM (q=0.0) | **0.01** | 99.97 | **4.02** | 98.50 | **8.08** | 98.14 | **96.79** | 43.77 |
| | ATOM (q=0.125) | 2.16 | 99.39 | 36.62 | 94.79 | 62.38 | 73.20 | 99.98 | 4.62 |
| SVHN | ATOM (q=0.25) | 1.87 | 99.38 | 41.86 | 94.20 | 75.85 | 57.75 | 100.00 | 1.26 |
| | ATOM (q=0.5) | 2.73 | 99.18 | 45.02 | 93.85 | 83.01 | 49.38 | 99.99 | 1.96 |
| | ATOM (q=0.75) | 4.97 | 98.83 | 56.51 | 92.13 | 85.98 | 40.89 | 100.00 | 1.16 |
| | ATOM (q=0.0) | 5.34 | 98.35 | 43.61 | 91.85 | 22.08 | 92.81 | 59.17 | 83.56 |
| CIFAR-10 | ATOM (q=0.125) | **4.77** | 98.31 | **27.49** | 94.24 | **5.42** | 98.19 | **29.02** | 93.76 |
| | ATOM (q=0.25) | 5.70 | 98.11 | 28.13 | 93.68 | 19.28 | 95.71 | 40.68 | 91.17 |
| | ATOM (q=0.5) | 8.83 | 97.66 | 39.74 | 91.53 | 9.80 | 97.42 | 44.10 | 90.45 |
| | ATOM (q=0.75) | 12.42 | 97.02 | 45.85 | 90.53 | 13.40 | 96.83 | 46.30 | 90.42 |
| | ATOM (q=0.0) | 44.85 | 91.58 | 98.76 | 64.78 | 53.17 | 85.26 | 98.95 | 58.38 |
| CIFAR-100 | ATOM (q=0.125) | 36.75 | 92.90 | 96.22 | 73.33 | 38.79 | 91.42 | 96.33 | 71.74 |
| | ATOM (q=0.25) | **34.66** | 92.62 | 94.13 | 73.86 | **35.84** | 91.24 | 94.33 | 70.35 |
| | ATOM (q=0.5) | 35.04 | 91.36 | **89.28** | 71.78 | 36.76 | 90.21 | **89.57** | 70.62 |
| | ATOM (q=0.75) | 43.49 | 87.95 | 91.80 | 62.47 | 59.34 | 78.00 | 92.94 | 58.53 |

Table 5: Evaluate models on validation dataset. We use DenseNet as network architecture. ↑ indicates larger value is better, and ↓ indicates lower value is better. All values are percentages and are averaged over six different OOD test datasets mentioned in section 5.1. **Bold** numbers are superior results.

| $\mathcal{D}_{\text{in}}^{\text{test}}$ | Method | FPR (5% FNR) ↓ | AUROC ↑ | FPR (5% FNR) ↓ | AUROC ↑ | FPR (5% FNR) ↓ | AUROC ↑ | FPR (5% FNR) ↓ | AUROC ↑ |
|---|---|---|---|---|---|---|---|---|---|
| | | Natural OOD | | Corruption OOD | | $L_\infty$ OOD | | Comp. OOD | |
| | MSP | 38.84 | 93.57 | 99.68 | 68.48 | 99.89 | 1.39 | 100.00 | 0.19 |
| | ODIN | 31.45 | 93.52 | 97.11 | 63.21 | 99.86 | 0.61 | 100.00 | 0.05 |
| | Mahalanobis | 22.80 | 95.57 | 93.14 | 60.78 | 97.33 | 8.89 | 99.89 | 0.23 |
| | SOFL | **0.02** | **99.99** | **5.93** | **98.57** | 58.53 | 68.85 | 67.34 | 61.42 |
| SVHN | OE | 0.13 | 99.96 | 15.76 | 97.51 | 68.76 | 49.57 | 98.80 | 6.21 |
| | ACET | 0.31 | 99.94 | 29.02 | 95.65 | 2.37 | 99.51 | 30.58 | 95.20 |
| | CCU | 0.17 | 99.96 | 18.64 | 96.94 | 45.38 | 69.14 | 92.30 | 20.88 |
| | ROWL | 2.04 | 98.87 | 55.03 | 72.37 | 77.24 | 61.27 | 99.79 | 50.00 |
| | NTOM (ours) | **0.02** | **99.99** | 6.39 | 98.54 | 59.27 | 68.53 | 58.12 | 72.08 |
| | ATOM (ours) | **0.02** | **99.99** | 7.03 | 98.38 | **0.14** | **99.95** | **7.30** | **98.32** |
| | MSP | 50.54 | 91.79 | 100.00 | 58.35 | 100.00 | 13.82 | 100.00 | 13.67 |
| | ODIN | 21.65 | 94.66 | 99.37 | 51.44 | 99.99 | 0.18 | 100.00 | 0.01 |
| | Mahalanobis | 26.95 | 90.30 | 91.92 | 43.94 | 95.07 | 12.47 | 99.88 | 1.58 |
| | SOFL | 6.96 | 98.71 | 22.30 | 95.89 | 97.61 | 12.39 | 99.74 | 7.49 |
| CIFAR-10 | OE | 9.70 | 98.35 | 49.84 | 91.76 | 91.30 | 43.88 | 98.82 | 31.12 |
| | ACET | 10.72 | 98.01 | 53.85 | 90.19 | 17.10 | 96.01 | 55.21 | 89.78 |
| | CCU | 10.30 | 98.25 | 44.42 | 92.34 | 93.02 | 20.88 | 99.17 | 9.95 |
| | ROWL | 25.03 | 86.96 | 94.34 | 52.31 | 99.98 | 49.49 | 100.00 | 49.48 |
| | NTOM (ours) | **4.01** | **99.17** | **15.13** | **97.14** | 96.89 | 11.42 | 99.87 | 3.18 |
| | ATOM (ours) | 4.08 | 99.14 | 16.17 | 96.94 | **7.46** | **98.50** | **18.35** | **96.60** |
| | MSP | 78.05 | 76.11 | 100.00 | 30.04 | 100.00 | 2.25 | 100.00 | 2.06 |
| | ODIN | 56.77 | 83.62 | 100.00 | 36.95 | 100.00 | 0.14 | 100.00 | 0.00 |
| | Mahalanobis | 42.63 | 87.86 | 95.92 | 42.96 | 95.44 | 15.87 | 99.86 | 2.08 |
| | SOFL | 20.95 | 96.06 | 73.33 | 83.31 | 93.41 | 12.90 | 99.98 | 3.36 |
| CIFAR-100 | OE | 18.52 | 95.27 | 86.83 | 66.95 | 96.27 | 18.79 | 99.97 | 4.88 |
| | ACET | 19.79 | 94.76 | 81.63 | 70.04 | 26.23 | 91.46 | 81.95 | 69.67 |
| | CCU | 19.44 | 95.05 | 84.11 | 69.09 | 84.89 | 35.85 | 99.61 | 15.67 |
| | ROWL | 93.35 | 53.02 | 100.00 | 49.69 | 100.00 | 49.69 | 100.00 | 49.69 |
| | NTOM (ours) | 18.09 | 96.74 | 66.61 | 87.97 | 86.88 | 15.40 | 99.98 | 0.77 |
| | ATOM (ours) | **15.49** | **97.18** | **57.79** | **89.49** | **18.32** | **96.57** | **58.49** | **89.36** |

Table 6: Comparison with competitive OOD detection methods. We use ImageNet-RC as the auxiliary OOD dataset (see section B.1 for the details) for SOFL, OE, ACET, CCU, NTOM and ATOM. We use DenseNet as network architecture for all methods. We evaluate on four types of OOD inputs: (1) natural OOD, (2) corruption attacked OOD, (3) $L_\infty$ attacked OOD, and (4) compositionally attacked OOD inputs. ↑ indicates larger value is better, and ↓ indicates lower value is better. All values are percentages and are averaged over six natural OOD test datasets described in section 5.1. **Bold** numbers are superior results.

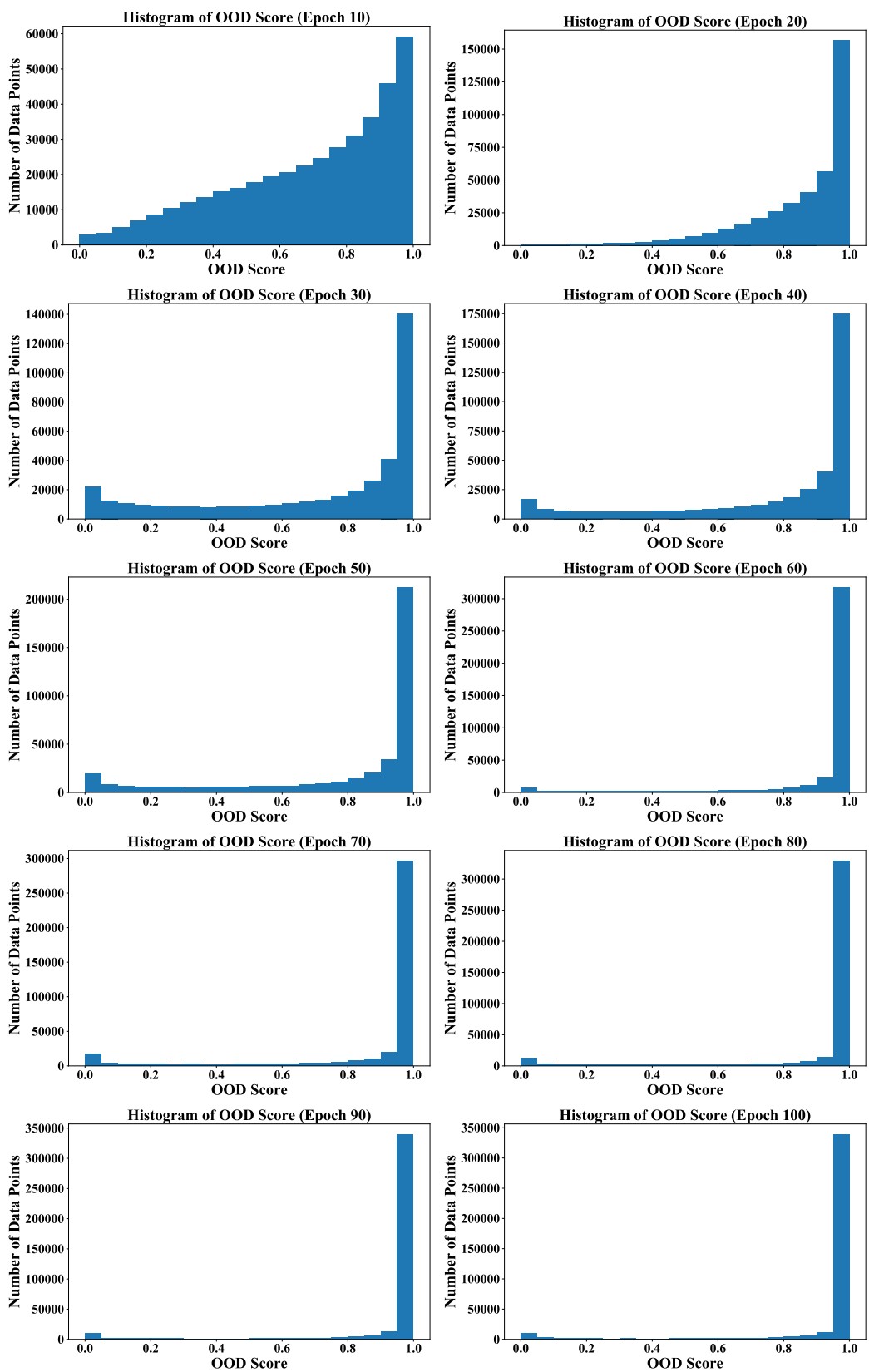

Figure 5: On CIFAR-10, we train the model with objective (4) for 100 epochs **without** informative outlier mining. For every 10 epochs, we randomly sample 400,000 data points from the large auxiliary OOD dataset and use the current model snapshot to calculate the OOD scores.

| $\mathcal{D}_{in}^{test}$ | Method | FPR (5% FNR) ↓ | AUROC ↑ | FPR (5% FNR) ↓ | AUROC ↑ | FPR (5% FNR) ↓ | AUROC ↑ | FPR (5% FNR) ↓ | AUROC ↑ |
|---|---|---|---|---|---|---|---|---|---|
| | | Natural OOD | | Corruption OOD | | $L_\infty$ OOD | | Comp. OOD | |
| **SVHN** | MSP | 42.78 | 91.15 | 99.99 | 51.84 | 100.00 | 0.15 | 100.00 | 0.05 |
| | ODIN | 43.65 | 89.37 | 99.85 | 44.43 | 100.00 | 0.06 | 100.00 | 0.00 |
| | Mahalanobis | 6.94 | 98.47 | 80.91 | 78.92 | 93.44 | 39.42 | 99.70 | 2.69 |
| | SOFL | **0.02** | **99.99** | **2.41** | **99.36** | 85.25 | 39.32 | 99.28 | 2.85 |
| | OE | 0.58 | 99.87 | 21.97 | 96.00 | 70.66 | 47.36 | 96.63 | 5.87 |
| | ACET | 0.43 | 99.92 | 18.12 | 97.11 | 20.75 | 89.55 | 99.66 | 1.80 |
| | CCU | 0.71 | 99.86 | 30.96 | 94.40 | 73.37 | 41.82 | 99.83 | 0.44 |
| | ROWL | 1.46 | 99.13 | 45.18 | 77.28 | 86.95 | 56.39 | 99.94 | 49.90 |
| | NTOM (ours) | **0.02** | **99.99** | 2.46 | 99.27 | 70.38 | 52.43 | 99.79 | 1.01 |
| | ATOM (ours) | 0.08 | 99.96 | 4.18 | 98.65 | **5.93** | **98.47** | **95.50** | **62.00** |
| **CIFAR-10** | MSP | 53.81 | 91.10 | 100.00 | 50.30 | 100.00 | 5.48 | 100.00 | 5.43 |
| | ODIN | 36.25 | 91.18 | 99.95 | 31.78 | 100.00 | 0.04 | 100.00 | 0.00 |
| | Mahalanobis | 23.93 | 92.48 | 86.78 | 56.90 | 85.75 | 42.63 | 99.22 | 17.51 |
| | SOFL | 2.15 | **99.14** | 38.03 | 93.35 | 99.99 | 0.43 | 100.00 | 0.24 |
| | OE | 3.03 | 98.83 | 56.00 | 91.19 | 99.97 | 0.26 | 100.00 | 0.04 |
| | ACET | 3.19 | 98.74 | 35.78 | 94.25 | 40.22 | 89.52 | 56.08 | 87.76 |
| | CCU | 2.61 | 98.75 | 36.31 | 93.75 | 99.93 | 0.86 | 99.99 | 0.41 |
| | ROWL | 17.21 | 90.79 | 86.09 | 56.35 | 99.94 | 49.43 | 99.99 | 49.40 |
| | NTOM (ours) | 1.86 | 99.10 | 23.77 | 94.58 | 99.92 | 0.84 | 99.96 | 0.33 |
| | ATOM (ours) | **1.50** | 98.87 | **13.17** | **96.36** | **33.66** | **90.89** | **40.07** | **88.99** |
| **CIFAR-100** | MSP | 81.92 | 74.32 | 100.00 | 28.10 | 100.00 | 3.20 | 100.00 | 3.14 |
| | ODIN | 68.76 | 79.13 | 100.00 | 26.44 | 100.00 | 0.15 | 100.00 | 0.01 |
| | Mahalanobis | 43.48 | 86.02 | 95.78 | 42.25 | 95.83 | 17.41 | 99.92 | 3.55 |
| | SOFL | 66.57 | 89.11 | 99.99 | 59.64 | 99.99 | 0.30 | 100.00 | 0.10 |
| | OE | 46.14 | 91.16 | 99.96 | 58.55 | 100.00 | 1.80 | 100.00 | 1.69 |
| | ACET | 46.79 | 91.67 | 99.76 | 62.48 | 95.66 | 41.38 | 99.99 | 36.65 |
| | CCU | 48.35 | 91.41 | 99.95 | 57.94 | 100.00 | 0.44 | 100.00 | 0.38 |
| | ROWL | 96.93 | 51.34 | 100.00 | 49.81 | 100.00 | 49.81 | 100.00 | 49.81 |
| | NTOM (ours) | 37.94 | 92.14 | 97.09 | 66.93 | 99.99 | 1.04 | 99.99 | 0.13 |
| | ATOM (ours) | **30.52** | **93.90** | **94.46** | **74.96** | **44.30** | **89.68** | **95.06** | **74.25** |

Table 7: Comparison with competitive OOD detection methods. We use WideResNet as network architecture for all methods. We evaluate on four types of OOD inputs: (1) natural OOD, (2) corruption attacked OOD, (3) $L_\infty$ attacked OOD, and (4) compositionally attacked OOD inputs. ↑ indicates larger value is better, and ↓ indicates lower value is better. All values are percentages and are averaged over six different OOD test datasets described in section 5.1. **Bold** numbers are superior results.

| $\mathcal{D}_{in}^{test}$ | Method | FPR (5% FNR) ↓ | AUROC ↑ | FPR (5% FNR) ↓ | AUROC ↑ |
|---|---|---|---|---|---|
| | | $L_\infty$ OOD | | Comp. OOD | |
| **SVHN** | ACET | 33.64 | 83.89 | 99.90 | 2.84 |
| | ATOM (ours) | **8.33** | **97.37** | **98.56** | **35.32** |
| **CIFAR-10** | ACET | 76.66 | 76.29 | 97.08 | 51.35 |
| | ATOM (ours) | **26.88** | **83.78** | **42.86** | **83.56** |
| **CIFAR-100** | ACET | 87.13 | 48.77 | 99.82 | 33.67 |
| | ATOM (ours) | **45.11** | **82.32** | **93.74** | **65.88** |

Table 8: Evaluation on $L_\infty$ attacked OOD and compositionally attacked OOD inputs with strong PGD attack (100 iterations and 5 random restarts). We use DenseNet as network architecture for all methods. ↑ indicates larger value is better, and ↓ indicates lower value is better. All values are percentages and are averaged over six different OOD test datasets described in section 5.1. **Bold** numbers are superior results.

| $\mathcal{D}_{out}^{test}$ | Method | FPR (5% FNR) ↓ | AUROC ↑ | FPR (5% FNR) ↓ | AUROC ↑ | FPR (5% FNR) ↓ | AUROC ↑ | FPR (5% FNR) ↓ | AUROC ↑ |
|---|---|---|---|---|---|---|---|---|---|
| | | **Natural OOD** | | **Corruption OOD** | | $L_\infty$ **OOD** | | **Comp. OOD** | |
| **Gaussian Noise** | MSP | 56.30 | 90.16 | 100.00 | 55.70 | 100.00 | 5.25 | 100.00 | 0.14 |
| | ODIN | 58.49 | 88.24 | 100.00 | 44.91 | 100.00 | 0.87 | 100.00 | 0.01 |
| | Mahalanobis | **0.00** | 99.92 | 97.34 | 77.03 | 1.51 | 98.73 | 100.00 | 0.00 |
| | SOFL | **0.00** | **100.00** | 0.02 | 99.93 | **0.00** | 99.98 | 100.00 | 0.42 |
| | OE | **0.00** | **100.00** | 0.93 | 99.74 | **0.00** | **100.00** | 98.71 | 2.20 |
| | ACET | **0.00** | **100.00** | **0.00** | **99.98** | **0.00** | 99.99 | 98.80 | 17.98 |
| | CCU | **0.00** | 99.99 | 0.11 | 99.94 | **0.00** | 99.97 | **34.14** | 65.91 |
| | ROWL | **0.00** | 99.89 | 3.83 | 97.97 | **0.00** | 99.89 | 91.98 | 53.90 |
| | NTOM (ours) | **0.00** | **100.00** | 0.03 | 99.94 | **0.00** | **100.00** | 100.00 | 0.72 |
| | ATOM (ours) | **0.00** | **100.00** | **0.00** | 99.96 | **0.00** | **100.00** | 63.53 | **93.40** |
| **Uniform Noise** | MSP | 53.63 | 90.75 | 100.00 | 52.67 | 100.00 | 0.05 | 100.00 | 0.00 |
| | ODIN | 59.07 | 87.96 | 100.00 | 39.84 | 100.00 | 0.00 | 100.00 | 0.00 |
| | Mahalanobis | **0.00** | 99.73 | 99.99 | 56.02 | 100.00 | 20.33 | 100.00 | 0.00 |
| | SOFL | **0.00** | **100.00** | 0.35 | 99.74 | 39.53 | 91.58 | 100.00 | 0.01 |
| | OE | **0.00** | **100.00** | 5.83 | 98.88 | 0.09 | 99.98 | 99.97 | 0.16 |
| | ACET | **0.00** | **100.00** | 0.32 | 99.70 | **0.00** | 99.99 | 100.00 | 1.17 |
| | CCU | **0.00** | **100.00** | 4.04 | 99.19 | **0.00** | 99.97 | 99.82 | 0.18 |
| | ROWL | **0.00** | 99.89 | 17.44 | 91.17 | 0.19 | 99.80 | 100.00 | 49.89 |
| | NTOM (ours) | **0.00** | **100.00** | 0.11 | 99.86 | 0.22 | 99.87 | 100.00 | 0.01 |
| | ATOM (ours) | **0.00** | **100.00** | **0.00** | **99.92** | **0.00** | **100.00** | **99.79** | **74.58** |

Table 9: Comparison with competitive OOD detection methods. We use SVHN as in-distribution dataset and use DenseNet as network architecture for all methods. We evaluate the performance on all four types of OOD inputs: (1) natural OOD, (2) corruption attacked OOD, (3) $L_\infty$ attacked OOD, and (4) compositionally attacked OOD inputs. ↑ indicates larger value is better, and ↓ indicates lower value is better. All values are percentages. **Bold** numbers are superior results.

| $\mathcal{D}_{out}^{test}$ | Method | FPR (5% FNR) ↓ | AUROC ↑ | FPR (5% FNR) ↓ | AUROC ↑ | FPR (5% FNR) ↓ | AUROC ↑ | FPR (5% FNR) ↓ | AUROC ↑ |
|---|---|---|---|---|---|---|---|---|---|
| | | **Natural OOD** | | **Corruption OOD** | | $L_\infty$ **OOD** | | **Comp. OOD** | |
| **Gaussian Noise** | MSP | 100.00 | 77.69 | 100.00 | 73.29 | 100.00 | 69.73 | 100.00 | 52.56 |
| | ODIN | 97.30 | 89.00 | 100.00 | 81.48 | 100.00 | 71.12 | 100.00 | 34.99 |
| | Mahalanobis | **0.00** | **100.00** | **0.00** | **99.95** | **0.00** | **100.00** | 85.03 | 23.21 |
| | SOFL | **0.00** | 99.99 | 48.06 | 94.12 | 0.03 | 98.83 | 100.00 | 4.42 |
| | OE | **0.00** | 99.91 | 0.05 | 98.48 | **0.00** | 99.08 | 99.65 | 32.44 |
| | ACET | **0.00** | 99.94 | **0.00** | 99.74 | **0.00** | 99.87 | **0.04** | 99.40 |
| | CCU | **0.00** | 98.84 | 0.09 | 98.09 | 0.85 | 97.21 | 84.47 | 69.56 |
| | ROWL | 0.71 | 99.13 | 100.00 | 49.48 | 99.99 | 49.48 | 100.00 | 49.48 |
| | NTOM (ours) | **0.00** | 99.97 | 0.08 | 98.90 | **0.00** | 99.87 | 99.12 | 37.97 |
| | ATOM (ours) | **0.00** | 99.93 | **0.00** | 99.65 | **0.00** | 99.93 | 0.21 | **99.54** |
| **Uniform Noise** | MSP | 99.67 | 86.55 | 100.00 | 76.54 | 100.00 | 71.48 | 100.00 | 36.82 |
| | ODIN | 91.48 | 90.73 | 100.00 | 79.19 | 100.00 | 51.93 | 100.00 | 13.86 |
| | Mahalanobis | **0.00** | **100.00** | **0.00** | **99.94** | **0.00** | **100.00** | 99.84 | 22.25 |
| | SOFL | **0.00** | 99.75 | 95.82 | 86.22 | 100.00 | 69.89 | 100.00 | 4.39 |
| | OE | **0.00** | 99.89 | 0.02 | 98.62 | 33.05 | 92.41 | 99.98 | 29.62 |
| | ACET | **0.00** | 99.95 | **0.00** | 99.22 | **0.00** | 99.88 | **36.48** | **95.06** |
| | CCU | **0.00** | 98.50 | 0.87 | 97.78 | 94.68 | 86.27 | 99.70 | 48.18 |
| | ROWL | 97.14 | 50.91 | 100.00 | 49.48 | 100.00 | 49.48 | 100.00 | 49.48 |
| | NTOM (ours) | **0.00** | 99.97 | 1.42 | 98.08 | 28.41 | 95.94 | 100.00 | 2.88 |
| | ATOM (ours) | **0.00** | 99.93 | **0.00** | 99.07 | **0.00** | 99.93 | 91.81 | 45.48 |

Table 10: Comparison with competitive OOD detection methods. We use CIFAR-10 as in-distribution dataset and use DenseNet as network architecture for all methods. We evaluate the performance on all four types of OOD inputs: (1) natural OOD, (2) corruption attacked OOD, (3) $L_\infty$ attacked OOD, and (4) compositionally attacked OOD inputs. ↑ indicates larger value is better, and ↓ indicates lower value is better. All values are percentages.

| $\mathcal{D}_{out}^{test}$ | Method | FPR (5% FNR) ↓ | AUROC ↑ | FPR (5% FNR) ↓ | AUROC ↑ | FPR (5% FNR) ↓ | AUROC ↑ | FPR (5% FNR) ↓ | AUROC ↑ |
|---|---|---|---|---|---|---|---|---|---|
| | | **Natural OOD** | | **Corruption OOD** | | $L_\infty$ **OOD** | | **Comp. OOD** | |
| **Gaussian Noise** | MSP | 100.00 | 18.62 | 100.00 | 11.90 | 100.00 | 5.29 | 100.00 | 2.04 |
| | ODIN | 100.00 | 16.53 | 100.00 | 12.57 | 100.00 | 2.57 | 100.00 | 1.55 |
| | Mahalanobis | **0.00** | **100.00** | **0.00** | **99.82** | **0.00** | **100.00** | 99.18 | **81.79** |
| | SOFL | **0.00** | **100.00** | 100.00 | 70.93 | 4.56 | 98.30 | 100.00 | 0.17 |
| | OE | 3.69 | 97.53 | 100.00 | 64.71 | 67.00 | 92.45 | 100.00 | 1.60 |
| | ACET | **0.00** | 99.02 | 22.38 | 95.49 | 0.14 | 97.68 | 99.71 | 75.87 |
| | CCU | 100.00 | 50.78 | 100.00 | 47.92 | 100.00 | 37.06 | 100.00 | 35.25 |
| | ROWL | 0.54 | 99.42 | 100.00 | 49.69 | 99.37 | 50.01 | 100.00 | 49.69 |
| | NTOM (ours) | **0.00** | 99.95 | 30.60 | 95.57 | **0.00** | 99.92 | 100.00 | 30.10 |
| | ATOM (ours) | **0.00** | 99.98 | **0.00** | 99.06 | **0.00** | 99.98 | **86.43** | 52.84 |
| **Uniform Noise** | MSP | 99.99 | 37.86 | 100.00 | 16.59 | 100.00 | 2.45 | 100.00 | 1.67 |
| | ODIN | 100.00 | 31.77 | 100.00 | 20.56 | 100.00 | 0.48 | 100.00 | 0.40 |
| | Mahalanobis | **0.00** | **100.00** | **0.00** | **99.69** | **0.00** | **100.00** | 100.00 | 32.70 |
| | SOFL | **0.00** | 99.99 | 100.00 | 68.28 | 99.98 | 79.63 | 100.00 | 0.59 |
| | OE | 0.19 | 99.21 | 100.00 | 57.39 | 100.00 | 55.40 | 100.00 | 0.62 |
| | ACET | **0.00** | 99.43 | 64.94 | 93.48 | 95.10 | 91.27 | 98.47 | **57.60** |
| | CCU | 30.96 | 95.93 | 100.00 | 71.51 | 100.00 | 46.37 | 100.00 | 5.29 |
| | ROWL | 98.45 | 50.46 | 100.00 | 49.69 | 100.00 | 49.69 | 100.00 | 49.69 |
| | NTOM (ours) | **0.00** | 99.79 | 64.35 | 94.08 | 99.97 | 87.14 | 100.00 | 18.81 |
| | ATOM (ours) | **0.00** | 99.98 | 0.04 | 98.44 | **0.00** | 99.98 | **89.90** | 41.34 |

Table 11: Comparison with competitive OOD detection methods. We use CIFAR-100 as in-distribution dataset and use DenseNet as network architecture for all methods. We evaluate the performance on all four types of OOD inputs: (1) natural OOD, (2) corruption attacked OOD, (3) $L_\infty$ attacked OOD, and (4) compositionally attacked OOD inputs. ↑ indicates larger value is better, and ↓ indicates lower value is better. All values are percentages.

| $\mathcal{D}_{\text{in}}^{\text{test}}$ | Method | FNR | Pred. Acc. | End-to-end. Pred. Acc. |
|---|---|---|---|---|
| **SVHN** | MSP | 5.01 | 95.83 | 93.00 |
| | ODIN | 5.01 | 95.83 | 92.44 |
| | Mahalanobis | 5.01 | 95.83 | 91.50 |
| | SOFL | 5.01 | 96.45 | 92.81 |
| | OE | 5.01 | 95.93 | 93.11 |
| | ACET | 5.01 | 95.58 | 92.79 |
| | CCU | 5.01 | 95.85 | 92.95 |
| | ROWL | 0.22 | 95.23 | 95.23 |
| | NTOM (ours) | 5.01 | 95.75 | 91.99 |
| | ATOM (ours) | 5.01 | 96.09 | 91.95 |
| **CIFAR-10** | MSP | 5.01 | 94.39 | 91.76 |
| | ODIN | 5.01 | 94.39 | 91.00 |
| | Mahalanobis | 5.01 | 94.39 | 89.72 |
| | SOFL | 5.01 | 95.11 | 91.60 |
| | OE | 5.01 | 94.79 | 91.86 |
| | ACET | 5.01 | 91.48 | 88.61 |
| | CCU | 5.01 | 94.89 | 91.88 |
| | ROWL | 1.04 | 93.18 | 93.18 |
| | NTOM (ours) | 5.01 | 95.42 | 91.57 |
| | ATOM (ours) | 5.01 | 95.20 | 91.33 |
| **CIFAR-100** | MSP | 5.01 | 75.05 | 73.87 |
| | ODIN | 5.01 | 75.05 | 73.72 |
| | Mahalanobis | 5.01 | 75.05 | 71.12 |
| | SOFL | 5.01 | 74.37 | 72.62 |
| | OE | 5.01 | 75.28 | 73.74 |
| | ACET | 5.01 | 74.43 | 72.72 |
| | CCU | 5.01 | 76.04 | 74.60 |
| | ROWL | 0.62 | 72.51 | 72.51 |
| | NTOM (ours) | 5.01 | 74.88 | 72.47 |
| | ATOM (ours) | 5.01 | 75.06 | 72.72 |

Table 12: The performance of OOD detector and classifier on in-distribution test data. We use DenseNet for all methods. We use three metrics: FNR, Prediction Accuracy and End-to-end Prediction Accuracy. We pick the threshold for the OOD detectors such that 95% of in-distribution test data points are classified as in-distribution. Prediction Accuracy measures the accuracy of the classifier on in-distribution test data. End-to-end Prediction Accuracy measures the accuracy of the open world classification system (detector+classifier), where an example is classified correctly if and only if the detector treats it as in-distribution and the classifier predicts its label correctly.

| $\mathcal{D}_{\text{in}}^{\text{test}}$ | Method | FNR | Pred. Acc. | End-to-end. Pred. Acc. |
|---|---|---|---|---|
| **SVHN** | MSP | 5.01 | 95.61 | 92.91 |
| | ODIN | 5.01 | 95.61 | 92.49 |
| | Mahalanobis | 5.01 | 95.61 | 91.34 |
| | SOFL | 5.01 | 95.65 | 92.34 |
| | OE | 5.01 | 95.76 | 92.91 |
| | ACET | 5.01 | 95.96 | 93.13 |
| | CCU | 5.01 | 95.83 | 93.05 |
| | ROWL | 0.27 | 95.87 | 95.87 |
| | NTOM (ours) | 5.01 | 95.76 | 91.98 |
| | ATOM (ours) | 5.01 | 95.96 | 91.82 |
| **CIFAR-10** | MSP | 5.01 | 94.92 | 92.38 |
| | ODIN | 5.01 | 94.92 | 91.60 |
| | Mahalanobis | 5.01 | 94.92 | 90.27 |
| | SOFL | 5.01 | 95.98 | 92.45 |
| | OE | 5.01 | 95.51 | 92.35 |
| | ACET | 5.01 | 95.33 | 92.47 |
| | CCU | 5.01 | 95.63 | 92.46 |
| | ROWL | 1.21 | 93.91 | 93.91 |
| | NTOM (ours) | 5.01 | 95.89 | 91.97 |
| | ATOM (ours) | 5.01 | 95.89 | 92.20 |
| **CIFAR-100** | MSP | 5.01 | 76.61 | 75.46 |
| | ODIN | 5.01 | 76.61 | 75.16 |
| | Mahalanobis | 5.01 | 76.61 | 72.61 |
| | SOFL | 5.01 | 75.12 | 73.78 |
| | OE | 5.01 | 75.58 | 74.47 |
| | ACET | 5.01 | 75.61 | 74.60 |
| | CCU | 5.01 | 76.36 | 75.34 |
| | ROWL | 0.39 | 75.94 | 75.94 |
| | NTOM (ours) | 5.01 | 76.30 | 74.04 |
| | ATOM (ours) | 5.01 | 77.51 | 75.32 |

Table 13: The performance of OOD detector and classifier on in-distribution test data. We use WideResNet for all methods. We use three metrics: FNR, Prediction Accuracy and End-to-end Prediction Accuracy. We pick the threshold for the OOD detectors such that 95% of in-distribution test data points are classified as in-distribution. Prediction Accuracy measures the accuracy of the classifier on in-distribution test data. End-to-end Prediction Accuracy measures the accuracy of the open world classification system (detector+classifier), where an example is classified correctly if and only if the detector treats it as in-distribution and the classifier predicts its label correctly.

| $\mathcal{D}_{out}^{test}$ | Method | FPR (5% FNR) ↓ | AUROC ↑ | FPR (5% FNR) ↓ | AUROC ↑ | FPR (5% FNR) ↓ | AUROC ↑ | FPR (5% FNR) ↓ | AUROC ↑ |
|---|---|---|---|---|---|---|---|---|---|
| | | Natural OOD | | Corruption OOD | | $L_\infty$ OOD | | Comp. OOD | |
| LSUN-C | MSP | 33.90 | 94.01 | 99.59 | 68.76 | 99.94 | 0.82 | 100.00 | 0.11 |
| | ODIN | 28.40 | 93.77 | 97.37 | 59.61 | 99.90 | 0.38 | 100.00 | 0.01 |
| | Mahalanobis | 0.20 | 99.64 | 79.20 | 80.24 | 98.01 | 10.95 | 99.99 | 0.07 |
| | SOFL | **0.00** | **100.00** | **2.10** | **99.38** | 68.89 | 54.15 | 99.98 | 0.31 |
| | OE | **0.00** | **100.00** | 23.06 | 96.38 | 54.03 | 77.79 | 99.91 | 0.52 |
| | ACET | **0.00** | **100.00** | 20.48 | 96.75 | 0.09 | 99.97 | 99.85 | 4.76 |
| | CCU | **0.00** | **100.00** | 22.26 | 96.46 | 35.10 | 80.37 | 99.92 | 0.55 |
| | ROWL | 2.50 | 98.64 | 75.48 | 62.15 | 96.22 | 51.78 | 99.99 | **49.90** |
| | NTOM (ours) | **0.00** | **100.00** | 2.64 | 99.06 | 54.69 | 75.51 | 99.99 | 0.17 |
| | ATOM (ours) | **0.00** | **100.00** | 16.71 | 96.91 | **0.00** | **100.00** | **97.85** | 39.64 |
| LSUN-R | MSP | 44.84 | 92.75 | 99.93 | 66.82 | 100.00 | 1.22 | 100.00 | 0.11 |
| | ODIN | 36.84 | 92.91 | 98.53 | 62.91 | 100.00 | 0.25 | 100.00 | 0.01 |
| | Mahalanobis | 18.35 | 96.64 | 98.92 | 54.27 | 99.04 | 10.01 | 100.00 | 0.00 |
| | SOFL | **0.00** | **100.00** | 0.55 | 99.75 | 49.23 | 77.50 | 99.88 | 4.77 |
| | OE | 0.02 | **100.00** | 9.44 | 98.34 | 41.86 | 79.82 | 99.91 | 1.54 |
| | ACET | **0.00** | **100.00** | 5.36 | 98.87 | 2.60 | 99.35 | 99.96 | 7.00 |
| | CCU | **0.00** | **100.00** | 9.44 | 98.46 | 12.25 | 94.36 | 98.90 | 2.78 |
| | ROWL | 0.01 | 99.89 | 22.57 | 88.61 | 39.54 | 80.12 | 99.68 | 50.05 |
| | NTOM (ours) | **0.00** | **100.00** | **0.12** | **99.82** | 22.97 | 91.97 | 99.97 | 2.34 |
| | ATOM (ours) | **0.00** | **100.00** | 0.13 | 99.67 | **0.09** | **99.97** | 96.08 | **63.20** |
| iSUN | MSP | 40.78 | 93.57 | 99.85 | 68.88 | 99.99 | 1.37 | 100.00 | 0.14 |
| | ODIN | 31.98 | 93.97 | 97.70 | 64.77 | 99.99 | 0.29 | 100.00 | 0.01 |
| | Mahalanobis | 19.73 | 96.25 | 98.35 | 56.95 | 98.73 | 10.21 | 100.00 | 0.01 |
| | SOFL | **0.00** | **100.00** | 0.75 | 99.70 | 56.78 | 69.82 | 99.87 | 4.39 |
| | OE | 0.01 | **100.00** | 9.50 | 98.36 | 45.45 | 76.08 | 99.76 | 1.72 |
| | ACET | 0.01 | **100.00** | 6.17 | 98.84 | 4.90 | 98.52 | 99.92 | 6.96 |
| | CCU | 0.02 | **100.00** | 9.94 | 98.32 | 17.36 | 90.54 | 99.13 | 2.56 |
| | ROWL | 0.04 | 99.87 | 24.43 | 87.68 | 44.62 | 77.58 | 99.29 | 50.24 |
| | NTOM (ours) | **0.00** | **100.00** | 0.38 | **99.77** | 30.90 | 87.40 | 99.98 | 2.26 |
| | ATOM (ours) | **0.00** | **100.00** | 0.24 | 99.59 | **0.25** | **99.94** | 95.20 | **61.70** |
| Textures | MSP | 41.91 | 92.08 | 99.33 | 61.77 | 99.45 | 2.84 | 100.00 | 0.44 |
| | ODIN | 40.48 | 89.91 | 97.36 | 52.69 | 99.29 | 2.14 | 100.00 | 0.27 |
| | Mahalanobis | 24.34 | 94.15 | 85.94 | 62.00 | 88.83 | 17.73 | 99.38 | 1.27 |
| | SOFL | 0.34 | **99.91** | 11.83 | 97.47 | 89.36 | 23.54 | 99.20 | 2.42 |
| | OE | 2.73 | 99.50 | 43.05 | 92.51 | 84.36 | 26.88 | 98.32 | 2.72 |
| | ACET | 2.55 | 99.54 | 32.55 | 94.67 | 64.57 | 58.46 | 99.40 | 5.73 |
| | CCU | 2.22 | 99.57 | 41.86 | 92.46 | 78.09 | 31.94 | 98.58 | 2.31 |
| | ROWL | 7.85 | 95.96 | 71.17 | 64.30 | 89.73 | 55.02 | 99.77 | **50.01** |
| | NTOM (ours) | **0.27** | 99.90 | **9.18** | **97.97** | 81.17 | 35.06 | 98.76 | 2.74 |
| | ATOM (ours) | 0.41 | 99.83 | 11.31 | 97.66 | **28.81** | **90.97** | **91.61** | 49.35 |
| Places365 | MSP | 36.14 | 94.37 | 99.60 | 71.71 | 99.99 | 1.03 | 100.00 | 0.14 |
| | ODIN | 26.84 | 95.01 | 96.25 | 68.40 | 99.99 | 0.32 | 100.00 | 0.01 |
| | Mahalanobis | 34.83 | 93.95 | 98.14 | 55.70 | 99.64 | 1.97 | 100.00 | 0.00 |
| | SOFL | **0.00** | **99.99** | 3.46 | 99.09 | 94.58 | 26.96 | 99.96 | 1.98 |
| | OE | 0.37 | 99.91 | 27.78 | 95.86 | 94.97 | 26.54 | 99.99 | 0.47 |
| | ACET | 0.15 | 99.96 | 19.28 | 97.06 | 49.09 | 83.63 | 99.99 | 2.65 |
| | CCU | 0.27 | 99.95 | 29.12 | 95.65 | 83.26 | 39.88 | 100.00 | 0.59 |
| | ROWL | 0.97 | 99.41 | 67.26 | 66.26 | 96.67 | 51.56 | 100.00 | **49.89** |
| | NTOM (ours) | **0.00** | **99.99** | 2.39 | **99.21** | 87.35 | 46.68 | 99.99 | 0.48 |
| | ATOM (ours) | **0.00** | **99.99** | **2.31** | 98.59 | **5.60** | **98.71** | **98.74** | 38.95 |
| CIFAR-10 | MSP | 35.45 | 94.66 | 99.79 | 72.97 | 100.00 | 1.07 | 100.00 | 0.19 |
| | ODIN | 24.15 | 95.54 | 95.45 | 70.89 | 100.00 | 0.28 | 100.00 | 0.01 |
| | Mahalanobis | 39.35 | 92.80 | 98.28 | 55.52 | 99.70 | 2.47 | 100.00 | 0.01 |
| | SOFL | 0.01 | **99.99** | 3.99 | 99.03 | 93.00 | 28.70 | 99.95 | 2.60 |
| | OE | 0.48 | 99.89 | 27.79 | 95.92 | 95.49 | 26.02 | 99.98 | 0.63 |
| | ACET | 0.21 | 99.94 | 18.32 | 97.18 | 54.71 | 80.56 | 99.98 | 3.69 |
| | CCU | 0.47 | 99.91 | 32.40 | 95.34 | 86.98 | 36.35 | 99.98 | 0.79 |
| | ROWL | 0.84 | 99.47 | 69.28 | 65.25 | 96.68 | 51.55 | 100.00 | **49.89** |
| | NTOM (ours) | **0.00** | 99.98 | 2.53 | **99.17** | 84.60 | 47.74 | 100.00 | 0.99 |
| | ATOM (ours) | **0.00** | 99.98 | **2.14** | 98.72 | **7.39** | **98.42** | **98.51** | 44.28 |

Table 14: Comparison with competitive OOD detection methods. We use SVHN as in-distribution dataset and use DenseNet as network architecture for all methods. We evaluate the performance on all four types of OOD inputs: (1) natural OOD, (2) corruption attacked OOD, (3) $L_\infty$ attacked OOD, and (4) compositionally attacked OOD inputs. ↑ indicates larger value is better, and ↓ indicates lower value is better. All values are percentages. **Bold** numbers are superior results.

| $\mathcal{D}_{out}^{test}$ | Method | FPR (5% FNR) ↓ | AUROC ↑ | FPR (5% FNR) ↓ | AUROC ↑ | FPR (5% FNR) ↓ | AUROC ↑ | FPR (5% FNR) ↓ | AUROC ↑ |
|---|---|---|---|---|---|---|---|---|---|
| | | Natural OOD | | Corruption OOD | | $L_\infty$ OOD | | Comp. OOD | |
| LSUN-C | MSP | 27.34 | 96.30 | 100.00 | 71.57 | 100.00 | 13.75 | 100.00 | 13.68 |
| | ODIN | 1.89 | 99.50 | 98.94 | 71.86 | 100.00 | 0.06 | 100.00 | 0.00 |
| | Mahalanobis | 14.82 | 94.63 | 93.31 | 46.23 | 98.07 | 8.04 | 99.99 | 1.36 |
| | SOFL | 0.39 | 99.40 | 56.07 | 93.02 | 100.00 | 2.36 | 100.00 | 1.96 |
| | OE | 0.97 | 99.52 | 40.85 | 93.99 | 99.99 | 0.36 | 100.00 | 0.19 |
| | ACET | 1.76 | 99.42 | 31.46 | 95.40 | 45.54 | 90.57 | 88.09 | 70.84 |
| | CCU | 0.62 | **99.65** | 33.44 | 94.56 | 99.97 | 0.31 | 100.00 | 0.05 |
| | ROWL | 10.92 | 94.02 | 91.67 | 53.64 | 100.00 | 49.48 | 100.00 | 49.48 |
| | NTOM (ours) | 0.27 | 99.59 | 20.32 | 96.31 | 99.95 | 3.12 | 100.00 | 0.36 |
| | ATOM (ours) | **0.25** | 99.53 | **16.38** | **96.75** | **0.38** | **99.49** | **17.27** | **96.49** |
| LSUN-R | MSP | 43.89 | 93.93 | 100.00 | 64.26 | 100.00 | 13.74 | 100.00 | 13.66 |
| | ODIN | 3.29 | 99.20 | 98.96 | 63.83 | 100.00 | 0.14 | 100.00 | 0.00 |
| | Mahalanobis | 7.43 | 97.88 | 98.71 | 39.24 | 94.59 | 16.62 | 100.00 | 0.45 |
| | SOFL | 1.67 | 99.29 | 56.41 | 90.62 | 100.00 | 0.55 | 100.00 | 0.43 |
| | OE | 0.99 | 99.43 | 51.55 | 92.25 | 99.99 | 0.15 | 100.00 | 0.02 |
| | ACET | 3.87 | 99.10 | 77.90 | 87.37 | 70.50 | **84.01** | 99.87 | 48.53 |
| | CCU | 1.53 | 99.28 | 57.03 | 90.53 | 100.00 | 0.04 | 100.00 | 0.10 |
| | ROWL | 36.03 | 81.47 | 98.72 | 50.12 | 100.00 | 49.48 | 100.00 | 49.48 |
| | NTOM (ours) | **0.38** | **99.57** | 17.84 | **96.76** | 100.00 | 0.20 | 100.00 | 0.27 |
| | ATOM (ours) | 0.41 | 99.33 | **16.77** | 96.63 | 52.76 | 71.84 | 50.33 | 75.18 |
| iSUN | MSP | 46.18 | 93.58 | 100.00 | 62.76 | 99.99 | 13.95 | 100.00 | 13.67 |
| | ODIN | 4.45 | 99.00 | 98.90 | 62.14 | 100.00 | 0.33 | 100.00 | 0.01 |
| | Mahalanobis | 8.58 | 98.00 | 98.10 | 42.96 | 89.64 | 22.78 | 100.00 | 0.74 |
| | SOFL | 2.24 | 99.22 | 53.99 | 90.97 | 100.00 | 0.51 | 100.00 | 0.50 |
| | OE | 1.14 | 99.40 | 48.25 | 92.46 | 99.97 | 0.13 | 100.00 | 0.02 |
| | ACET | 6.16 | 98.59 | 75.36 | 87.00 | 78.59 | **79.84** | 99.63 | 47.11 |
| | CCU | 1.74 | 99.27 | 52.44 | 91.10 | 100.00 | 0.05 | 100.00 | 0.08 |
| | ROWL | 35.44 | 81.76 | 97.02 | 50.97 | 100.00 | 49.48 | 100.00 | 49.48 |
| | NTOM (ours) | **0.63** | **99.59** | 16.25 | **96.90** | 99.99 | 0.32 | 100.00 | 0.38 |
| | ATOM (ours) | 0.66 | 99.34 | **15.10** | 96.79 | 52.48 | 68.44 | **51.94** | 72.08 |
| Textures | MSP | 64.66 | 87.64 | 100.00 | 51.85 | 100.00 | 14.14 | 100.00 | 13.72 |
| | ODIN | 52.45 | 84.81 | 99.56 | 38.20 | 99.95 | 0.56 | 100.00 | 0.07 |
| | Mahalanobis | 25.39 | 92.20 | 71.42 | 61.60 | 88.85 | 17.24 | 99.27 | 3.60 |
| | SOFL | 3.78 | 99.04 | 56.81 | 89.38 | 99.88 | 2.06 | 99.98 | 1.36 |
| | OE | 6.24 | 98.43 | 53.32 | 88.83 | 99.68 | 1.40 | 99.95 | 0.68 |
| | ACET | 11.74 | 97.96 | 54.41 | 90.52 | 64.49 | 77.66 | 94.26 | 55.24 |
| | CCU | 5.83 | 98.45 | 54.61 | 86.33 | 99.49 | 1.68 | 99.84 | 0.99 |
| | ROWL | 19.33 | 89.82 | 82.87 | 58.04 | 99.89 | 49.53 | 99.98 | 49.49 |
| | NTOM (ours) | 2.27 | 99.42 | 24.66 | 95.09 | 99.49 | 3.63 | 99.95 | 1.67 |
| | ATOM (ours) | **1.81** | **99.47** | **20.05** | **95.92** | **10.05** | **96.47** | **29.01** | **91.40** |
| Places365 | MSP | 62.03 | 88.29 | 100.00 | 57.68 | 100.00 | 13.66 | 100.00 | 13.66 |
| | ODIN | 43.84 | 90.45 | 99.89 | 52.81 | 100.00 | 0.01 | 100.00 | 0.00 |
| | Mahalanobis | 85.77 | 65.76 | 99.47 | 22.75 | 99.79 | 1.93 | 100.00 | 0.66 |
| | SOFL | 7.73 | 97.81 | 61.66 | 88.07 | 100.00 | 0.47 | 100.00 | 0.27 |
| | OE | 11.08 | 97.00 | 67.91 | 87.39 | 100.00 | 0.03 | 100.00 | 0.01 |
| | ACET | 18.63 | 95.97 | 79.42 | 85.00 | 93.09 | 66.83 | 99.61 | 46.19 |
| | CCU | 8.49 | 97.63 | 67.68 | 85.75 | 100.00 | 0.03 | 100.00 | 0.03 |
| | ROWL | 43.76 | 77.60 | 97.67 | 50.64 | 100.00 | 49.48 | 100.00 | 49.48 |
| | NTOM (ours) | 6.63 | **97.94** | 37.01 | 92.75 | 100.00 | 0.06 | 100.00 | 0.04 |
| | ATOM (ours) | **6.30** | 97.92 | **31.44** | **93.41** | **6.96** | **97.78** | **32.99** | **92.93** |
| SVHN | MSP | 59.15 | 90.99 | 100.00 | 41.97 | 100.00 | 13.67 | 100.00 | 13.66 |
| | ODIN | 23.96 | 95.00 | 100.00 | 19.78 | 100.00 | 0.00 | 100.00 | 0.00 |
| | Mahalanobis | 19.73 | 93.33 | 90.52 | 50.89 | 99.48 | 8.18 | 100.00 | 2.67 |
| | SOFL | 0.85 | 99.47 | 87.50 | 79.85 | 100.00 | 0.13 | 100.00 | 0.07 |
| | OE | 1.55 | 99.16 | 75.62 | 89.02 | 100.00 | 0.01 | 100.00 | 0.02 |
| | ACET | 31.50 | 94.99 | 83.04 | 85.25 | 94.49 | 69.42 | 99.81 | 54.36 |
| | CCU | 2.14 | 99.25 | 75.34 | 88.01 | 100.00 | 0.00 | 100.00 | 0.00 |
| | ROWL | 4.73 | 97.12 | 98.09 | 50.43 | 100.00 | 49.48 | 100.00 | 49.48 |
| | NTOM (ours) | 1.06 | 99.59 | 67.40 | 90.23 | 100.00 | 0.01 | 100.00 | 0.00 |
| | ATOM (ours) | **0.69** | **99.63** | **51.84** | **92.22** | **0.69** | **99.63** | **51.81** | **92.18** |

Table 15: Comparison with competitive OOD detection methods. We use CIFAR-10 as in-distribution dataset and use DenseNet as network architecture for all methods. We evaluate the performance on all four types of OOD inputs: (1) natural OOD, (2) corruption attacked OOD, (3) $L_\infty$ attacked OOD, and (4) compositionally attacked OOD inputs. ↑ indicates larger value is better, and ↓ indicates lower value is better. All values are percentages. **Bold** numbers are superior results.

| $\mathcal{D}_{out}^{test}$ | Method | FPR (5% FNR) ↓ | AUROC ↑ | FPR (5% FNR) ↓ | AUROC ↑ | FPR (5% FNR) ↓ | AUROC ↑ | FPR (5% FNR) ↓ | AUROC ↑ |
|---|---|---|---|---|---|---|---|---|---|
| | | Natural OOD | | Corruption OOD | | $L_\infty$ OOD | | Comp. OOD | |
| **LSUN-C** | MSP | 62.03 | 84.78 | 100.00 | 32.47 | 100.00 | 2.52 | 100.00 | 2.31 |
| | ODIN | 15.47 | 97.34 | 100.00 | 42.25 | 100.00 | 0.20 | 100.00 | 0.01 |
| | Mahalanobis | 47.44 | 93.47 | 98.80 | 58.23 | 98.94 | 15.07 | 99.97 | 3.01 |
| | SOFL | 17.38 | 96.66 | 100.00 | 51.59 | 100.00 | 1.10 | 100.00 | 0.58 |
| | OE | 14.75 | 97.33 | 99.91 | 54.29 | 100.00 | 1.61 | 100.00 | 0.64 |
| | ACET | 14.60 | 97.41 | 98.65 | 67.81 | 23.07 | 95.35 | 98.88 | 56.02 |
| | CCU | **12.03** | **97.84** | 99.60 | 61.24 | 100.00 | 0.86 | 100.00 | 0.50 |
| | ROWL | 88.67 | 55.35 | 100.00 | 49.69 | 100.00 | 49.69 | 100.00 | 49.69 |
| | NTOM (ours) | 20.61 | 96.58 | 95.91 | 83.28 | 100.00 | 1.31 | 100.00 | 0.25 |
| | ATOM (ours) | 21.40 | 96.31 | **79.72** | **87.67** | **21.50** | **96.26** | **79.98** | **87.11** |
| **LSUN-R** | MSP | 77.48 | 76.40 | 100.00 | 32.21 | 100.00 | 1.94 | 100.00 | 1.78 |
| | ODIN | 34.81 | 93.37 | 100.00 | 45.41 | 100.00 | 0.22 | 100.00 | 0.00 |
| | Mahalanobis | **14.87** | 97.06 | 99.89 | 31.14 | 94.81 | 24.18 | 100.00 | 0.37 |
| | SOFL | 50.27 | 90.28 | 99.88 | 50.12 | 100.00 | 0.11 | 100.00 | 0.20 |
| | OE | 56.25 | 84.35 | 99.96 | 41.17 | 100.00 | 0.70 | 100.00 | 0.52 |
| | ACET | 56.35 | 88.17 | 99.50 | 51.85 | 98.67 | 18.64 | 99.74 | 22.45 |
| | CCU | 38.44 | 91.83 | 99.94 | 50.62 | 100.00 | 0.59 | 100.00 | 0.47 |
| | ROWL | 88.25 | 55.57 | 100.00 | 49.69 | 100.00 | 49.69 | 100.00 | 49.69 |
| | NTOM (ours) | 37.92 | 94.31 | 99.11 | 65.00 | 100.00 | 0.36 | 100.00 | 0.06 |
| | ATOM (ours) | 17.93 | 96.94 | **95.72** | **71.51** | **31.39** | **87.38** | **95.87** | **66.15** |
| **iSUN** | MSP | 78.87 | 75.69 | 100.00 | 31.83 | 100.00 | 2.08 | 100.00 | 1.82 |
| | ODIN | 38.92 | 92.15 | 100.00 | 43.43 | 100.00 | 0.31 | 100.00 | 0.00 |
| | Mahalanobis | **16.46** | **96.75** | 99.76 | 33.38 | 89.71 | 28.16 | 100.00 | 0.38 |
| | SOFL | 53.51 | 89.27 | 99.96 | 48.75 | 100.00 | 0.19 | 100.00 | 0.21 |
| | OE | 61.59 | 81.51 | 99.99 | 39.94 | 100.00 | 0.81 | 100.00 | 0.54 |
| | ACET | 60.49 | 86.80 | 99.70 | 49.61 | 98.54 | 19.92 | 99.94 | 23.51 |
| | CCU | 40.97 | 90.89 | 99.98 | 49.21 | 100.00 | 0.79 | 100.00 | 0.44 |
| | ROWL | 90.42 | 54.48 | 100.00 | 49.69 | 100.00 | 49.69 | 100.00 | 49.69 |
| | NTOM (ours) | 43.66 | 93.24 | 99.51 | 61.40 | 100.00 | 0.41 | 100.00 | 0.02 |
| | ATOM (ours) | 20.09 | 96.62 | **97.05** | **68.31** | **35.70** | **85.87** | **97.13** | **63.16** |
| **Textures** | MSP | 85.57 | 70.08 | 100.00 | 25.93 | 100.00 | 2.63 | 100.00 | 2.28 |
| | ODIN | 83.58 | 70.71 | 100.00 | 27.30 | 100.00 | 0.11 | 100.00 | 0.01 |
| | Mahalanobis | **34.59** | 89.82 | **78.35** | 62.36 | 89.27 | 20.33 | 99.18 | 5.43 |
| | SOFL | 57.00 | 87.35 | 99.73 | 44.11 | 99.98 | 0.59 | 100.00 | 0.34 |
| | OE | 59.86 | 86.17 | 99.88 | 42.95 | 99.98 | 1.45 | 100.00 | 0.73 |
| | ACET | 62.02 | 86.26 | 99.70 | 49.02 | 83.17 | 63.61 | 99.86 | 40.58 |
| | CCU | 60.80 | 86.34 | 99.88 | 44.90 | 100.00 | 1.39 | 100.00 | 0.62 |
| | ROWL | 97.00 | 51.19 | 100.00 | 49.69 | 100.00 | 49.69 | 100.00 | 49.69 |
| | NTOM (ours) | 38.21 | **91.06** | 95.34 | 66.86 | 99.82 | 2.43 | 100.00 | 0.60 |
| | ATOM (ours) | 40.11 | 90.28 | 90.21 | **69.99** | **47.27** | **82.52** | **91.45** | **65.91** |
| **Places365** | MSP | 83.65 | 73.71 | 100.00 | 32.13 | 100.00 | 1.83 | 100.00 | 1.91 |
| | ODIN | 79.19 | 76.48 | 100.00 | 39.16 | 100.00 | 0.00 | 100.00 | 0.00 |
| | Mahalanobis | 94.64 | 59.52 | 99.80 | 19.43 | 99.96 | 1.13 | 100.00 | 0.50 |
| | SOFL | 60.49 | **87.57** | 100.00 | 40.25 | 100.00 | 0.06 | 100.00 | 0.15 |
| | OE | 58.37 | 86.39 | 100.00 | 50.83 | 100.00 | 0.57 | 100.00 | 0.54 |
| | ACET | 56.26 | 86.75 | 99.66 | 57.36 | 83.60 | 72.35 | 99.83 | 44.27 |
| | CCU | **55.23** | 87.21 | 100.00 | 44.03 | 100.00 | 0.46 | 100.00 | 0.43 |
| | ROWL | 96.86 | 51.26 | 100.00 | 49.69 | 100.00 | 49.69 | 100.00 | 49.69 |
| | NTOM (ours) | 56.54 | 84.27 | 99.33 | 57.32 | 100.00 | 0.01 | 100.00 | 0.02 |
| | ATOM (ours) | 56.52 | 84.53 | **96.65** | **63.43** | **58.68** | **82.50** | **96.68** | **62.36** |
| **SVHN** | MSP | 80.71 | 76.00 | 100.00 | 25.69 | 100.00 | 2.47 | 100.00 | 2.29 |
| | ODIN | 88.66 | 71.65 | 100.00 | 24.17 | 100.00 | 0.00 | 100.00 | 0.00 |
| | Mahalanobis | 47.78 | 90.54 | **98.93** | 53.25 | 99.96 | 6.34 | 100.00 | 2.78 |
| | SOFL | **21.50** | 96.15 | 100.00 | 36.59 | 100.00 | 0.07 | 100.00 | 0.16 |
| | OE | 44.47 | 92.58 | 100.00 | 40.86 | 100.00 | 0.53 | 100.00 | 0.57 |
| | ACET | 55.86 | 90.36 | 100.00 | 49.51 | 70.55 | 86.86 | 99.99 | 44.93 |
| | CCU | 50.79 | 91.59 | 100.00 | 40.01 | 100.00 | 0.41 | 100.00 | 0.40 |
| | ROWL | 98.88 | 50.25 | 100.00 | 49.69 | 100.00 | 49.69 | 100.00 | 49.69 |
| | NTOM (ours) | 24.67 | **96.20** | 99.83 | 60.37 | 100.00 | 0.01 | 100.00 | 0.00 |
| | ATOM (ours) | 37.78 | 93.68 | 99.54 | **70.86** | **37.78** | **93.68** | **99.55** | **70.23** |

Table 16: Comparison with competitive OOD detection methods. We use CIFAR-100 as in-distribution dataset and use DenseNet as network architecture for all methods. We evaluate the performance on all four types of OOD inputs: (1) natural OOD, (2) corruption attacked OOD, (3) $L_\infty$ attacked OOD, and (4) compositionally attacked OOD inputs. ↑ indicates larger value is better, and ↓ indicates lower value is better. All values are percentages. **Bold** numbers are superior results.

