# OpenReview forum: "Informative Outlier Matters: Robustifying Out-of-distribution Detection Using Outlier Mining"
_ICLR.cc/2021/Conference — Reject_

### Official Review · AnonReviewer3 · 2020-10-27
**Simple approach for improving OOD detection.**

**Rating:** 3
**Confidence:** 4

**Review:**

In this paper the authors propose a method for training a classifier to be more effective at OOD (out of distribution) detection. Many OOD detection methods work by utilizing an auxiliary dataset as examples of OOD-ness. This is the approach taken in this paper and OOD is trained as being a k+1 classification class.  When training the OOD class the proposed method allows for adversarial perturbation of the OOD examples to help improve training. This is a pretty common technique in deep learning, see for example "Deep Robust One Class Classification."  Finally the main novelty of the method proposed by the authors is to sort a collection of OOD examples and sort according to the OOD score of the current model and use the "qNth" to be presented as OOD examples during the next epoch during training. The authors term this "Informative Outlier Mining." The authors demonstrate that this method works well experimentally.

The authors present some theoretical justification for their approach via a very toyish analysis of a OOD detection with a Gaussian data model.

While the proposed method seems to work reasonably well I find the novelty of this method to be too low to be interesting being more of a small trick rather than deserving of being the topic of an entire paper. I do think, however that this idea, in combination with some other OOD tricks could be an interesting paper.

---

> ### Author Response · Authors · 2020-11-12
> **We believe the extent of our thoroughness in experimental evaluation, together with our theoretical formalization not only deserves to be the topic of an entire paper, but also opens up interesting and promising future directions.**
>
> We thank the reviewer and would like to clarify some concerns about the novelty and significance of our paper.
>
> We believe our proposed method “informative outlier mining” is more than just a “small trick”, as seen in our strong evidence, both empirically and theoretically. While previous works [1][2][3] have utilized auxiliary outlier data for OOD detection, to the best of our knowledge, we are the first to explore the novel connection between hard example mining and out-of-distribution detection. This may appear simple in retrospect, but establishing this connection both theoretically and empirically is a non-trivial scientific contribution to the field. In fact, the idea of hard example mining has **fundamentally** transformed various learning problem domains such as object recognition [4].
>
> Moreover, we believe the extent of our thoroughness in experimental evaluation, together with our theoretical formalization **not only deserves to be the topic of an entire paper, but also opens up interesting and promising future directions to consider using auxiliary outlier data more effectively**. Below we provide a few hard facts about the significance of this work:
>
> 1. empirically:
>
> We show that informative outlier mining leads to substantial improvement over existing methods (better by a large margin on adversarial OOD evaluation tasks, with up to 53.9% FPR reduction over current SOTA ACET[3]). We emphasize that we have provided extensive empirical evaluation, including those considered in previous work and also new ones introduced by us. In the evaluation, our method **achieves state-of-the-art results, significantly outperforming 8 competitive OOD-detection methods across 4 evaluation tasks and 6 OOD datasets**. **The extent of thoroughness in evaluation is beyond any prior work we are aware of**. To this end, we believe the simplicity of a method should not be the sole judgment of significance. In a similar analogy, Dropout [5] is nowadays one of the most successful regularization techniques in deep learning, yet the merit isn’t discounted because of its simplicity. Perhaps quite the opposite, simple methods are desirable in practice as they are easy to use and adopt.
>
> 2. theoretically:
>
> Our **theoretical formalization is new, significant, and directly connects to our strong empirical results in Section 5** (see Prop. 1 in Section 3).  As R4 recognizes, quote, our paper _"provides theoretical analysis formalizing the intuition of mining hard outliers for improving the robustness of OOD detection_.” The simplicity of the Gaussian model helps to illustrate the intuition why informative outlier mining is important. Our analysis can be generalized to a more general setting at the cost of more calculations and more complicated bounds. For example, we can replace each Gaussian with a mixture of sub-Gaussians, which is believed to be a good model for simple practical datasets like MNIST. However, the complication there will hinder the intuition rather than help the illustration.  Besides the analysis in the Gaussian data model, we also have a general error bound (see Appendix A.1). The Gaussian analytical framework has also been widely used in robustness literature (e.g., see Carmon et al. 2019).
>
> Lastly, the reviewer suggested combining outlier mining with other OOD techniques. We agree this would be interesting to explore, however, we believe our central contributions in the current scope are already **substantial and significant** (see comments above).
>
> [1] D. Hendrycks, M. Mazeika, and T. G Dietterich. Deep anomaly detection with outlier exposure. ICLR 2019
>
> [2] S. Mohseni, M.Pitale, JBS Yadawa, and Z. Wang. Self-supervised learning for generalizable out-of-distribution detection. AAAI 2020
>
> [3] M. Hein, M. Andriushchenko, and J. Bitterwolf. Why relu networks yield high confidence predictions far away from the training data and how to mitigate the problem. CVPR 2019.
>
> [4] Sung, Kah-Kay. "Learning and example selection for object and pattern detection." (1996).
>
> [5] Srivastava, N., Hinton, G., Krizhevsky, A., Sutskever, I. and Salakhutdinov, R., 2014. Dropout: a simple way to prevent neural networks from overfitting. The journal of machine learning research, 15(1), pp.1929-1958.

---

> ### Author Response · Authors · 2020-11-22
> **Follow up**
>
> Dear reviewer,
>
> We thank you again for the constructive feedback. We believe your comments raised have been addressed in our response and updated draft. We'd like to kindly follow up and clarify any remainder confusion. Your feedback has been very important and valuable for us to improve the work!
>
> Thank you,
>
> Authors of Paper1050

---

### Official Review · AnonReviewer1 · 2020-10-28
**Needs better motivation and identification of novelty**

**Rating:** 4
**Confidence:** 4

**Review:**

The paper attempts to formulate the task of robust OOD detection by expanding the usual definition of OODs, and then presents a theoretical analysis in the simple setting of a Gaussian data model by porting results from adversarial defense. An experimental comparison is also presented.

Pro:

+ The paper addresses a very important problem which is growing in importance. A number of approaches have been recently proposed to address this problem that attempt statistical, explanation-based, and relational/semantic techniques. The paper takes a simple approach of class augmentation and then picking hard OODs to solve this problem.

Cons:

- Why is it surprising that the "majority of auxiliary OOD examples may not provide useful information to improve the decision boundary of OOD detector"? This seems to be rather obvious both for the toy case of Gaussian model or in general. It is well known that hard examples help train. The triplet loss function and other methods have been used for years based on this insight. The paper appears to present this obvious observation as the central finding of the paper.  For e.g. in the context of embedding, see how finding hard examples has been found to be useful in https://openaccess.thecvf.com/content_ICCV_2017/papers/Yuan_Hard-Aware_Deeply_Cascaded_ICCV_2017_paper.pdf

- The paper appears to be mixing a lot of different problems to formulate a very generic notion of OOD. Typically, OODs have been used for out of distribution datasets (SVNH for CIFAR10), new classes on which model was not trained (leave some classes out of CIFAR10), or transforms such as rotations or other transformations of inputs. The paper mixes adversarial examples to it and then considers even further compositions. While there is significant effort to break OOD by building a useful taxonomy of it to refine the sources of uncertainty and address these in a principled way, the paper creates a monolithic problem of detecting when inputs don't belong to the training distribution for any reason. This creates a rather ill-defined mathematical problem because modeling the exact distribution of the training data is quite infeasible. The theoretical treatment of the experiments in the paper fail to convince the reviewer that there is an advantage in merging these separate problems into a single problem.

- The theoretical treatment are directly borrowed from http://papers.nips.cc/paper/9298-unlabeled-data-improves-adversarial-robustness.pdf where the treatment was for adversarial examples - since, the paper's definition of OOD includes these, the previous results are directly applicable. It would help to make this obvious in the main text instead of having this reference in the appendix. The connection is very strong. Also, this reviewer currently does not see the challenge in lifting the result from the reference to the theoretical analysis in this paper (please see question below).

- The experimental evaluation is limited and does not meet the usual set of experiments reported for OOD detection which makes it difficult to understand the empirical benefit of the presented approach.

- While there are other methods that also use "exposure to outlier" in OOD detection, it is debatable such an approach would be useful in practice, particularly, if one also included adversarial examples. But given past literature on this topic, the reviewer is less concerned about this aspect.

In summary, the idea of using hard examples is not new. The definition of OOD to include adversarial examples is not well motivated. The theoretical results are borrowed without significant extension from literature on adversarial defenses. The experimental results do not meet the expectation of a paper on this topic to judge its empirical value against existing approaches.

Questions for the author:

- Why is it helpful to merge different problems of adversarial example detection, detection of data away from training distribution due to new classes, detection of corrupted inputs into a single problem? Does a uniform treatment yield some new insight or enable some particular approach? Does it beat the state of the art in any of these sub-problems?

- Can you elaborate on the nature of "natural OODs" in the experiments so that it is easier to understand comparison with ODIN, Mahalanobis, etc methods? They perform differently when OODs are because of new datasets or because of held-out classes. This "natural OOD" is better split into at least these two different types before evaluating the methods.

- How is the theoretical analysis presented here not a trivial application (from adversarial setting to a superset definition of OOD) of results in http://papers.nips.cc/paper/9298-unlabeled-data-improves-adversarial-robustness.pdf  and https://arxiv.org/pdf/1804.11285.pdf ? Could you help identify the challenge and the novelty in this extension?

The paper currently appears to be building on the premise that if we think of adversarial examples as a subclass of OODs, theoretical results from adversarial examples are applicable to OODs. A simple supervised approach using hard examples is then presented to solve this problem. Its empirical evaluation appears to behind the state-of-the-art in OOD detection as well as adversarial defense, but mixing both does not make this any stronger a paper. In its current form, the paper appears to need better motivation and identification of novelty.

After discussion with authors
------------------------------------------

Some of the concerns of the reviewer have been addressed and the reviewer is raising the score to reflect it. The paper still has some major concerns preventing the reviewer from recommending acceptance of the paper:

- Similarity of the theoretical analysis to https://proceedings.neurips.cc/paper/2019/file/32e0bd1497aa43e02a42f47d9d6515ad-Paper.pdf  and https://arxiv.org/pdf/1804.11285.pdf .

- Hard negative mining is pretty standard in many learning domains.

Both of these issues can be addressed by a better review of the related work and more accurate identification of the novelty in the paper.

Given the limited theoretical novelty, a more robust empirical evaluation establishing the value of the extreme outliers against state of the art approaches would have been useful. Also, if the authors want to investigate theoretical results on OODs, one challenge is the lack of a formal definition of OODs.

The paper is a good work in progress but not yet ready for publication. The reviewer is hopeful that the above suggestions will make the paper stronger.

---

> ### Author Response · Authors · 2020-11-11
> **Adversarial OOD is different from the typical adversarial example**
>
> Thanks for the review. R1 has some misunderstandings, and we want to respond right away.
>
> First of all, the idea of using hard negative mining **for OOD detection** is novel since it hasn’t been considered for the OOD detection problem before. Our key contribution is to point out its great importance of this technique for OOD detection: As demonstrated in our paper, informative outlier mining could significantly improve the generalization and robustness of OOD detection and advance state-of-the-art (SOTA). Furthermore, we have also made contributions to the technical details of this high-level idea, e.g., specifying how to select the outliers, providing analysis to justify our selection scheme, and thoroughly evaluating the designed method.
>
> Second, there is some misunderstanding about adversarial OOD. An adversarial OOD example is created by adding some imperceivable adversarial perturbation to an OOD example so that the detector fails to realize it is OOD. A concrete example is given in case (b) in Figure 1 of our paper. Given a mailbox image (which is an OOD example for a traffic sign classifier), the adversary can perturb it so that the OOD detector believes the perturbed image is in-distribution and passes it to the traffic sign classifier. This can lead to severe security consequences and thus motivate our study. In fact, the concept of adversarial OOD inputs have been explored in the literature, see [1][2][3]. In contrast, adversarial examples of traffic sign images are created to fool the traffic sign classifier, but not the OOD detector. These points have been discussed in our paper. Please refer to the second paragraph and Figure 1 in Section 1, and the problem definition in Section 2. We will further clarify this in our writing.
>
> Third, our method achieves SOTA results on both the traditional OOD detection evaluation tasks, and the robust OOD detection evaluation tasks introduced in our paper. The evaluation on “natural OODs” is exactly **the same task** that ODIN and Mahalanobis techniques evaluate (e.g. using CIFAR-10 as in-distribution and SVHN as OOD). We can confirm that our method achieves better results than previous approaches under their evaluation tasks. Furthermore, we introduce new OOD detection evaluation tasks like corrupted OOD inputs, which haven’t been considered previously. Finally, we want to highlight that we have performed extensive experimental evaluations. We compare our method to 8 competitive OOD detection methods and also perform ablation studies for various aspects. In summary, our experimental evaluation indeed meets the usual set of experiments reported for OOD detection, and furthermore provides additional experiments.
>
> Last but not least, our analysis is not directly adapted from [4]. First, the major part of our analysis is novel. For example, the proof of Prop. 3 is completely new. We did adopt the same assumptions on the data (the Gaussian model) and use some results from [4] in our proofs (more precisely, we used their expressions for robust error, and used their lower bound to prove Prop. 1). But our analysis is not an application of their analysis. Second, our setting is not the same as theirs. For example, the test OOD distribution $Q_X$ is not known in training time but is only known to be from a family $\mathcal{Q}$, which requires extra analysis. See the definition in Section 3.
>
> [1] Vikash Sehwag, Arjun Nitin Bhagoji, Liwei Song, Chawin Sitawarin, Daniel Cullina, Mung Chiang, and Prateek Mittal. Analyzing the robustness of open-world machine learning. In Proceedings of the 12th ACM Workshop on Artificial Intelligence and Security, pp. 105–116, 2019.
>
> [2] Matthias Hein, Maksym Andriushchenko, and Julian Bitterwolf. Why relu networks yield high confidence predictions far away from the training data and how to mitigate the problem. In Proceedings of the IEEE Conference on Computer Vision and Pattern Recognition, pp. 41–50, 2019.
>
> [3] Julian Bitterwolf, Alexander Meinke, and Matthias Hein. Provable worst case guarantees for the detection of out-of-distribution data. arXiv preprint arXiv:2007.08473, 2020.
>
> [4] Yair Carmon, Aditi Raghunathan, Ludwig Schmidt, John C Duchi, and Percy S Liang. Unlabeled data improves adversarial robustness. In Advances in Neural Information Processing Systems, pp. 11190–11201, 2019.

---

> > ### Comment · AnonReviewer1 · 2020-11-15
> > **Concerns remain .. in particular, on the theoretical derivation**
> >
> > The reviewer thanks the authors for clarification. Some of the concerns of the reviewer has been addressed, and she/he will raise the score to encourage this work. But the paper still does not meet the expected novelty threshold of being recommended for acceptance.
> >
> > The primary concerns are:
> >
> > - Theoretical analysis is very similar to those previously presented in  https://proceedings.neurips.cc/paper/2019/file/32e0bd1497aa43e02a42f47d9d6515ad-Paper.pdf . Also see Theorem 6 in https://arxiv.org/pdf/1804.11285.pdf .  Discussion of these are not in the paper and the response is unable to convince the reviewer that the new delta is significant.
> >
> > - Hard negative mining is pretty standard in many learning domains. A better review of related work would have benefited the paper.
> >
> > - Given the limited theoretical novelty, a more robust empirical evaluation establishing the value of the extreme outliers against state of the art approaches would have been useful.
> >
> > - Also, if the authors want to investigate theoretical results on OODs, one challenge is the lack of a formal definition of OODs.
> >
> > The paper is a good work in progress but not yet ready for publication. The reviewer is hopeful that the above suggestions will make the paper stronger. In particular, the reviewer will encourage a better discussion of the related work and identification of novelty.

---

> > > ### Author Response · Authors · 2020-11-15
> > > **We respectfully disagree due to some perceived misunderstandings about our paper, which we tried to clear in the responses**
> > >
> > > 1. The reviewer challenged the novelty of our theory analysis, and argued that it is too similar to [paper](/https://proceedings.neurips.cc/paper/2019/file/32e0bd1497aa43e02a42f47d9d6515ad-Paper.pdf) and [paper](/https://arxiv.org/pdf/1804.11285.pdf). However, our paper provides theoretical justifications for **the benefits of outlier mining (See Proposition 1 on page 4) in the context of OOD, which is entirely new**. We believe that this is a big delta for theory, and we want to know the reviewer's thoughts if this is deemed otherwise.
> > >
> > > 2. The reviewer challenged the novelty of hard negative mining, quote: *"Hard negative mining is pretty standard in many learning domains."*, and further, quote: *"In summary, the idea of using hard examples is not new. "*. While we understand where this comment comes from, and also that hard example idea has been studied in settings other than OOD, previous literature of OOD detection using auxiliary outlier data ([1][2][3])  has **never** utilized this idea. The current work has provided strong evidence, both theoretically and empirically, of the utility in doing so. To this end, we believe that **the evaluation of novelty should be established in the appropriate research context**. As a perhaps subjective analogy, the famous Yao's minimax principle in the context of randomized algorithms is essentially nothing but a simple application of von Neumann's minimax principle, yet it is important and very novel because of its utility in the new context.
> > >
> > > 3. The reviewer challenged the quality of our experiments, quote: *"a more robust empirical evaluation… would have been useful"*, and quote: *"Its empirical evaluation **appears to behind** the state-of-the-art in OOD detection as well as adversarial defense"*, and challenged that we lack experiments compared with SOTA. First, as a fact check, our empirical results **achieve state-of-the-art results, significantly outperforming 8 competitive OOD-detection methods across 4 evaluation tasks and 6 OOD datasets. The extent of thoroughness in evaluation is beyond any prior work we are aware of**. We would like to emphasize that our evaluation include the exact setting of those work, which includes:  (1) Non-adversarial setting: We compared with ODIN, Mahalanobis, OE, SOFL (comparable and often better) (2) Adversarial setting with $L_\infty$ PGD attack: We compared with ACET, CCU, ROWL (better with a significant margin, e.g. **with over 53.9% FPR reduction over current SOTA ACET[3]**). Besides these existing settings, we have also considered challenging new settings (and show significant robustness) with common corruptions that are unseen during training time. We refer the reviewer to the 5th paragraph of the introduction, Table 1 in Section 5.2, and an extensive amount of empirical results from pp. 29--pp. 40. We sincerely want to learn from the reviewer in what (concrete) aspects our evaluation should be more robust.
> > >
> > > 4. The reviewer challenged that there is no formal definition of OOD, quote: *"one challenge is the lack of a formal definition of OODs."* We have provided a definition in Section 2 "Problem Statement", and have stated clearly the tasks to evaluate the robustness. To this end, we remark that many previous works have studied  **exactly the same adversarial OOD tasks**, we have described and motivated this in the 2nd paragraph in the introduction, with a series of papers cited for that purpose (all appeared in major ML conferences [3][4][5]). To this end, we remark that the consideration of common corruptions and their composition with adversarial examples are very natural, as they clearly belong to the same issue of OOD robustness, which we have further clarified in revisions.
> > >
> > >
> > > [**Contribution and significance**] In summary, we believe our work contributes to the field by studying an **underexplored problem** (robust OOD detection), proposing an **unexplored and effective solution space** (informative outlier mining) for this problem that establishes SOTA performance. We also provided theoretical analysis and guarantees that support the proposed method. Furthermore, our work opens up an **interesting and promising** future direction for OOD detection to consider using auxiliary outlier data more effectively.
> > >
> > > [1] D. Hendrycks, M. Mazeika, and T. G Dietterich. Deep anomaly detection with outlier
> > > exposure. ICLR 2019
> > >
> > > [2] S. Mohseni, M.Pitale, JBS Yadawa, and Z. Wang.  Self-supervised learning for
> > > generalizable out-of-distribution detection. AAAI 2020
> > >
> > > [3] M. Hein, M. Andriushchenko, and J. Bitterwolf. Why relu networks yield high confidence predictions far away from the training data and how to mitigate the problem. CVPR 2019.
> > >
> > > [4] V. Sehwag, A. Nitin Bhagoji, L. Song, C. Sitawarin, D. Cullina, M. Chiang, and P. Mittal. Analyzing the robustness of open-world machine learning. 2019.
> > >
> > > [5] J. Bitterwolf, A. Meinke, and M. Hein. Provable worst case guarantees for the detection of out-of-distribution data. NeurIPS 2020.

---

> > > > ### Comment · AnonReviewer1 · 2020-11-24
> > > > **After reading the revised draft and the appendix ...**
> > > >
> > > > The observations of this  reviewer are just objective statements meant to help identify the strengths and weaknesses of the proposed contributions, and also to help improve each paper individually. The reviewer would request the authors not to view them as "The reviewer challenged ... ".   The reviewer went over the revised draft and the response from the authors. The reviewer did not find any new information, but the authors have integrated some suggestions in the revised draft which is helpful in improving the clarity of the paper.
> > > >
> > > > The reviewer again emphasizes that the use of hard examples is not entirely novel and it is not surprising that an outlier exposure approach would benefit from extreme outliers even if one accepts outlier exposure to be a realistic and practical approach to detecting OODs.  For now, let us set aside the experimental evaluation or the choice of simple toy example for theoretical analysis or the supervised formulation for detecting outliers.
> > > >
> > > > These are concerns but the central contribution of the work appears to be the theoretical analysis for the simple case of a Gaussian distribution. Clarification of its novelty would have been enough for this reviewer to further raise the score.  Unfortunately, it still appears to be a 'direct adoption of" / 'heavily inspired from' the toy setting and results from  https://proceedings.neurips.cc/paper/2019/file/32e0bd1497aa43e02a42f47d9d6515ad-Paper.pdf  and https://arxiv.org/pdf/1804.11285.pdf  (as mentioned in my previous review) without explicit identification of what is new - there are a bunch of places where these two papers are mentioned followed with claim that some modifications were needed.
> > > >
> > > > Let me draw attention to a few such instances which leave the novelty of the theoretical analysis unclear. In the revised draft, authors mention that: "Different from previous work by Schmidt et al. (2018) and Carmon et al. (2019), our analysis gives rise to a separation result with or without informative outlier mining for OOD detection." This "separation result" is then never defined as a term anywhere else in the paper.  "Separation result" has a standard meaning in complexity theory and such terms should be used in the right context or else explained later in the paper. But the reviewer can guess what the authors mean by separation result here - it is about the benefit of using outliers and extreme outliers for supervised OOD detection compared to not using them.  Let us now consider the case of the separation of the in-distribution sample sizes with and without auxiliary OOD data for training. This case is the same as Schmidt et al. (2018) and Carmon et al. (2019).   Looking at the text around Equation 30 in the revised draft,
> > > > "It has been shown that (Theorem 6 in (Schmidt et al., 2018) or Theorem 1 in (Carmon et al., 2019)) that when ....... Then the above robust classification problem can be reduced to this variant of robust OOD, by viewing the in-distribution data as with label +1 and viewing outliers as with label −1. "
> > > > So, the simple argument (straightforward and not really something that we can claim as a novel contribution) for porting results from these existing papers to the result in this draft is to think of outlier detection as supervised problem and then having +1/-1 labels for in-distr and OOD data. This allows directly importing results from the above two papers and hence, it is not a novel contribution.  Existence of robust classifiers and benefit of auxiliary OOD data " largely follows that in (Carmon et al., 2019) with some slight modification." as authors note in the Appendix. Elaboration of this modification would be useful. Yes, looking at supervised learning between OOD and normal samples instead of class A and class B, and thinking about robust  supervised identification of OOD vs the robust classification of A/B are different applications - but the theoretical analysis is  isomorphic.

---

> > > > > ### Author Response · Authors · 2020-11-24
> > > > > **Quick clarifications**
> > > > >
> > > > > We thank the reviewer for reading our revised draft and replied in such detail, which is truly appreciated. We want to list below some quick clarifications:
> > > > >
> > > > > 1. Our main theory novelty is **Proposition 1 and its proof (page 4 and equation (3)), which gives an upper bound with outlier mining**. This is our main result and what we highlighted in the revised draft. We'd love to hear your thoughts, and/or evaluation, about it.
> > > > >
> > > > > 2. You are right that the lower bound part (Proposition 3) basically follows from previous work by a reduction. To this end, we emphasize we respect the existing work and have clearly stated where we use the existing results for our proof, as you have also observed and cited.
> > > > >
> > > > > 3. We did state what we mean by separation in Section 2, with bolded text paragraphs. What we mean is that a simple detector can be failed with non-informative data, but can then be easily fixed with outlier mining. You are right that this is not the standard usage of "separation" used in complexity theory (because we did not prove that no algorithm can work without outlier mining). We will clarify this in the next version (either with better wording, or clarify more clearly what separation means in this particular context).
> > > > >
> > > > > 4. In the risk of repeating ourselves, we believe that research should be evaluated within an appropriate context. The vast literature of OOD has never explored hard example mining (to the best of our knowledge), so if one identifies this opportunity and shows that it works very well and establishes a new state of the art (we have performed, again to us, comprehensive evaluation), then we believe it constitutes meaningful contributions. To this end, identifying and reusing this simple technique, to us, is an advantage of this work, not a disadvantage.
> > > > >
> > > > > Again, we appreciate your input, which has truly helped us improve this work.

---

> ### Author Response · Authors · 2020-11-18
> **Follow up**
>
> Dear reviewer,
>
> We thank you again for the constructive feedback. We believe your comments raised have been addressed in our response and updated draft. We'd like to kindly follow up and clarify any remainder confusion. Your feedback has been very important and valuable for us to improve the work!
>
> Thank you,
>
> Authors of Paper1050

---

### Official Review · AnonReviewer2 · 2020-10-28
**An out of distribution data detector that selects informative OOD data during training so that the performance is good and robust.**

**Rating:** 7
**Confidence:** 3

**Review:**

1. The paper presents a lot of theory but insufficient evidence. It only employs limited image data (SVHN, CIFAR variants). The paper should be clear that the scope is limited to well-known image datasets only. This is because the approach is dependent on auxiliary data which is available for the image datasets. It is not clear if the approach might be more generally applicable to (say) network traffic, credit card transactions, natural language, etc. It would be better to include other types of data and along with auxiliary data generated through more generic means.

2. The presentation of the theory in Section 3 is not proper and makes it too simplistic. A better presentation should first present the general result first (rather than leave it to Appendix) and then show how that applies to a specific case such as Gaussian distribution.

3. Propositions 1, 2, 3: There seems to be a disconnect: what is the significance of n_0 in Proposition 1? What is its connection to the algorithm?

If we try to do a ballpark estimate with reasonable range of values: n_0 ~ 100, d ~ 50, \epsilon = 1/2, then the statement basically says that if n < 2 (optimistically), then we will have large errors. This type of limit on n is not very useful in practice as it is too loose. In fact, it seems that Proposition 1 would only be useful if d is very large. The paper should clarify that.

By and large, the Propositions 1, 2 and 3 are not helping much as the messages they are trying to convey (that auxiliary data helps and large number of samples are more accurate) are already well known. They do not add new knowledge. These might be moved to appendix.

4. Appendix B2, the general error bound (Proposition 5): The assumption is that d(Q_x, U_x) is small. The proposition basically states that if Q_x is similar to U_x, then a small FPR on U_x will also have a small FPR on Q_x. This is rather trivial. It is *much* harder in practice to get a U_x that is similar to Q_x. A more interesting case would be to show that even if U_x was not very similar to Q_x, then is the FPR within reasonable limit?

---

> ### Author Response · Authors · 2020-11-12
> **Clarify the theoretical results and emphasize the strong empirical results**
>
> We would like to clarify some concerns of the reviewer about our theoretical results, and also kindly draw the attention of the reviewer to our strong empirical results.
>
> Re: empirical evidence
>
> Our evaluation is primarily focusing on image data since the problem on image data alone already requires extensive study and we would like to provide a thorough evaluation. In fact, **most existing papers similarly focus on images** (see for example, [1][2][3][4]). We emphasize that we have provided extensive empirical evaluation, including those considered in previous work and also new ones introduced by us. In the evaluation, our method **achieves state-of-the-art results, significantly outperforming 8 competitive OOD-detection methods across 4 evaluation tasks and 6 OOD datasets**. **The extent of thoroughness in evaluation is beyond any prior work we are aware of**. Finally, we believe our insights and method can be applied to other types of data, which is left for future work. We have highlighted this in our revision (**see Section 5**).
>
> Re: theoretical results
>
> TL;DR Your feedback and suggestions have greatly helped improve our theoretical section. We revised the theory section (Section 3) to highlight our theoretical contributions. Specifically, we focus on presenting the error bound with outlier mining for the Gaussian data model, which is **completely new** and **directly connects to our strong empirical results in Section 5**. Below we address each concern in detail.
>
> 1. Our analysis in the Gaussian data model highlight the importance of our key contribution informative outlier mining (by comparing Prop. 1, 2 to Prop. 3). Therefore, we decided to present it in the main text and include the generic bound in the appendix due to space limitations. We would be happy to move the generic bound to the main text when more space is allowed. The simplicity of the Gaussian model helps to illustrate the intuition why informative outlier mining is important. Our analysis can be generalized to a more general setting at the cost of more calculations and more complicated bounds. For example, we can replace each Gaussian with a mixture of sub-Gaussians, which is believed to be a good model for simple practical datasets like MNIST. However, the complication there will hinder the intuition rather than help the illustration.
>
> 2. The significance of $n_0$: for each $n_0>0$, one can construct a corresponding family of data distributions for which our bounds in Prop. 1 to 3 hold. Roughly speaking, $n_0$ determines the number of labeled data needed for learning over the corresponding family. This allows us to compare the propositions on the same family of data distributions, and different errors of different methods there thus highlight the importance of our outlier mining technique. Indeed, our bounds are most meaningful for high dimensional data with sufficiently large $d$. Please see our discussion in the paragraph after Prop. 2. We would like to argue that the image data typically are in high dimensions of up to millions, for which our bounds are appropriate. We disagree that the messages in the propositions are already known. For example, comparing Prop 1 and Prop. 3 shows there are cases of limited labeled data, where without outliers the detector can simply fail, while with outlier mining it gets small errors. **As far as we know, this has not been considered or formalized**.
>
> 3. A small divergence $d_G(U_X, Q_X)$ does not mean $Q_X$ is similar to $U_X$ in the typical sense. It only means $Q_X$ and $U_X$ align with respect to G, that is, the error rankings of hypotheses in G are roughly similar on $U_X$ and $Q_X$. This is much weaker than that $Q_X$ is similar to $U_X$. Please see the two paragraphs before the Prop. 5. Furthermore, we also provide an illustrative example after Prop. 5 in Figure 3. In this example, $Q_X$ and $U_X$ even have completely disjoint supports but $d_G(U_X, Q_X)$ is small. Finally, our empirical results demonstrate that even if $U_X$ is not very similar to $Q_X$, the proposed method ATOM could still achieve a small FPR (e.g. on the CIFAR-10 experiments, the auxiliary dataset ImageNet-RC doesn’t contain images that are similar to the images in the test OOD dataset SVHN).
>
> [1] Sina Mohseni, Mandar Pitale, JBS Yadawa, and Zhangyang Wang. Self-supervised learning for
> generalizable out-of-distribution detection. AAAI 2020.
>
> [2] Shiyu Liang, Yixuan Li, and Rayadurgam Srikant. Enhancing the reliability of out-of-distribution
> image detection in neural networks. ICLR, 2018.
>
> [3] Kimin Lee, Kibok Lee, Honglak Lee, and Jinwoo Shin. A simple unified framework for detecting
> out-of-distribution samples and adversarial attacks. In NeurIPS 2018.
>
> [4] Matthias Hein, Maksym Andriushchenko, and Julian Bitterwolf. Why relu networks yield high confidence predictions far away from the training data and how to mitigate the problem. In CVPR 2019.

---

> ### Author Response · Authors · 2020-11-18
> **Follow up**
>
> Dear reviewer,
>
> We thank you again for the constructive feedback. We believe your comments raised have been addressed in our response and updated draft. We'd like to kindly follow up and clarify any remainder confusion.
> Your feedback has been very important and valuable for us to improve the work!
>
> Thank you,
>
> Authors of Paper1050

---

> ### Comment · AnonReviewer2 · 2020-11-22
> **Updates strengthen the paper**
>
> I thank the authors for updates to the papers and I believe the paper is now much stronger.

---

> > ### Author Response · Authors · 2020-11-22
> > **Thank you!**
> >
> > We are really glad to hear the concerns have been addressed.
> >
> > Once again thank you for the great comments which helped us strengthen our paper!

---

### Official Review · AnonReviewer4 · 2020-10-30
**This paper provides a theoretically motivated method Adversarial Training with informative Outlier Mining (ATOM), which can improve the robustness of OOD detection. It conducts extensive experiments and achieves impressive results under a broad family of OOD evaluation tasks.**

**Rating:** 7
**Confidence:** 3

**Review:**

This paper provides theoretical analysis formalizing the intuition of mining hard outliers for improving the robustness of OOD detection. Motivated by the theoretical analysis, it proposes Adversarial Training with informative Outlier Mining (ATOM) and achieves SOTA results on both clean and perturbed OOD inputs with extensive experiments compared to a bunch of baselines. It also provides an ablation study on the effectiveness of adversarial training, which further proves its effectiveness. I think it's a good paper overall and should be accepted.

---

> ### Author Response · Authors · 2020-11-18
> **Thank you for the positive feedback!**
>
> We thank R4 for the positive review. Yes, our method ATOM achieves state-of-the-art performance on both clean OOD and adversarial OOD evaluation tasks and our novel theoretical analysis formalizes the intuition of mining hard outliers for improving the robustness of OOD detection. We think our work opens up an interesting and promising future direction for OOD detection to consider using auxiliary outlier data more effectively.

---

### Author Response · Authors · 2020-11-17
**Revision #1: Clarifying theory novelty and significance of empirical results**

We thank the reviewers for their constructive reviews. We have responded to each reviewer in detail. We have incorporated feedback from all reviewers in our updated draft. Below we summarize our changes.

[-->R1, R2] We revised the experimental section (*Section 5*) to highlight our strong empirical results.  In the evaluation, our method **achieves state-of-the-art results, significantly outperforming 8 competitive OOD-detection methods across 4 evaluation tasks and 6 OOD datasets**. **The extent of thoroughness in evaluation is beyond any prior work we are aware of**. We would like to emphasize that our evaluation includes comparisons with the current state of the art, in the exact same setting of those work, which includes: (1) Non-adversarial setting: We compared with ODIN, Mahalanobis, OE, SOFL (comparable and often better) (2) Adversarial setting with $L_\infty$ PGD attack: We compared with ACET, CCU, ROWL based detection (better with a significant margin, e.g. **with over 53.9% FPR reduction over current SOTA ACET**). (3) Besides these existing settings, we have also considered challenging new adversarial settings such as common image corruptions that are unseen during training time (and show significant robustness).

[-->R1, R2, R3] We revised the theory section (*Section 3*) to highlight our theoretical contributions. Specifically, we focus on presenting the error bound with outlier mining for the Gaussian data model, which is **completely new** and distinct from previous works by Carmon et al. 2019 and Schmidt et al. 2018. Our theory novelty is **Proposition 1 and its proof (page 4 and equation (3)), which gives an upper bound with outlier mining**. We also followed R2’s suggestion to present the general error bound first and then show the theoretical results under the Gaussian data model in Appendix A. Due to space limitations, we prioritize the analysis central to outlier mining in the main paper.

[-->R1, R3] We expanded the discussion of related work about hard example mining (see both introduction in *Section 1* and related work in *Section 6*). Although hard example mining has been used in various other domains (e.g. object detection and deep metric learning), to the best of our knowledge, we are **the first to explore the novel connection between hard example mining and out-of-distribution detection**. More importantly, our work demonstrates, both theoretically and empirically, hard example mining significantly improves the generalization and robustness of out-of-distribution detection;

[Contributions and significance] In summary, we believe our work contributes to the field by studying an **underexplored problem** (robust OOD detection), proposing an **unexplored and effective solution space** (informative outlier mining) that establishes state-of-the-art performance. We also provided theoretical analysis and guarantees that support the proposed method. Furthermore, our work opens up an **interesting and promising** future direction for OOD detection to consider using auxiliary outlier data more effectively.

We thank the reviewers again for the constructive comments which greatly helped improve our work! We hope that these revisions can help reviewers assess our contributions.

---

### Decision · Program_Chairs · 2021-01-07
**Final Decision**

**Decision:**

Reject

**Comment:**

In this paper, the authors studied a robust method for detecting out-of-distribution (OOD) instances. OOD instance detection is an important practical problem, and multiple reviewers recognized the proposed approach is interesting. However, it was the common opinion of several reviewers that the main theoretical analysis was imported from existing studies, and the novelty is not sufficiently high. It was also observed that the relationship between the proposed method and closely related studies was not properly discussed. Although this point has been improved in the revision, a reviewer and area chair still concern that enough evidence is not provided for some of the points the authors claim as advantages over existing studies. Although the proposed method is interesting and could be an important contribution to the ICLR community, the current paper needs non-trivial revision before publication.